# On the Effect of Misspecifying the Embedding Dimension in Low-rank Network Models

**Roddy Taing** [1]    **Keith Levin** [1]

## Abstract

As networks have become ubiquitous in the sciences, there has been growing interest in network models whose structure is driven by latent node-level variables in a (typically low-dimensional) geometric space. These "latent positions" are often estimated via embeddings, whereby the nodes of a network are mapped to points in Euclidean space so that "similar" nodes are mapped to nearby points. Under certain model assumptions, these embeddings are consistent estimates of the latent positions, but most such results require the embedding dimension to be chosen correctly. Methods for choosing the embedding dimension have been studied extensively, but little is known about the behavior of embeddings when the dimension is misspecified. In this work, we provide a theoretical description of the effects of dimension misspecification under the random dot product graph, a class of latent space network models that includes several widely-used network models, most notably the stochastic blockmodel, as special cases. We show that when the dimension is chosen too large, consistent estimation still holds, albeit at a slower rate than when the embedding dimension is chosen correctly. On the other hand, when the dimension is chosen too small, there is a fundamental estimation error lower bound that need not go to zero in the large-network limit. A range of synthetic data experiments support our theoretical results. Our main technical result, which may be of independent interest, is a generalization of earlier work in random matrix theory showing that all non-signal eigenvectors of a low-rank matrix subject to additive noise are delocalized.

[1]Department of Statistics, University of Wisocnsin–Madison, Madison, WI USA. Correspondence to: Roddy Taing <rtaing@wisc.edu>.

*Proceedings of the $43^{rd}$ International Conference on Machine Learning*, Seoul, South Korea. PMLR 306, 2026. Copyright 2026 by the author(s).

## 1. Introduction

Networks, which describe collections of interacting entities, have become central to a range of applications in the past twenty years. As mathematical objects, networks are graphs, in which vertices (also called nodes; we use the terms interchangeably here) correspond to the entities under study, and edges describe interactions among these entities. Networks arise in genomics (van Dam et al., 2017; Marku & Pancaldi, 2023; Treiber et al., 2010; Larremore et al., 2013), where they describe which pairs of genes are co-expressed or are involved in the same processes. In neuroscience, networks describe connectomes (Bullmore & Sporns, 2009; Sporns, 2012), in which vertices correspond to brain regions and edges encode which brain regions interact closely. In ecology, networks describe food webs and species co-occurrence (Guseva et al., 2022). In the social sciences, networks arise in disciplines ranging from political science (Porter et al., 2005) to sociology (Lazega, 2001) to economics (Chaney, 2014; Acemoglu et al., 2015; Elliott et al., 2014).

With the growing prevalence of network data, there has been a proliferation of accompanying statistical models, beginning with the stochastic blockmodel (SBM; Holland et al., 1983) and extenstions thereof (see, e.g., Airoldi et al., 2008; Karrer & Newman, 2011). Under these models, each vertex belongs to one of $r$ communities, and the probability of an edge joining a pair of vertices is determined by their community memberships: network structure is driven by latent node-level variables. Subsequent models have extended this idea to allow for geometric latent structure, in which each node has an associated *latent position* in $r$-dimensional Euclidean space. These models are often called latent space network models (though this term is sometimes reserved for the model in Hoff et al., 2002). Prominent examples include the random geometric graph (Gilbert, 1961; Méliot, 2019; García Trillos et al., 2020), wherein two nodes form an edge if the distance between their latent positions is below a set threshold; and the random dot product graph (RDPG; Young & Scheinerman, 2007; Athreya et al., 2018; Rubin-Delanchy et al., 2022), in which edge probabilities are determined by inner products between latent positions. These models are all special cases of the graphon (Borgs et al., 2008; Airoldi et al., 2013; Lei, 2021; Lovász, 2012).

A typical statistical task in latent space models is to estimate the latent positions of the nodes from an observed network. A common approach is to use *node embeddings*, which map the nodes of a network to a low-dimensional (typically Euclidean) space so that "similar" nodes are mapped to nearby points. Examples of node embeddings corresponding to different notions of node similarity include (Sussman et al., 2012; Eldridge et al., 2018; Rohe et al., 2011; Tang & Priebe, 2018; Grover & Leskovec, 2016), to name a few. Node embeddings are often used to obtain Euclidean representations of vertices for use in downstream tasks such as clustering (Rohe et al., 2011; Sussman et al., 2012), inference (Du & Tang, 2023; Zheng, 2025), and visualization (Modell et al., 2022), though our focus in the present work is on embeddings as estimators of latent positions.

The first step in constructing a node embedding is to choose the embedding dimension. The majority of results showing consistent estimation of latent positions assume this embedding dimension is chosen correctly (see, e.g., Athreya et al., 2018; Lei & Rinaldo, 2015; Cape et al., 2019a; Rohe et al., 2011; Jones & Rubin-Delanchy, 2020; Levin et al., 2019; 2022; Sussman et al., 2012; Cape et al., 2019b; Lyzinski et al., 2014; Rubin-Delanchy et al., 2022), though see (Lei, 2021; Tang & Cape, 2025), which instead rely on spectral properties of the latent structure. This latent dimension is typically (but not always; see Rubin-Delanchy, 2020; Athreya et al., 2021; Tang et al., 2022; Sansford et al., 2025) equal to the rank of the expected adjacency matrix conditional on the latent positions. In practice, the embedding dimension must be chosen based on data. This model selection problem is widely recognized by the statistical network analysis community as important and challenging, and it has received commensurate attention. The earliest methods relied on finding elbows in the scree plot of the adjacency matrix (Zhu & Ghodsi, 2006). More recent approaches include cross-validation and likelihood-based methods (Li et al., 2020; Chen & Lei, 2018; Chen et al., 2021a; Yang et al., 2021; Jiang et al., 2025; Wang & Bickel, 2017), parallel analysis (Hong & Cape, 2025; Hong et al., 2020), hypothesis testing (Bickel & Sarkar, 2015; Lei, 2016; Han et al., 2023), and thresholding approaches (Chatterjee, 2015).

Though the latent dimension can be chosen using the methods listed above, there is always the possibility that the embedding dimension is chosen incorrectly. Thus, there is a need to better understand how node embeddings behave under model misspecification. This question has received little attention in the literature (though see Fishkind et al., 2013, for an early result in the setting of the SBM). We consider this question under the RDPG. In particular, we examine the behavior of the adjacency spectral embedding (ASE; Sussman et al., 2012) when the embedding dimension is chosen incorrectly. We show that when the embedding dimension is too large, embeddings produced via ASE are still consistent, though possibly at a slower rate than when the dimension is chosen correctly, settling an open question posed in Section 7 of Athreya et al. (2018). We also prove an estimation lower bound for the setting where the embedding dimension is chosen too small, which captures the cost of failing to capture all signal dimensions present in the data. Our main technical contribution, which may be of independent interest, shows that under the RDPG, all eigenvectors of the adjacency matrix delocalize under mild conditions. Our results extend earlier work in random matrix theory (Erdős et al., 2013a) on the eigenvectors of a deterministic rank-one matrix subject to additive noise.

**Notation.** Before proceeding, we pause to establish notation. Matrices will be denoted by capital bold letters, e.g., $\mathbf{Z}$. $\|\mathbf{Z}\|$ and $\|\mathbf{Z}\|_F$ denote, respectively, the spectral norm and Frobenius norm of $\mathbf{Z}$. Given the full spectral decomposition of a $n \times n$ symmetric matrix $\mathbf{Z} = \mathbf{U}\Lambda\mathbf{U}^\top$, the eigenvalues and associated eigenvectors are sorted to be non-increasing. We write $\mathbb{O}_r$ to denote the set of all $r$-by-$r$ orthogonal matrices. For sequences $a_n, b_n > 0$, we use the notation $a_n \lesssim b_n$ to mean that for some $C > 0$, $a_n \leq Cb_n$ for sufficiently large $n$, and the notation $a_n \ll b_n$ to mean that $a_n/b_n \to 0$ in the large-$n$ limit. We use $a_n \asymp b_n$ to mean that there exist constants $c, C > 0$ such that $ca_n \leq b_n \leq Ca_n$ for all suitably large $n$. When $a_n$ and $b_n$ are random quantities, we use these same conventions in the probabilistic sense. For example, $a_n \lesssim b_n$ means that $a_n/b_n$ converges in probability to a quantity bounded by a constant.

## 2. Setup and Modeling

We consider network models in which the adjacency matrix $\mathbf{A}$ is given by a low-rank signal matrix subject to additive noise, often termed "signal-plus-noise" models in the literature (see, e.g., Cape et al., 2019a; Yan & Levin, 2024):

$$\mathbf{A} = \mathbf{P} + \mathbf{E} \in \mathbb{R}^{n \times n}, \qquad (1)$$

where $\mathbf{E}$ is a mean-zero noise matrix and $\mathbf{P}$ is an $n \times n$ symmetric matrix of rank $r$ with spectral decomposition

$$\mathbf{P} = \mathbf{U}_{1:r}\mathbf{S}_{1:r}\mathbf{U}_{1:r}^\top, \qquad (2)$$

where $\mathbf{S}_{1:r} = \mathrm{diag}(s_1, \ldots, s_r)$ is the matrix of non-zero eigenvalues $s_1 \geq s_2 \geq \cdots \geq s_r > 0$ and $\mathbf{U}_{1:r} \in \mathbb{R}^{n \times r}$ is the matrix of corresponding orthonormal eigenvectors.

In latent variable network models such as the SBM, RDPG and their variants discussed in Section 1, $\mathbf{P}$ is expressible as

$$\mathbf{P} = (\sqrt{\rho_n}\mathbf{X})(\sqrt{\rho_n}\mathbf{X})^\top,$$

where the rows of $\mathbf{X} \in \mathbb{R}^{n \times r}$ encode $r$-dimensional vertex-level latent variables and $\rho_n \in [0, 1]$ can be interpreted as a signal strength parameter. Based on the spectral decomposition in Equation (2), we may take

$$\sqrt{\rho_n}\mathbf{X} = \mathbf{U}_{1:r}\mathbf{S}_{1:r}^{1/2}. \qquad (3)$$

We consider the task of recovering $\sqrt{\rho_n}\mathbf{X}$ from the observed adjacency matrix $\mathbf{A}$. Equation (3) suggests a natural approach, namely using the leading eigenvalues and eigenvectors of $\mathbf{A}$ in place of those of $\mathbf{P}$. This motivates the widely-used adjacency spectral embedding, first considered in Sussman et al. (2012).

**Definition 2.1** (Adjacency Spectral Embedding; Sussman et al. (2012)). Given a real, symmetric $\mathbf{A} \in \mathbb{R}^{n \times n}$ with spectral decomposition

$$\mathbf{A} = \hat{\mathbf{U}}\hat{\mathbf{S}}\hat{\mathbf{U}}^\top,$$

we define the $d$-dimensional adjacency spectral embedding (ASE) of $\mathbf{A}$ to be

$$\hat{\mathbf{X}}_{1:d} = \hat{\mathbf{U}}_{1:d}|\hat{\mathbf{S}}|^{1/2}_{1:d} \in \mathbb{R}^{n \times d},$$

where $\hat{\mathbf{S}} \in \mathbb{R}^{d \times d}$ is a diagonal matrix containing the $d$ largest-magnitude eigenvalues of $\mathbf{A}$ and $\hat{\mathbf{U}}_{1:d} \in \mathbb{R}^{n \times d}$ has as its columns the corresponding orthonormal eigenvectors. We use $|\hat{\mathbf{S}}|$ since the largest-magnitude eigenvalues of $\mathbf{A}$ need not be positive, even if the diagonal entries of $\mathbf{S}$ are. The hat notation ($\hat{\mathbf{U}}$, $\hat{\mathbf{S}}$, etc.), is meant to evoke the fact that these quantities estimate their analogues in Equation (3).

Much recent work on low-rank network models evaluates estimation error via the maximum Euclidean norm between the true structure $\sqrt{\rho_n}\mathbf{X}$ and its estimate (see, e.g., Athreya et al., 2018; Rubin-Delanchy et al., 2022; Levin et al., 2019; Cape et al., 2019b; Levin et al., 2022). A complication arises from the fact that for any $\mathbf{W} \in \mathbb{O}_r$,

$$\mathbf{X}\mathbf{X}^\top = (\mathbf{X}\mathbf{W})(\mathbf{X}\mathbf{W})^\top. \tag{4}$$

That is, any orthogonal rotation of $\mathbf{X}$ gives rise to the same expected adjacency matrix $\mathbf{P}$. Accounting for this non-identifiability, a good estimator $\widetilde{\mathbf{X}} \in \mathbb{R}^{n \times r}$ will be one that achieves small values of

$$\min_{\mathbf{W} \in \mathbb{O}_r} \left\| \widetilde{\mathbf{X}}\mathbf{W} - \sqrt{\rho_n}\mathbf{X} \right\|_{2,\infty},$$

where $\|\mathbf{Z}\|_{2,\infty}$ is defined to be the maximum Euclidean norm of the rows of matrix $\mathbf{Z}$. Convergence in this $(2, \infty)$-norm implies uniform recovery of vertex-level structures encoded in the rows of $\mathbf{X}$ (Lyzinski et al., 2014). Such convergence can be guaranteed under a number of latent variable network models, among which we consider the random dot product graph (RDPG; Athreya et al., 2018; Young & Scheinerman, 2007), a model in which network structure is driven by latent vertex-level geometric structure.

**Definition 2.2** (Random Dot Product Graph). Let $F$ be a probability distribution on $\mathbb{R}^r$ with $\mathbf{X}_1, \mathbf{X}_2, \ldots, \mathbf{X}_n$ drawn i.i.d. according to $F$ and collect these into the rows of $\mathbf{X} \in \mathbb{R}^{n \times r}$. Let $(\rho_n)_{n=1}^\infty$ be a sequence with $\rho_n \in [0, 1]$ for all $n$. We say that $\mathbf{A}$ is distributed according to a *weighted RDPG* with latent positions $\mathbf{X}$ and signal strength $\rho_n$ if, conditional

on $\mathbf{X}$, $\mathbf{E} = \mathbf{A} - \rho_n\mathbf{X}\mathbf{X}^\top$ has independent entries (up to symmetry). When $F$ is such that $\mathbf{x}^\top\mathbf{y} \in [0, 1]$ for all $\mathbf{x}, \mathbf{y} \in \text{supp }\mathbf{F}$, we say that $\mathbf{A}$ is distributed according to a *binary RDPG* with latent positions $\mathbf{X}$ and sparsity parameter $\rho_n$ if

$$\mathbb{P}(\mathbf{A} \,|\, \mathbf{X}) = \prod_{i<j}(\rho_n\mathbf{X}_i^\top\mathbf{X}_j)^{A_{ij}}(1 - \rho_n\mathbf{X}_i^\top\mathbf{X}_j)^{1-A_{ij}}. \tag{5}$$

In either case, we write $(\mathbf{A}, \mathbf{X}) \sim \text{RDPG}(F, n)$ with dimension $r$ and parameter $\rho_n$ clear from context, and refer to the rows of $\mathbf{X}$ as the *latent positions* of the network.

The RDPG extends a number of widely-used network models. Taking the latent positions to be drawn from a suitably-chosen mixture of point masses recovers the SBM as a special case. The RDPG recovers the degree-corrected SBM (Karrer & Newman, 2011) as a special case by taking $\mathbf{X}_i = c_i Y_i$, where $c_i \in (0, 1)$ and $Y_i$ is randomly chosen to be one of $r$ distinct vectors $y_1, y_2, \ldots, y_r \in \mathbb{R}^r$. We recover the mixed-membership SBM (Airoldi et al., 2008) by letting each $\mathbf{X}_i$ be a convex combination of $y_1, y_2, \ldots, y_r$. The RDPG can be extended and generalized in a number of ways. For example, one may take edge probabilities to be given by a kernel function on the latent space $\kappa : \mathcal{X} \times \mathcal{X} \to [0, 1]$ (Tang et al., 2013). In the RDPG as given in Definition 2.2, $\kappa$ is the inner product and $\mathcal{X} = \mathbb{R}^r$. The positive semidefinite structure of $\mathbf{P} = \rho_n\mathbf{X}\mathbf{X}^\top$ can also be relaxed (Rubin-Delanchy et al., 2022). See Athreya et al. (2018) for an in-depth discussion of the RDPG.

**Remark 2.3.** The parameter $\rho_n$ in Definition 2.1 plays distinct but related roles in the weighted and binary RDPGs. In the weighted RDPG, $\rho_n$ describes a signal strength: if $\rho_n = 1$, the eigenvalues of $\mathbf{P}$ grow linearly with $n$ (see Lemma C.2 in Appendix C). In the binary RDPG, $\rho_n$ describes sparsity: as the number of vertices $n$ grows, the probability of an edge appearing between two particular nodes shrinks to zero. Having $\rho_n$ shrink to zero as the network size grows allows us to make the estimation problem harder by having the eigenvalues of $\mathbf{P}$ grow sublinearly. The key distinction between these two settings is that in the weighted RDPG, we allow for the variance of the entries of $\mathbf{E}$ to be specified separately from the signal strength $\rho_n$, while in the case of the binary RDPG, the variances of the edges, being Bernoullis, depend on $\rho_n$.

### 2.1. Model Misspecification

The consistency of the ASE has been studied extensively in the setting where the embedding dimension is correctly specified (i.e., $d = r$; see Sussman et al., 2012; Lyzinski et al., 2017; Levin & Lyzinski, 2017; Tang & Priebe, 2018; Athreya et al., 2018; Levin et al., 2021; 2022; Rubin-Delanchy et al., 2022). In practice, the model rank $r$ is typically unknown and must be estimated from data. Methods to estimate the embedding dimension have been widely

explored in the literature, as discussed in Section 1. Even with the best embedding dimension estimation techniques, however, it is still possible to misspecify the embedding dimension. Thus, our goal in this paper is to characterize the behavior of the ASE when the embedding dimension is incorrectly chosen. To the best of our knowledge, this is the first work to do so.

Showing consistency of the ASE in the $(2,\infty)$-norm when the true model dimension $r$ is known requires bounding

$$\min_{\mathbf{W}\in\mathbb{O}_r}\left\|\hat{\mathbf{X}}_{1:r}\mathbf{W}-\sqrt{\rho_n}\mathbf{X}_{1:r}\right\|_{2,\infty}, \qquad (6)$$

where the minimization over $\mathbf{W}$ accounts for the non-identifiability of the latent positions. Quantities like that in Equation (6) have been controlled in a range of settings when $r$ is known (Lyzinski et al., 2017; Levin et al., 2022; 2019; Rubin-Delanchy et al., 2022; Cape et al., 2019b). These results rely on the Davis-Kahan $\sin\Theta$ theorem or variants thereof (Davis & Kahan, 1970; Yu et al., 2014; El-dridge et al., 2018), which requires a non-zero eigengap. When we embed into $d\neq r$ dimensions, these approaches are no longer feasible. In particular, when the embedding dimension is chosen too large, the ASE includes eigenvectors whose associated population eigenvalues are zero, resulting in a zero eigengap. Thus, if we are to characterize the behavior of the ASE when the embedding dimension is misspecified, different tools are needed.

In adapting the quantity in Equation (6) to the setting where the embedding dimension is incorrectly chosen, we must account for the fact that our estimate and estimand are of different dimensions. When choosing our embedding dimension too small (i.e., $d < r$), we pad our estimate $\hat{\mathbf{X}}_{1:d}$ with additional columns of zeros so that it can sensibly be compared with the $n$-by-$r$ matrix $\mathbf{X}_{1:r}$. Similarly, when the embedding dimension is chosen too large (i.e., $d > r$), we pad $\mathbf{X}_{1:r}$ with additional columns of zeros to ensure that our matrix of embeddings and true latent positions are conformable. Hence, in order to study the setting where $d < r$, we define $\hat{\mathbf{X}}_{1:r}^{\circ} \in \mathbb{R}^{n\times r}$ to be a zero-padded version of $\hat{\mathbf{X}}_{1:d}$. That is, when $d - r < 0$, we define

$$\hat{\mathbf{X}}_{1:r}^{\circ} = \begin{bmatrix}\hat{\mathbf{X}}_{1:d} & \mathbf{0}_{d+1:r}\end{bmatrix}. \qquad (7)$$

Similarly, when $d > r$, we define $\mathbf{X}_{1:d}$ to be

$$\mathbf{X}_{1:d} = \begin{bmatrix}\mathbf{X}_{1:r} & \mathbf{0}_{r+1:d}\end{bmatrix} \in \mathbb{R}^{n\times d}. \qquad (8)$$

In understanding the estimation error of the ASE, in the setting where the embedding dimension is chosen too large ($d > r$), we perform the following decomposition, ignoring the orthogonal non-identifiability for the time being:

$$\left\|\hat{\mathbf{X}}_{1:d}-\sqrt{\rho_n}\mathbf{X}_{1:d}\right\|_{2,\infty}$$
$$\leq \underbrace{\left\|\hat{\mathbf{X}}_{1:r}-\sqrt{\rho_n}\mathbf{X}_{1:r}\right\|_{2,\infty}}_{\text{Controlled via existing techniques}}+\underbrace{\left\|\hat{\mathbf{X}}_{r+1:d}\right\|_{2,\infty}}_{\text{Trailing dimensions}}.$$

From this decomposition, we see that there are two terms, one corresponding to the rate that would hold if the embedding dimension were correctly chosen, and the other corresponding to the "trailing" dimensions. We formalize this intuition in the following lemma, which gives results both for the case where the embedding dimension is chosen too large and for the case where it is chosen too small. A proof is given in Appendix B.

**Lemma 2.4** (Misspecified Model Bounds). *Suppose that* $\mathbf{A} = \rho_n\mathbf{X}\mathbf{X}^{\top} + \mathbf{E}$, *and denote the $d$-dimensional ASE of* $\mathbf{A}$ *as* $\hat{\mathbf{X}}_{1:d} = \hat{\mathbf{U}}_{1:d}|\hat{\mathbf{S}}|_{1:d}^{1/2}$. *When* $d < r$,

$$\min_{\mathbf{W}\in\mathbb{O}_d}\left\|\hat{\mathbf{X}}_{1:r}^{\circ}\mathbf{W}-\sqrt{\rho_n}\mathbf{X}_{1:r}\right\|_{2,\infty} \geq \sqrt{\sum_{j=d+1}^{r}\frac{s_j}{n}}.$$

*On the other hand, when* $d > r$, *suppose that there exists a sequence of* $\mathbf{W}^{*} \in \mathbb{O}_r$ *such that*

$$\left\|\hat{\mathbf{X}}_{1:r}\mathbf{W}^{*}-\sqrt{\rho_n}\mathbf{X}_{1:r}\right\|_{2,\infty} \lesssim \phi_n, \qquad (9)$$

*for some sequence* $(\phi_n)_{n=1}^{\infty}$. *Then there exists a sequence of* $\mathbf{W} \in \mathbb{O}_d$ *such that*

$$\left\|\hat{\mathbf{X}}_{1:d}\mathbf{W}-\sqrt{\rho_n}\mathbf{X}_{1:d}\right\|_{2,\infty} \lesssim \phi_n + \left\|\hat{\mathbf{U}}_{r+1:d}\right\|_{2,\infty}\|\mathbf{E}\|^{1/2}.$$

The term $\phi_n$ in Equation (9) captures the estimation rate that would hold if the embedding dimension were correctly chosen. As mentioned earlier, this setting has been studied extensively in the literature, and the precise behavior of $\phi_n$ is known under several different variants of the RDPG corresponding to different modeling choices for $\mathbf{X}$ and the edge noise $\mathbf{E}$. We collect two representative results below.

For binary networks, we present the following bound, adapted from Lyzinski et al. (2017) (see Rubin-Delanchy et al., 2022, for a similar result for the "generalized" RDPG).

**Theorem 2.5** (Lyzinski et al. (2017), Theorem 5). *Suppose that* $(\mathbf{A}, \mathbf{X}) \sim \mathrm{RDPG}(F, n)$, *then there exists a sequence of* $\mathbf{W} \in \mathbb{O}_r$ *such that*

$$\left\|\hat{\mathbf{X}}_{1:r}\mathbf{W}-\sqrt{\rho_n}\mathbf{X}_{1:r}\right\|_{2,\infty} \lesssim r^{1/2}(\log n)^2(\rho_n n)^{-1/2}.$$

The following theorem, adapted from Levin et al. (2022), provides a bound for $\phi_n$ in the weighted network setting, when the entries of $\mathbf{E}$ obey sub-gamma tail decay (see Chapter 2 in Boucheron et al., 2013).

**Theorem 2.6** (Levin et al. (2022), Theorem 6). *Suppose* $\mathbf{A} = \mathbf{P} + \mathbf{E} = \rho_n\mathbf{X}\mathbf{X}^{\top} + \mathbf{E}$, *and suppose that conditional on* $\mathbf{X} \in \mathbb{R}^{n\times r}$, *the entries of* $\mathbf{E}$ *are independent* $(\nu, b)$-*sub-gamma random variables, with* $n(\nu + b^2)\log^2 n \ll s_r^2$. *Then there exists a sequence of* $\mathbf{W} \in \mathbb{O}_r$ *such that*

$$\left\|\hat{\mathbf{X}}_{1:r}\mathbf{W}-\sqrt{\rho_n}\mathbf{X}_{1:r}\right\|_{2,\infty} \lesssim \frac{r(\log n)\sqrt{\nu + b^2}}{s_r^{1/2}}$$
$$+ \frac{rn(\log n)^2(\nu + b^2)s_1}{s_r^{5/2}}.$$

From Lemma 2.4, we see that showing consistency when the embedding dimension is chosen too large (i.e., $d > r$) amounts to controlling the $(2, \infty)$-norm of $\hat{\mathbf{X}}_{r+1:d}$. By Definition 2.1, $\hat{\mathbf{X}}_{r+1:d} \in \mathbb{R}^{n \times (d-r)}$ can be expressed as

$$\hat{\mathbf{X}}_{r+1:d} = \left[ \hat{u}_{r+1} \sqrt{|\hat{s}_{r+1}|} \quad \cdots \quad \hat{u}_d \sqrt{|\hat{s}_d|} \right].$$

Bounding the eigenvalues as $\hat{s}_{r+k} \leq \|\mathbf{E}\|$ for $k \in [d-r]$, we may apply one of several matrix concentration inequalities (see, e.g., Lei & Rinaldo, 2015; Erdős et al., 2013a; Vershynin, 2018), depending on the distribution of the entries of $\mathbf{E}$. The challenge lies in controlling the entries of the eigenvectors associated with these eigenvalues. Intuitively, the entries of the ASE associated with these eigenvectors must shrink to zero, as these are zero in the true latent positions $\mathbf{X}$. Since $\|\mathbf{E}\| \lesssim \sqrt{n}$ (ignoring log factors), the entries of $\hat{\mathbf{U}}_{r+1:d}$ must vanish. Many random matrix models produce eigenvectors whose entries are bounded as $C(\log n)^c / \sqrt{n}$ for some $c > 0$. These eigenvectors are called *delocalized*, and they have been widely explored in the random matrix literature (Erdős et al., 2007; 2013a; Erdős & Yau, 2017; Rudelson & Vershynin, 2015; O'Rourke et al., 2016; Benigni & Lopatto, 2020). To show delocalization in our setting of a real, symmetric matrix $\mathbf{A}$, we assume that $\mathbf{E}$ is an $n \times n$ real, symmetric matrix whose entries are mean-zero random variables, conditionally independent given $\mathbf{P}$ (up to symmetry). We make the following assumptions on the spectral properties of $\mathbf{P}$ and the moments of $E_{ij}$, both of which are allowed to vary with $n$:

(A1) The eigenvectors associated with non-zero eigenvalues of $\mathbf{P}$ are delocalized. That is, for some $\gamma \geq 0$ with no dependence on $n$,

$$|u_{ij}| \lesssim \frac{(\log n)^\gamma}{n^{1/2}}$$

for all $i \in [n]$, and $j \in [r]$ where $u_{ij}$ refers to the $i$-th entry of the $j$-th eigenvector of $\mathbf{P}$.

(A2) The signal strength obeys $\rho_n \gg 1/\sqrt{n}$.

(A3) The non-zero eigenvalues $s_1, \ldots, s_r$ of $\mathbf{P}$ are distinct.

(A4) For all $j \in [r]$, $s_j \asymp \rho_n n$.

(A5) $r = \text{rank}(\mathbf{P}) \lesssim (\log n)^\zeta$ for some $\zeta \geq 0$.

(A6) The eigengap $\Delta = \min_{j \in [r]} \min_{i \neq j} |s_i - s_j|$ obeys $\Delta \gtrsim \rho_n n$.

(A7) The entries of $\mathbf{E}$ satisfy the moment conditions

$$\mathbb{E}[E_{ij}] = 0, \quad \mathbb{E}[|E_{ij}|^2] = \sigma^2, \quad \text{and}$$

$$\mathbb{E}[|E_{ij}|^p] \leq \frac{C_1^p}{(q_n^*)^{(p/2-1)}} \quad (p > 2),$$

where $q_n^* \in [n^{-1}(\log n)^6, 1]$ is a parameter that controls the tail behavior of $\mathbf{E}$.

**Remark 2.7.** Assumption **A1** is similar to the incoherence condition often found in spectral methods (Chen et al., 2021b; Yan & Levin, 2024; Zheng, 2025). It is crucial for our main delocalization result to hold. In general, without such conditions, we do not necessarily expect eigenvectors of random matrices to delocalize (see O'Rourke et al., 2016). In the case of the RDPG, it can be shown that $\|\mathbf{U}_{1:r}\|_{2,\infty} \lesssim \sqrt{r/n}$ (see Corollary C.4 in the appendix). Assumption **A3** allows us to show that the $j$-th signal eigenvector is close to the $j$-th sample eigenvector. This is no longer true when the eigenvalues are repeated, but we expect that Assumption **A3** can be removed at the expense of increased notational complexity. In particular, one must modify the proof of Theorem 3.1 to keep track of all the eigenvectors that span an eigenspace associated to any eigenvalue with multiplicity greater than one, a task we leave to future work. Assumptions **A2** and **A6** are necessary to allow us to apply results from spectral methods (Chen et al., 2021b). We suspect that delocalization of the eigenvectors of $\mathbf{A}$ still holds even when these assumptions are relaxed, as suggested by results in Yan & Levin (2024). Finally, the moment conditions for $\mathbf{E}$ in Assumption **A7** allow for sub-exponential decay (see Remark 2.5 from Erdős et al., 2013a, for details). We note that Assumptions **A1** through **A6** are, by and large, standard in the statistical network modeling literature, though precise assumptions on the sparsity $\rho_n$ vary based on the intended downstream tasks. Assumption **A7** is less common, but amounts to an assumption on the tail decay of the edge noise distribution. See Levin et al. (2022) for an example of this in the RDPG.

## 3. Theoretical Results

We now state our main theoretical results for the case of weighted networks. These results rely on basic properties of the $(2, \infty)$-norm, as well as on fundamental results in spectral methods and random matrix theory. Our first result establishes that the trailing eigenvectors of the adjacency matrix $\mathbf{A}$ delocalize under the assumptions given above. A proof is given in Appendix D.

**Theorem 3.1.** *Let* $\mathbf{A} = \mathbf{P} + \mathbf{E}$, *and suppose Assumptions* (**A1**) – (**A7**) *hold. Then for all* $j \in [n]$,

$$\max_{\alpha > r} |\hat{u}_{j\alpha}| \lesssim \frac{r^2 (\log n)^{4+6\gamma}}{n^{1/2}}. \tag{10}$$

The proof of this result is based on the approach given in Erdős et al. (2013a), which establishes the semicircle law for the eigenvalues of $\mathbf{A}$ when $\mathbf{P}$ in Equation (1) has rank one. Our extension to a more general low-rank signal matrix requires controlling a non-trivial interaction between the eigenspaces corresponding to the non-zero signal eigenvalues of $\mathbf{P}$. We first obtain a local law for a low-rank expectation matrix whose spectral norm is not bounded by

a constant after scaling, which is possible due to our assumption on $\|\mathbf{U}_{1:r}\|_{2,\infty}$. Eigenvector delocalization is then a corollary of the local law. There are similar works that prove a local law for an expectation matrix whose spectral norm is either bounded by a constant or assumed to be diagonal but with moment conditions different from our setting (Erdős et al., 2025; Lee et al., 2016). Capitaine et al. (2009) investigates the distribution of the largest eigenvalue for a low-rank-plus-noise model, but in the setting where the spectral norm of $\mathbf{P}$ is bounded by a constant. Our proof requires the matrix of entry-wise variances of $\mathbf{E}$ to be doubly stochastic, which binary networks do not satisfy, in general. However, we suspect that delocalization still occurs even when this assumption is relaxed (see Section 3.1 for more discussion of this point). We also expect that Assumption (A5) can be relaxed to allow $r$ to grow as fast as $r \ll n^{1/4}$, possibly at the expense of a small polynomial factor in the delocalization bound.

**Remark 3.2.** Results from random matrix theory concerning the distribution of the eigenvalues have found use in network model selection (see, e.g., Lei, 2016; Bickel & Sarkar, 2015). We emphasize that we are concerned with the behavior of the ASE when it is (potentially) misspecified, which occurs downstream of selecting an embedding dimension and requires understanding the behavior of the eigenvectors associated with the perturbed null space, rather than the behaviors of the eigenvalues alone.

Most important for our work, Theorem 3.1 allows us to characterize the eigenvectors associated with the perturbed null space. In particular, we show that for some $c > 4$,

$$\left\|\hat{\mathbf{U}}_{r+1:d}\right\|_{2,\infty} \lesssim \frac{\sqrt{d-r}(\log n)^c}{n^{1/2}}.$$

We highlight the use of this bound in our main contribution: quantifying the behavior of the ASE when its embedding dimension is misspecified. A proof is given in Appendix B.

**Theorem 3.3.** *Under Assumptions* (A1) − (A7)*, suppose* $(\mathbf{A}, \mathbf{X}) \sim \mathrm{RDPG}(F, n)$ *is an $r$-dimensional weighted RDPG with signal strength parameter $\rho_n$. Denote the $d$-dimensional adjacency spectral embedding of $\mathbf{A}$ as $\hat{\mathbf{X}}_{1:d} = \hat{\mathbf{U}}_{1:d}|\hat{\mathbf{S}}|_{1:d}^{1/2}$. Then, for $d < r$,*

$$\min_{\mathbf{W} \in \mathbb{O}_r} \left\|\hat{\mathbf{X}}_{1:r}^\circ \mathbf{W} - \sqrt{\rho_n} \mathbf{X}_{1:r}\right\|_{2,\infty} \gtrsim \sqrt{|d-r|\rho_n}.$$

*On the other hand, for $d \geq r$, assume the existence of a sequence of $\mathbf{W}^* \in \mathbb{O}_r$ and $(\phi_n)_{n=1}^\infty$ as in Equation* (9)*. Then there exists a sequence of $\mathbf{W} \in \mathbb{O}_d$ such that*

$$\left\|\hat{\mathbf{X}}_{1:d} - \sqrt{\rho_n} \mathbf{X}_{1:d} \mathbf{W}\right\|_{2,\infty}$$
$$\lesssim \phi_n + \frac{\sqrt{\sigma^2(d-r)}r^2(\log n)^{5+6\gamma}}{n^{1/4}}. \quad (11)$$

Theorem 3.3 characterizes the behavior of the ASE under misspecification of the embedding dimension. When the embedding dimension is chosen too small, there is a fundamental limit to how well one can recover the true latent positions $\mathbf{X}_{1:r}$, and that this limit is driven by the sparsity $\rho_n$. In the extreme case of dense networks, where $\rho_n = \Theta(1)$, we see that under-estimating the true model rank $r$ results in inconsistent estimation. On the other hand, when the embedding dimension is chosen too large, $d > r$, we see that consistent estimation still holds, albeit at a slower rate than the $n^{-1/2}$ rate more typically seen in the literature, as summarized after Lemma 2.4, which holds only when $d = r$. In short, Theorem 3.3 suggests that when performing model selection for network embeddings, it is better to err on the side of choosing the embedding dimension too large, rather than too small. This is in line with existing folklore in statistical network analysis.

### 3.1. Conjecture: Consistency for Binary Networks

Theorem 3.3 shows consistency in $(2, \infty)$-norm for weighted networks when the embedding dimension is chosen at least as large as $r$. The proof relies on showing that the eigenvectors of the observed matrix $\mathbf{A}$ are delocalized (see Theorem 3.1). Our proof of Theorem 3.1 requires the matrix of entry-wise variances of $\mathbf{E}$ to be doubly stochastic, which binary networks do not always satisfy due to the heterogeneous entries in $\mathbf{P}$. These entries determine the variance of the entries of $\mathbf{E}$, since the entries of $\mathbf{A}$ are Bernoulli distributed. However, recent developments in random matrix theory (Ajanki et al., 2015; 2017) have relaxed this doubly stochastic assumption. Based on these recent theoretical advancements, along with the experimental results for binary networks shown in Section 4.1, we suspect that we can also relax the doubly stochastic assumption. This would allow us to extend the delocalization result in Theorem 3.1 to a much broader class of networks, including those with sparse binary edges. In particular, we make the following conjecture.

**Conjecture 1.** Suppose that $\mathbf{A} = \mathbf{P} + \mathbf{E}$, and Assumptions (A1) − (A7) hold, with Assumption (A7) relaxed to the condition that $c \leq \mathbb{E}\left[E_{ij}^2\right] \leq C$. Then for all $j \in [n]$,

$$\max_{\alpha > r} |\hat{u}_{j\alpha}| \lesssim \frac{r^2(\log n)^{4+6\gamma}}{\sqrt{n}}$$

Conjecture 1 would show delocalization for a class of matrices that, to our knowledge, there are currently no results for. Current results, such as those in Ajanki et al. (2017) concern Wigner-type matrices which have moment conditions that sparse binary matrices do not satisfy in general. An immediate corollary to Conjecture 1, if it can be proven true, would be consistent estimation of the latent positions under the binary RDPG when the embedding dimension is chosen too large ($d > r$), provided that the sparsity parameter $\rho_n$ does not converge to zero too fast.

# 4. Experiments

We now present an experimental investigation of our theoretical results presented in Section 3. We first consider the behavior of the ASE in the weighted RDPG setting of Theorem 3.3, in which the signal strength parameter $\rho_n$ is held constant. To obtain $\mathbf{P} = \rho_n \mathbf{X}\mathbf{X}^\top$, we generate the rows of $\mathbf{X} \in \mathbb{R}^{n \times r}$ i.i.d. from a Dirichlet distribution with parameter $\alpha = (1,1,1,1,1)$, so that the true latent dimension is $r = 5$. Given these latent positions, a network is specified by generating an adjacency matrix $\mathbf{A}$ as

$$\mathbf{A} = \mathbf{X}\mathbf{X}^\top + \mathbf{E}, \qquad (12)$$

where $\mathbf{E}$ has entries $E_{ij} = E_{ji}$ for $i > j$ drawn i.i.d. from a mean-zero distribution. For the entries of $\mathbf{E}$, we consider

(a) $E_{ij} \overset{\text{i.i.d.}}{\sim} \mathcal{N}(0,1)$   and   (b) $E_{ij} + 1 \overset{\text{i.i.d.}}{\sim} \text{Exp}(1)$.

In setting (b), we center the distribution about 1 to ensure that the entries of $\mathbf{E}$ are mean zero as required by our theory. Given $\mathbf{A}$, we construct its $d$-dimensional ASE. Our goal is to examine how well the ASE recovers the latent positions $\mathbf{X}$, but we must account for the orthogonal non-identifiability of the model, as illustrated in Equation (4). To do this, we compute the singular value decomposition of

$$\hat{\mathbf{X}}_{1:d}^\top \mathbf{X}_{1:d} = \mathbf{U}\boldsymbol{\Sigma}\mathbf{V}^\top,$$

and set $\mathbf{Q} = \mathbf{U}\mathbf{V}^\top$, which solves the Procrustes problem

$$\mathbf{Q} = \mathbf{U}\mathbf{V}^\top = \operatorname*{argmin}_{\mathbf{W} \in \mathbb{O}_d} \left\| \hat{\mathbf{X}}_{1:d}\mathbf{W} - \mathbf{X}_{1:d} \right\|_F . \qquad (13)$$

We use this orthogonal matrix to compute

$$\left\| \hat{\mathbf{X}}_{1:d}\mathbf{Q} - \mathbf{X}_{1:d} \right\|_{2,\infty},$$

the error between our estimate $\hat{\mathbf{X}}_{1:d}$ and the true positions.

We conducted the above experiment for networks of $n = 300, 600, \ldots, 7800$ nodes and embedding dimensions $d = 4, 5, 6, 10, 20, 40$, with 80 Monte Carlo trials per condition. Figure 1 shows the results of this experiment.

Examining Figure 1, consistency appears to hold for embedding dimensions that are at least as large as the true embedding dimension ($r = 5$). When the embedding dimension is chosen too small (in this case, $d = 4$, indicated by the orange line), consistency no longer holds: the error between the ASE and the true latent positions appears to approach a constant for suitably large $n$ under both error models. With the value of $\rho_n = 1$ in these experiments, we see that the lower bound provided in Theorem 3.3, is concordant with the inconsistency observed here when the embedding dimension is chosen too small. Examining the correctly-chosen embedding dimension ($d = 5$, gold line) we see that as $n$ grows, the estimation error appears to match the predicted $n^{-1/2}$ rate predicted by Theorem 3.3.

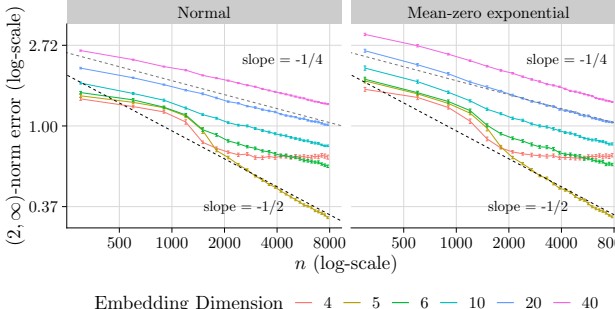

*Figure 1.* ASE estimation error in $(2,\infty)$-norm as a function of number of nodes $n$ for weighted networks as in Equation (12) with normal (left) and exponential (right) edges, under six choices of embedding dimension (colored lines). The correctly-chosen embedding dimension ($d = r = 5$; in gold) is shown alongside five misspecified embedding dimensions ($d = 4, 6, 10, 20, 40$). The black and gray dashed lines indicate the convergence rate when the embedding dimension is chosen correctly and too large, respectively. Error bars represent two standard errors of the mean.

When the dimension is chosen too large, corresponding to $d = 6, 10, 20, 40$ in the figure, we see that consistency still appears to hold, albeit at the slower $n^{-1/4}$ rate, again in line with Theorem 3.3. The similarity in the convergence rate for when the embedding dimension is chosen slightly larger ($d = 6$, green) than the true embedding dimension compared to when it is much larger ($d = 40$, magenta) shows that the ASE is comparatively insensitive to the embedding dimension once it is chosen too large. Additional experiments with different noise distributions, presented in Appendix A, exhibit similar behavior to that seen here.

For fixed $n$, Theorem 3.3 predicts that estimation error under the $(2,\infty)$-norm should exhibit square root behavior as the number of embedding dimensions increases past the true embedding dimension (i.e., $d > r$). We explore this experimentally under the setting of (b) described in the previous experiment. We computed the ASE for embedding dimensions of $d = 1, 2, \ldots, 10, 11$ and $d = 20, 40, 45$. Figure 2 appears to indicate that, for fixed $n$, our error in estimating the latent positions grows slightly slower than the $\sqrt{d-r}$ rate predicted by Theorem (3.3). We may explain this discrepancy by noting that when $d > r$ the alignment of $\hat{\mathbf{X}}_{1:d}$ to $\mathbf{X}_{1:d}$ involves aligning the trailing $d-r$ columns of $\hat{\mathbf{X}}_{1:d}$ to the $d-r$ columns of zeros that were padded to the end of $\mathbf{X}_{1:r}$ to form $\mathbf{X}_{1:d}$. Since the rows of $\mathbf{X}_{1:d}$ are in a subspace of $\mathbb{R}^d$, it is relatively easy to use the extra noisy $d-r$ dimensions of $\hat{\mathbf{X}}_{1:d}$ to "overfit" the alignment of the embeddings in $\hat{\mathbf{X}}_{1:d}$ to the latent positions in $\mathbf{X}_{1:d}$. This is also evident from the fact that no similar phenomenon appears in the other figures, where the embedding dimension is not growing. We also observe that the $(2,\infty)$-norm is smallest (for $n > 1500$) when the embedding dimension of the ASE and the rank ($r = 5$) are the same. We remark that the $(2,\infty)$-norm is not minimized at $d = r = 5$ for $n \le 1500$, likely due to the asymptotic nature of our results.

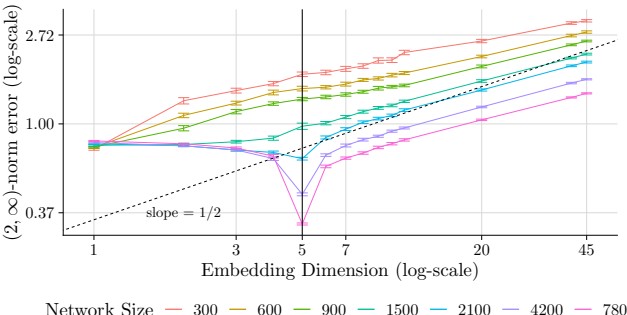

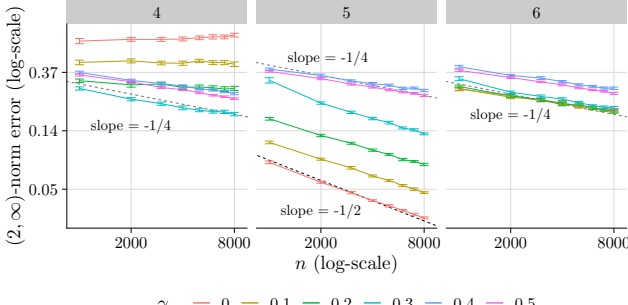

*Figure 2.* ASE estimation error in $(2, \infty)$-norm, as a function of embedding dimension for varying network sizes $n$, shown by line color. Both axes are on a logarithmic scale. The dashed black line indicates the $\sqrt{d - r}$-like behavior predicted by Theorem 3.3 The black vertical line indicates the true embedding dimension, $r = 5$. Error bars indicate two standard errors of the mean.

*Figure 3.* ASE estimation error in $(2, \infty)$-norm as a function of network size for varying signal strength (line color) for three choices of embedding dimension. Subplots show behavior under correctly-specified embedding dimension ($d = r = 5$; middle), as well as when the dimension is chosen too low ($d = 4$; left) and too high ($d = 6$; right). The black and gray dashed lines indicate the convergence rate predicted by Theorem 3.3 for $d = r$ and $d > r$, respectively. Error bars represent two standard errors of the mean.

To examine the effect of the signal strength $\rho_n$ on the behavior of the ASE, we again sampled the rows of $\mathbf{X}$ i.i.d. from $\text{Dir}(\alpha)$, where $\alpha = (1, 1, 1, 1, 1)^\top$, so that the true latent dimension is $r = 5$. The network is then generated as

$$\mathbf{A} = \rho_n \mathbf{X}\mathbf{X}^\top + \mathbf{E},$$

where $E_{ij} \overset{\text{i.i.d.}}{\sim} \mathcal{N}(0, 0.1^2)$ for $i > j$ and $\rho_n = n^{-\gamma}$, $\gamma \geq 0$, is a signal strength parameter that will shrink to zero with $n$ (see Remark 2.3). Given $\mathbf{A}$, we constructed its $d$-dimensional ASE for varying choices of $d$ and computed its (approximate) estimation error in $(2, \infty)$-norm via the orthogonal Procrustes problem in Equation (13). We conducted this experiment for network sizes $n = 1000, 2000, \ldots, 8000$, sparsity exponent $\gamma = 0, 0.1, \ldots, 0.5$, and embedding dimensions $d = 4, 5, 6$, with 40 Monte Carlo replicates per condition. The results of this experiment are given in Figure 3. Additional experiments for $d = 7, 10, 20$ are given in Appendix A.1.

From Figure 3, we see that when the embedding dimension is too small (left), the estimation error decreases at a rate at most like $\sqrt{\rho_n}$, as predicted by Theorem 3.3. Under the well-specified case of $d = r = 5$ (middle), we see the estimation error decreases at rate of $\sqrt{n\rho_n}$ predicted by Theorem 6 from Levin et al. (2022). Finally, when $d$ is too large (right), the estimation error decreases at the $n^{-1/4}$ rate, regardless of $\rho_n$, as predicted by Theorem 3.3.

### 4.1. Binary Networks

We now investigate the behavior of the ASE applied to binary networks. As discussed in Section 3.1, our main results do not apply to binary networks, since the matrix of entry-wise variances of $\mathbf{A}$ is not, in general, doubly-stochastic, rendering Theorems 3.1 and 3.3 inapplicable. Nonetheless, we expect similar behavior to that under weighted networks. We consider the sparse binary RDPG here. Additional experiments under the SBM setting are in Appendix A.3.

To obtain our latent positions, we simulate from a Dirichlet distribution with the rows of $\mathbf{X} \in \mathbb{R}^{n \times r}$ drawn i.i.d. according to $\text{Dir}(\alpha)$ with $\alpha = (1, 1, 1, 1, 1)^\top$. The resulting adjacency matrix, $\mathbf{A}$, was then generated according to

$$\mathbb{P}(\mathbf{A} \mid \mathbf{X}) = \prod_{i < j} (\rho_n \mathbf{X}_i^\top \mathbf{X}_j)^{A_{ij}} (1 - \rho_n \mathbf{X}_i^\top \mathbf{X}_j)^{1 - A_{ij}}, \quad (14)$$

where $\rho_n = n^{-\gamma}$ for $\gamma \geq 0$. Recall that smaller values of $\rho_n$ usually imply more difficult estimation. Given $\mathbf{A}$, we calculate its ASE with embedding dimension $d = 4, 5, 6$ and determined the error in $(2, \infty)$-norm between these estimated latent positions and the true latent positions $\mathbf{X}$, after aligning them as described in Equation (13). We conducted this experiment with network sizes $n = 1000, 2000, \ldots, 15000$ and sparsity $\rho_n = n^{-\gamma}$ for $\gamma = 0, 0.1, 0.2, \ldots, 0.5$, with 40 Monte Carlo replicates per condition. Figure 4 summarizes the results of this experiment. Additional experiments with $d = 7, 10, 20$ are in Appendix A.5.

Figure 4 suggests that when the embedding dimension is correctly chosen (middle panel), the ASE converges in $(2, \infty)$-norm at approximately a $n^{-1/2}$ rate for all values of $\gamma$, in line with existing results (up to logarithmic factors) in the literature (Lyzinski et al., 2017; Rubin-Delanchy et al., 2022). When the embedding dimension is chosen too large (right panel), the ASE still converges for all values of $\gamma$, at a rate no slower than $n^{-1/4}$ (see Appendix A.5 for further discussion). When the embedding dimension is chosen too small (left panel), we see that the ASE is no longer consistent when the network is dense (i.e., $\gamma = 0$). When the network is sparse (i.e., $\gamma > 0$), the estimation error decays at a rate that is $\rho_n$-dependent, similar to the weighted network behavior predicted by Theorem 3.3 and seen in Figure 3.

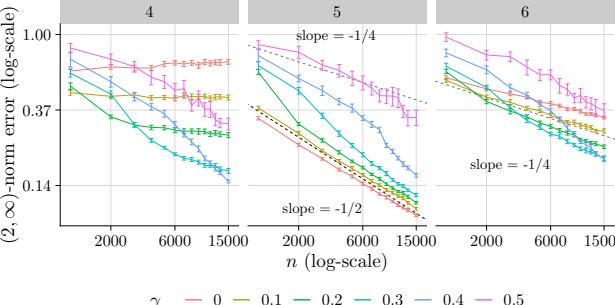

*Figure 4.* Estimation error of ASE in $(2, \infty)$-norm as a function of network size $n$ under the binary model in Equation (14) for sparsity $\rho_n = n^{-\gamma}$ (indicated by line color) for embedding dimensions of $d = 4, 5, 6$. The $n^{-1/2}$ rate suggested by our theory for the well-specified case ($d = r = 5$; middle) is shown by a black dashed line. The $n^{-1/4}$ rate suggested by our theory when the embedding dimension is too large ($d = 6$; right) is shown by a gray dashed line. Errors bars show two standard errors of the mean.

## 5. Discussion

We have considered how the choice of embedding dimension impacts estimation of the latent positions under signal-plus-noise matrix models $\mathbf{A} = \mathbf{P} + \mathbf{E}$ such as the RDPG. Our results characterize how different choices of embedding dimension affect the estimation rate of the ASE, measured in $(2, \infty)$-norm. Theorem 3.3 and our experiments in Section 4 show that under a weighted version of the RDPG, when the embedding dimension is chosen too small, consistency in the $(2, \infty)$-norm is not guaranteed for the ASE. When the embedding dimension is chosen too small, the estimation error is lower bounded by the signal strength parameter $\rho_n$, implying that the ASE is inconsistent when $\rho_n$ is non-vanishing and the embedding dimension is chosen too small. We may think of this as the embedding failing to capture all of the information contained in the network. When the embedding dimension is correctly specified, all signal present in the network is captured, with no trailing noise eigenvalues, and we achieve the $n^{-1/2}$ convergence rate (ignoring log-factors) seen in the literature and widely believed to be optimal (see, e.g., discussion in Yan & Levin, 2023; 2024). On the other hand, Theorem 3.3 and our experiments in Section 4 show that when the embedding dimension is chosen too large, consistency may still hold, so long as the embedding dimension is not too much larger than the true dimension. This result confirms the conjecture in Section 7 of Athreya et al. (2018): as long as all of the signal information is captured, the ASE should still be consistent. This consistency is a direct consequence of delocalization of the eigenvectors associated with the perturbed null space. Because of the stark differences in the asymptotic behaviors under these three regimes (embedding dimension either too small, chosen correctly, or too large), our results lend support to the folklore in network embeddings (and model selection more broadly) that it is better to err on the side of choosing model rank too large. An interesting question for future work concerns the estimation error upper bound rate of $n^{-1/4}$ established in Lemma 2.4 when the embedding dimension is chosen too large (i.e., $d > r$). We conjecture based on existing minimax results in the well-specified case (i.e., $d = r$; see Yan & Levin, 2023; 2024) that this rate is optimal subject to the constraint that $d > r$.

In practice, embeddings such as those produced by the ASE are not an end in themselves, but are typically used in downstream tasks such as clustering (see, e.g., Lyzinski et al., 2014; 2017) and hypothesis testing (Tang et al., 2017a;b). An avenue for future work is to examine the implications of the results established here for performance in these downstream tasks. As an example, we anticipate that under typical community detection settings, in which one applies the ASE to a network and clusters the embeddings using $k$-means, our lower-bound in Theorem 3.3 can be leveraged to prove a lower-bound showing that a non-vanishing fraction of vertices are misclassified when the embedding dimension is chosen too small. On the other hand, when the embedding dimension is chosen too large, we expect that clustering performance will depend more precisely on the structure of the underlying SBM edge probabilities. Precisely characterizing these downstream consequences of model misspecification would provide useful guidance for practitioners.

## Acknowledgements

We thank Joshua Cape, Karl Rohe, Hanbaek Lyu, Joshua Agterberg and Zachary Lubberts for their helpful conversations and insights during the development of this work. Support for this research was provided in part by NSF Award DMS-2023239 and by the University of Wisconsin–Madison Office of the Vice Chancellor for Research and Graduate Education with funding from the Wisconsin Alumni Research Foundation.

## Impact Statement

This paper addresses a theoretical question that arises frequently in network analysis and related fields. Its primary practical impact is in its implications for practitioners' approach to model selection in low-rank network models. While practitioners and theorists alike should remain cognizant of the potential negative societal impacts of their work, we do not anticipate any direct societal impacts from this work.

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

# A. Additional Experiments

Here we collect additional simulation experiments illustrating our main theoretical results.

## A.1. Weighted Networks

We again consider the behavior of the ASE in the weighted RDPG setting of Theorem 3.3, in which the signal strength parameter is held constant at $\rho_n = 1$. To obtain $\mathbf{P} = \rho_n \mathbf{X}\mathbf{X}^\top$, we generate the rows of $\mathbf{X} \in \mathbb{R}^{n \times r}$ i.i.d. from a Dirichlet distribution with parameter $\alpha = (1, 1, 1, 1, 1)$, so that the true latent dimension is $r = 5$. Given these latent positions, we generate our adjacency matrix $\mathbf{A}$ according to

$$\mathbf{A} = \mathbf{X}\mathbf{X}^\top + \mathbf{E}, \tag{15}$$

where $\mathbf{E}$ has entries $E_{ij} = E_{ji}$ for $i > j$ drawn i.i.d. with mean zero. We consider two different distributions for the entries of $\mathbf{E}$:

(a) $E_{ij} \overset{\text{i.i.d.}}{\sim} \text{Laplace}(0, 1)$;

(b) $E_{ij} + 1 \overset{\text{i.i.d.}}{\sim} \text{Pois}(1)$.

Given the adjacency matrix $\mathbf{A}$, we construct its $d$-dimensional ASE and compute its estimation error of the true latent positions $\mathbf{X}$ in $(2, \infty)$-norm as described by Equation (13).

We conducted the above experiment for varying networks of $n = 300, 600, \ldots, 7800$ vertices and varying embedding dimension $d = 4, 5, 6, 10, 20, 40$, with 80 Monte Carlo trials per condition. Figure 5 shows the results of this experiment, plotting estimation error of the ASE in $(2, \infty)$-norm as a function of network size $n$ for varying choices of embedding dimension. The axes are on a log-log scale to highlight the convergence rate predicted by Theorem 3.3 and demonstrate consistency of the ASE for when the embedding dimension is chosen larger than the true embedding dimension. The dashed black line indicates the $n^{-1/2}$ convergence rate (ignoring logarithmic factors) predicted by Theorem 3.3 when the dimension is correctly specified, while the dashed gray line indicates the $n^{-1/4}$ convergence rate (ignoring logarithmic factors) predicted to hold when the embedding dimension is chosen too large.

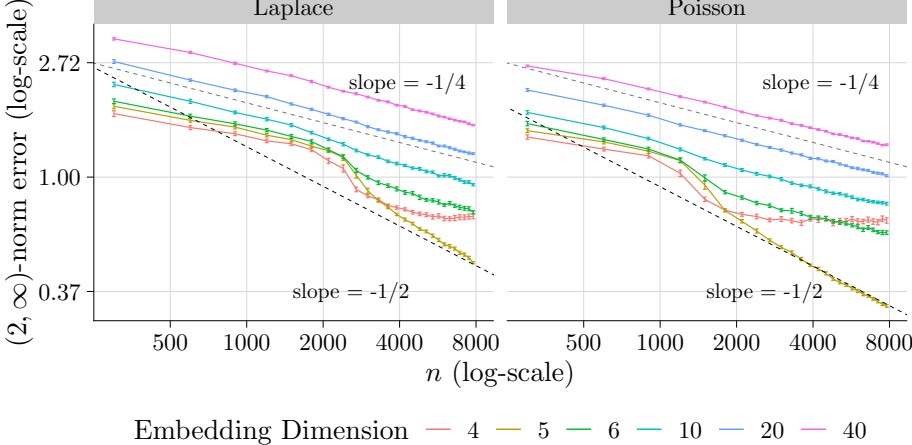

*Figure 5.* Estimation error of ASE in $(2, \infty)$-norm as a function of number of vertices $n$ for weighted networks as in Equation (15) under two different choices of noise models (left) Laplace and (right) Poisson, under six choices of embedding dimension (colored lines). The correctly-specified embedding dimension ($d = r = 5$; in gold) is displayed alongside five misspecified embedding dimensions ($d = 4, 6, 10, 20, 40$; orange, green, teal, purple and magenta, respectively). The black and gray dashed lines indicate, respectively, the convergence rate predicted for the correctly-specified setting and the setting where the embedding dimension is chosen too large. Error bars represent two standard errors of the mean.

Examining Figure 5, consistency appears to hold for embedding dimensions that are at least as large as the true embedding dimension ($r = 5$). When the embedding dimension is chosen too small ($d = 4$, indicated by the orange line), consistency no longer holds: the error between the ASE and the true latent positions appears to approach a constant for suitably large $n$ for both error models. Theorem 3.3 predicts that when $\rho_n$ is a constant, the ASE is inconsistent when the embedding

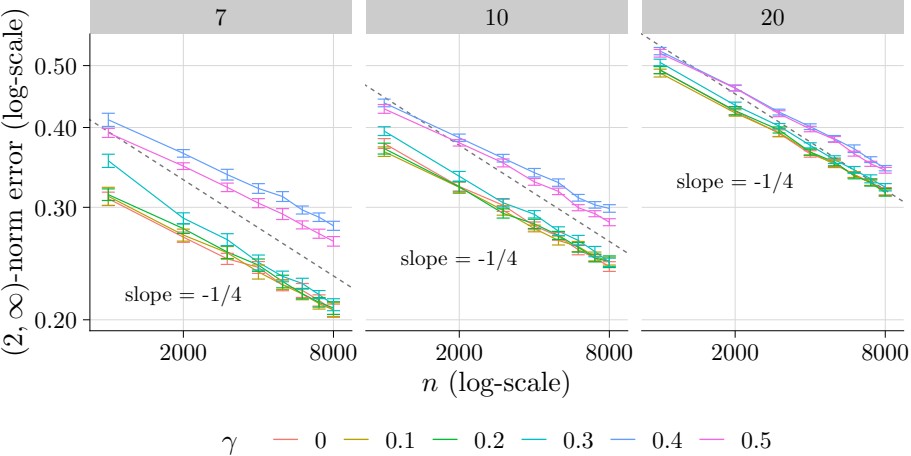

*Figure 6.* Estimation error of ASE in $(2, \infty)$-norm as a function of network size for varying signal strength levels (indicated by line color) for three choices of embedding dimension ($d = 7, 10, 20$). All three subplots show the behavior for when the embedding dimension is chosen too high. The gray dashed line with a slope of $-1/4$ indicates the predicted convergence rate of $n^{-1/4}$ from Theorem 3.3 for when the embedding dimension is chosen too high. Error bars represent two standard errors of the mean.

dimension is chosen too small, and indeed, when $d$ is chosen to small ($d = 4$ in Figure 5), the ASE is no longer consistent. Turning our attention to the case of well-specified dimension, indicated by the gold line in Figure 5, we see that as $n$ grows, the estimation error appears to match the predicted $n^{-1/2}$ rate predicted by Theorem 3.3. When the dimension is chosen too large, indicated by the green, blue, purple and magenta lines in the figure (corresponding embedding dimensions $d = 6, 10, 20$, and $40$, respectively), we see that consistency still appears to hold, albeit at the slower $n^{-1/4}$ rate, again in line with Theorem 3.3.

In our next set of experiments, we explore the behavior of the ASE when the embedding dimension is chosen much larger than the true dimension under a weak signal strength parameter $\rho_n$. We obtain $\mathbf{P} = \rho_n \mathbf{X}\mathbf{X}^\top$ and generate the rows of $\mathbf{X} \in \mathbb{R}^{n \times r}$ i.i.d. from $\mathrm{Dir}(\alpha)$, where $\alpha = (1, 1, 1, 1, 1)$. Then the true embedding dimension $r = 5$. Then, the adjacency matrix $\mathbf{A}$ is obtained as

$$\mathbf{A} = \rho_n \mathbf{X}\mathbf{X}^\top + \mathbf{E},$$

where $E_{ij} = E_{ji} \overset{\text{i.i.d.}}{\sim} N(0, 0.1^2)$ for $i > j$. From $\mathbf{A}$, we calculate its $d$-dimension ASE and compute its estimation error of the true latent positions $\mathbf{X}$ in $(2, \infty)$-norm, as described in Equation (13).

We performed the above experiment for network sizes of $n = 1000, 2000, \ldots, 8000$, embedding dimensions of $d = 7, 10, 20$, and signal strength parameter $\gamma = 0, 0.1, \ldots, 0.5$, with 40 Monte Carlo replicates per condition. The results of this experiment are collected in Figure 6. Examining the figure, we see that the $(2, \infty)$-norm error of the ASE behaves similarly across all of the considered embedding dimensions $d$ and signal strength parameters $\rho_n$, decreasing at approximately the $n^{-1/4}$ rate predicted by our theory. That is, once the embedding dimension of the ASE is chosen too large, the behavior of its estimation error in $(2, \infty)$-norm is largely the same, regardless of the amount of signal strength present in the observed network.

### A.2. Effect of Repeated Eigenvalues

We now consider the case where our low-rank expectation matrix $\mathbf{P}$ has repeated eigenvalues. Our theoretical results do not cover the case of repeated eigenvalues, but as discussed in Section 2, we suspect delocalization of the trailing eigenvectors still holds even when the signal matrix contains repeated eigenvalues. To investigate this experimentally, we generate $\mathbf{P} = \mathbf{U}_{1:r}\mathbf{S}_{1:r}\mathbf{U}_{1:r}^\top$, where $r = 10$, $S$ is a $r \times r$ diagonal matrix with all diagonal entries equal to $n/10$, and $\mathbf{U} \in \mathbb{R}^{n \times r}$ is an orthogonal matrix sampled from the Haar distribution on the Stiefel manifold. The adjacency matrix is then obtained as

$$\mathbf{A} = \mathbf{P} + \mathbf{E}, \tag{16}$$

with $E_{ji} = E_{ij} \overset{\text{i.i.d.}}{\sim} \mathcal{N}(0, 0.1^2)$. Having generated $\mathbf{A}$, we compute its $d$-dimensional ASE and then compute its $(2, \infty)$-norm estimation error in recovering the latent positions as described in Equation (13). We performed this experiment for network

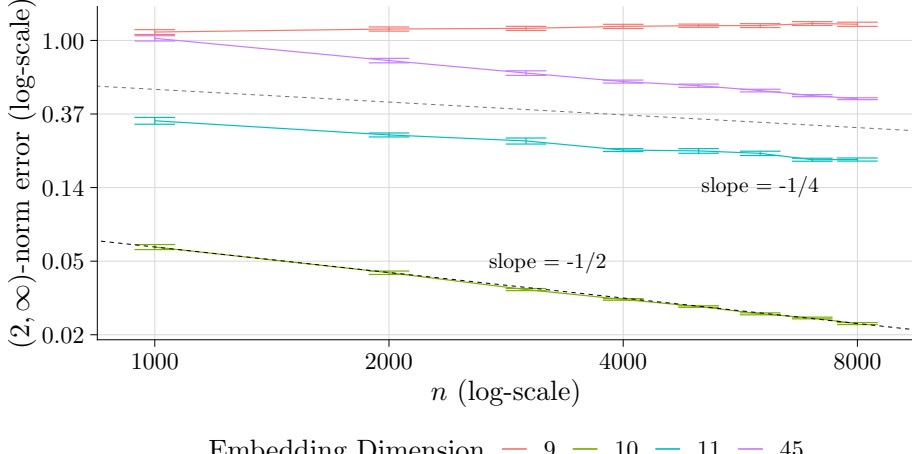

*Figure 7.* ASE estimation error in $(2, \infty)$-norm as a function of the number of nodes $n$ for four different choices of embedding dimension ($d$; line color) for the degree-corrected SBM described in Equation (16). Both axes are on a log-log scale to highlight convergence rate. The black and gray dashed lines indicate convergence rates of $n^{-1/2}$ and $n^{-1/4}$ respectively. Error bars represent two standard errors of the mean.

sizes of $n = 1000, 2000, \ldots, 8000$, and embedding dimensions of $d = 9, 10, 11, 45$ with 40 Monte Carlo replicates per condition. The results of this experiment are shown in Figure 7.

Examining Figure 7, we see that even with repeated eigenvalues, the ASE still behaves as our characterization in Theorem 3.3 would suggest: when the embedding dimension is chosen too small ($d = 9$; red line) relative to the true dimension ($r = 10$), the ASE is inconsistent. When the embedding dimension is chosen correctly ($d = r = 10$; green line), the ASE converges at the typical $n^{-1/2}$ rate (up to log factors). When the embedding dimension is chosen too large ($d = 11, 45$; teal and purple lines, respectively), the ASE is still consistent, although at a slower rate that agrees with the $n^{-1/4}$ rate predicted by our theory. These experiments suggest that the ASE is still consistent when the embedding dimension is chosen too large, even when the distinct eigenvalue assumption **A3** is relaxed.

### A.3. Binary Stochastic Blockmodel

The stochastic blockmodel (SBM) models network formation by assigning each vertex to one of $r$ communities, and generating edges conditionally independently given these community assignments, in such a way that the probability of two vertices forming an edge is determined by their community memberships. We store these probabilities in the entries of a symmetric matrix $\mathbf{B} \in [0, 1]^{r \times r}$, so that $B_{k, \ell}$ encodes the probability that $A_{ij} = 1$, given that vertex $i$ is in community $k$ and vertex $j$ is in community $\ell$. We generate networks from the SBM as follows:

1. Draw a random probability vector $\pi \sim \text{Dir}(\alpha)$, where $\alpha = (1, 1, \ldots, 1)^\top \in \mathbb{R}^r$ is a vector of all ones.

2. Assign each node $i \in [n]$ to one of $r$ communities independently according to $\pi$ and record these memberships as one-hot encodings in the rows of $\mathbf{Z} \in \mathbb{R}^{n \times r}$.

3. Construct $\mathbf{B} \in [0, 1]^{r \times r}$ by setting the within-community probabilities (i.e., on-diagonal entries) to $0.9$ and the inter-community edge probabilities (i.e., off-diagonal entries) to $0.1$.

4. Set $\mathbf{P} = \mathbf{Z}\mathbf{B}\mathbf{Z}^\top$ and obtain $\mathbf{X}$ from the eigendecomposition of $\mathbf{P} = \mathbf{U}\mathbf{S}\mathbf{U}^\top$ as $\mathbf{X} = \mathbf{U}_{1:r}\mathbf{S}_{1:r}^{1/2}$.

5. Generate $\mathbf{A} \in \{0, 1\}^{n \times n}$ according to

$$\mathbb{P}(\mathbf{A} \mid \mathbf{X}) = \prod_{i<j} P_{ij}^{A_{ij}}(1 - P_{ij})^{1 - A_{ij}}. \tag{17}$$

We generated $\mathbf{A}$ according to the procedure described above, with true number of communities (and hence true latent dimension) again set to $r = 5$. From $\mathbf{A}$, we then calculated the $d$-dimensional ASE with the embedding dimension

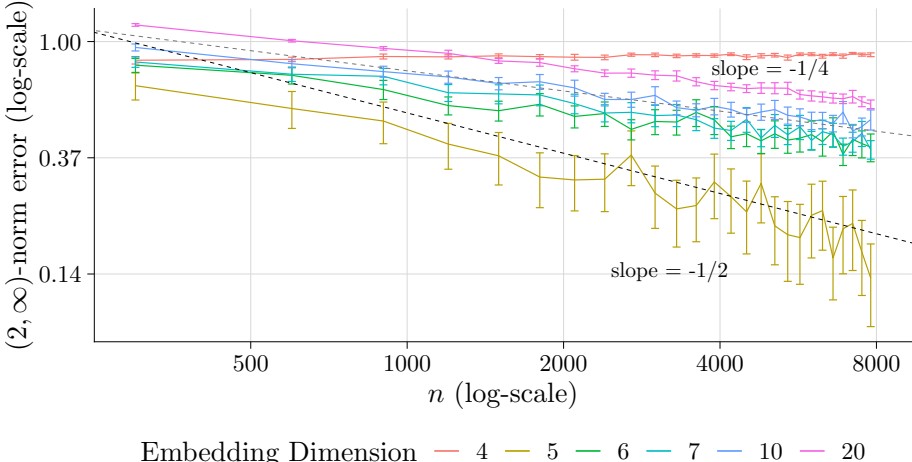

*Figure 8.* Average estimation error of ASE in $(2, \infty)$-norm as a function of number of vertices $n$ for different choices of embedding dimension $d$ (line colors) under the SBM described in Equation (17). Both axes are on logarithmic scales, with dashed lines in black and gray indicating, respectively, the $n^{-1/2}$ and $n^{-1/4}$ rates predicted by our Conjecture 1 under the settings where the dimension is chosen, respectively, either correctly or too large. Error bars indicate two standard errors of the mean.

chosen correctly (i.e., $d = r$) and incorrectly ($d = 4, 6, 7, 10, 20$), and computed the error in $(2, \infty)$-norm after solving the Procrustes alignment described in Equation (13). We repeated this process for network size $n = 300, 600, \ldots, 7800$, with 80 Monte Carlo replicates per condition. Figure 8 shows the results of this experiment. As in the previous figures, the axes are on a log-log scale to highlight the convergence rate and show consistency of the ASE when the embedding dimension is chosen suitably large. The black dashed line indicates our conjectured $n^{-1/2}$ convergence rate for the correctly-specified case. The gray dashed line indicates our conjectured $n^{-1/4}$ convergence rate for when the embedding dimension is chosen too large. Different choices of embedding dimension are indicated by different line colors, with the gold line indicating the correctly-specified dimension $d = r = 5$. Inspecting the figure, we see that the convergence behaviors are broadly similar to the weighted networks considered in Section 4 (see Figure 1), lending evidence in favor of our Conjecture 1 its implications for model selection. The predicted $n^{-1/2}$ convergence rate is again obtained when the embedding dimension is chosen correctly ($d = r = 5$, in gold), and consistency still appears to hold when the embedding dimension is chosen too large (i.e., $d > r$, in green, teal, purple and magenta), albeit at the slower $n^{-1/4}$ rate, as conjectured. When the dimension is chosen too small ($d < r$, in orange), we no longer see consistency, again in line with our theoretical results and conjectured behavior for binary networks. We remark that the proof of the lower bound for when the embedding dimension is chosen smaller than the true dimension in Theorem 3.3 is applicable to binary networks as well.

## A.4. Effect of Degree Heterogeneity

To investigate the behavior of the ASE when there is greater degree heterogeneity and heterogeneity in the variances of the edge noise $\mathbf{E}$, we turn to the degree-corrected stochastic blockmodel (DCSBM; Karrer & Newman, 2011). Like the SBM, the DCSBM models each node as belonging to one of $r$ communities. To model degree heterogeneity, the DCSBM includes an additional degree correction factor $\theta_i \geq 0$ for every node $i \in [n]$ such that the probability of forming an edge between vertices $i$ and $j$ is given by $P_{ij} = \theta_i \theta_j B_{c_i, c_j}$, where $c_i$ is the community of node $x$. More formally, we generate a network from the DCSBM as follows:

1. Assign each node $i \in [n]$ to one of $r$ communities independent with probability $1/r$ and record these memberships as one-hot encodings in the rows of $\mathbf{Z} \in \mathbb{R}^{n \times r}$.

2. Construct $\mathbf{B} \in [0, 1]^{r \times r}$ with the within-community probabilities $B_{kk}$ set to $0.6$ and the inter-community probabilities (i.e., off-diagonal entries of $\mathbf{B}$) set to $0.18$.

3. Construct the diagonal degree-correction matrix $\mathbf{\Theta} = \mathrm{diag}(\theta_1, \theta_2, \ldots, \theta_n) \in \mathbb{R}^{n \times n}$ by sampling $\theta_1, \theta_2, \ldots, \theta_n$ i.i.d. from a Pareto distribution with shape parameter $\alpha = 5$ and mode parameter $m = 0.8$.

4. Set $\mathbf{P} = \mathbf{\Theta Z B Z^\top \Theta^\top}$ with the entries hollowed out to prevent self-loops and obtain its eigendecomposition as

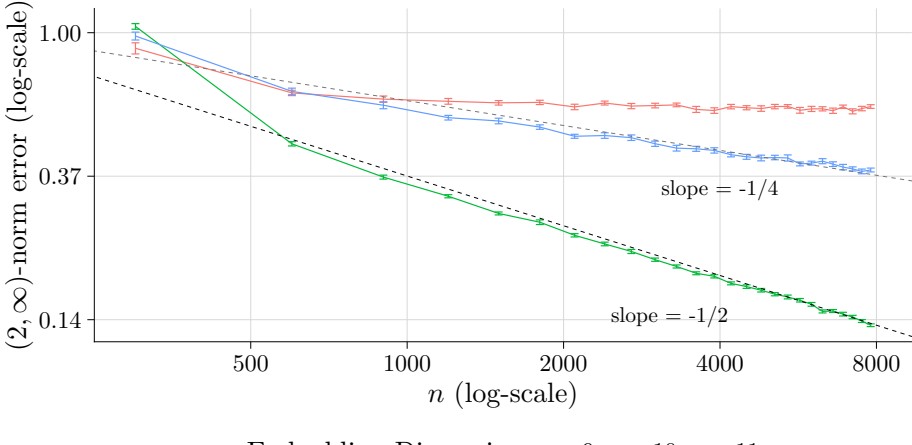

*Figure 9.* ASE estimation error in $(2, \infty)$-norm as a function of the number of nodes $n$ for three different choices of embedding dimension ($d$; line color) for the degree-corrected SBM described in Equation (18). The black and gray dashed lines indicate convergence rates of $n^{-1/2}$ and $n^{-1/4}$ respectively. Error bars represent two standard errors of the mean.

$\mathbf{P} = \mathbf{U}\mathbf{S}\mathbf{U}^{\top}$. The matrix of latent positions is then constructed as $\mathbf{X} = \mathbf{U}_{1:r}\mathbf{S}_{1:r}^{1/2}$.

5. Generate $\mathbf{A} \in \{0, 1\}^{n \times n}$ according to

$$\mathbb{P}(\mathbf{A} \mid \mathbf{X}) = \prod_{i<j} P_{ij}^{A_{ij}}(1 - P_{ij})^{1-A_{ij}}. \tag{18}$$

We generated $\mathbf{A}$ as above with the number of communities set to $r = 10$. From $\mathbf{A}$, we then constructed the $d$-dimensional ASE with the embedding dimension chosen correctly ($d = r = 10$) and incorrectly ($d = 9, 11$). We then computed the ASE's $(2, \infty)$-norm estimation error in recovering $\mathbf{X} = \mathbf{\Theta}\mathbf{Z}\mathbf{B}^{1/2}$, after solving the Procrustes alignment as described in Equation (13). We repeated the above process for network sizes of $n = 300, 600, \ldots, 7800$ with 80 Monte Carlo replicates per condition. We collect the results of this experiment in Figure 9, with the axes on a log-log scale to highlight convergence rates.

Examining Figure 9, we see that the behavior of the ASE under the DCSBM resembles that of the SBM shown in Figure 8. When the embedding dimension is chosen too small ($d = 9$; red line) the ASE is no longer consistent in estimating the latent positions. When the embedding dimension is chosen correctly ($d = r = 10$; green line), the ASE converges at a $n^{-1/2}$ rate, and when the ASE is chosen too large ($d = 11$, blue line), the ASE converges at a slower $n^{-1/4}$ rate. The convergence rates in Figure 9 mirror the rates seen in the SBM setting and are concordant with our conjectured characterization of the ASE in the case of binary networks described in Conjecture 1. Comparing Figures 8 and 9, one key difference jumps out: there is greater variability in estimation error when the embedding dimension is chosen correctly in the SBM setting (gold line in Figure 8) as compared to the DCSBM setting (green line in Figure 9). This is due to the community proportion sizes being drawn according to a uniform distribution on $[r]$ in the DCSBM experiments, while the community membership probability vector $\pi$ is drawn from a Dirichlet distribution for each trial in the SBM experiments of Section A.3.

### A.5. Sparse Binary RDPG

In our next set of experiments, we revisit the sparse binary RDPG setting explored in Section 4.1, this time investigating the estimation error of the ASE when the embedding dimension is much greater than the true embedding dimension, $r$. We again draw our latent positions $\mathbf{X}_1, \mathbf{X}_2, \ldots, \mathbf{X}_n$ i.i.d. according to $\mathrm{Dir}(\alpha)$, where $\alpha = (1, 1, 1, 1, 1)^{\top} \in \mathbb{R}^5$. The adjacency matrix, $\mathbf{A}$, is then generated according to

$$\mathbb{P}(\mathbf{A} \mid \mathbf{X}) = \prod_{i<j} (\rho_n \mathbf{X}_i^{\top} \mathbf{X}_j)^{A_{ij}}(1 - \rho_n \mathbf{X}_i^{\top} \mathbf{X}_j)^{1-A_{ij}}, \tag{19}$$

where, as in Section 4.1, $\rho_n = n^{-\gamma}$ for $\gamma \geq 0$.

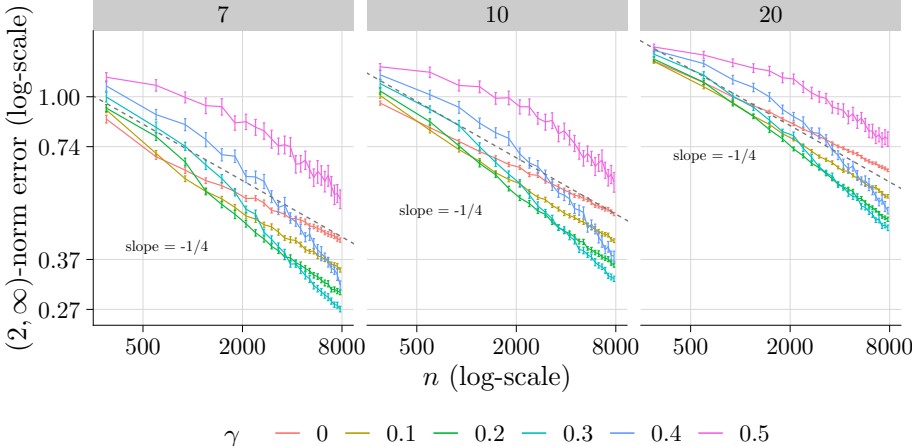

*Figure 10.* Estimation error of ASE in $(2, \infty)$-norm as a function of network size $n$ under the sparse binary model in Equation (19) for varying sparsity $\rho_n = n^{-\gamma}$ ($\gamma = 0, 0.1, 0.2, 0.3, 0.4, 0.5$; indicated by line color) for varying choices of embedding dimension $d = 7, 10, 20$. The $n^{-1/4}$ rate suggested by our theory for when the embedding dimension is too large ($d > 5$; all panels) is indicated by a gray dashed line. Errors bars indicate two standard errors of the mean.

From $\mathbf{A}$, we calculated its $d$-dimensional ASE for $d = 7, 10, 20$ and determined its $(2, \infty)$-norm error in recovering the true latent positions $\mathbf{X}$, after accounting for the orthogonal Procrustes problem as described in Equation (13). We conducted this experiment for network sizes $n = 300, 600, \dots, 7800$, and sparsity rates $\rho_n = n^{-\gamma}$ for $\gamma = 0, 0.1, \dots, 0.5$, with 80 Monte Carlo replicates per condition. We collect the results of this experiment in Figure 10.

From Figure 10, we see that the ASE converges for all choices of $d$ and $\gamma$ considered. The convergence rate, however, appears to depend only on the value of $\gamma$, with smaller values of $\gamma$ (less sparsity), converging near the $n^{-1/4}$ rate seen in SBM (Figure 8) and the weighted networks. Larger values of $\gamma$ (higher sparsity), e.g., $\gamma = 0.4$ (blue line), appear to converge faster than $n^{-1/4}$. We can explain this departure in behavior from that seen in weighted and dense, binary networks by recalling the the trailing dimensions of the over-specified ASE can be described as

$$\hat{\mathbf{X}}_{r+1:d} = \begin{bmatrix} \hat{u}_{r+1}\sqrt{|\hat{s}_{r+1}|} & \cdots & \hat{u}_d\sqrt{|\hat{s}_d|} \end{bmatrix}.$$

For binary networks, the first non-signal eigenvalue of $\mathbf{A}$, $\hat{s}_{r+1}$, grows at a $\rho_n$-dependent rate of $\sqrt{\rho_n n}$ (Lei & Rinaldo, 2015). Indeed, smaller values of $\rho_n$ here would also mean $\hat{s}_{r+1}$ grows slower. Because the true latent positions, $\mathbf{X}_{r+1:d}$, are trivial zero vectors here, smaller $|\hat{s}_{r+1}|$ (e.g., $\gamma = 0.4$; blue line in Figure 10) implies a smaller estimation error for the ASE compared to larger $|\hat{s}_{r+1}|$ (e.g., $\gamma = 0.1$). This is in contrast to weighted networks, where the growth rate of $\hat{s}_{r+1}$ is driven by the edge-level variance, which need not depend on $\rho_n$ (see Remark 2.3).

In our final set of experiments for binary networks, we investigate the effects of high levels of sparsity on the eigenstructure of the observed adjacency matrix $\mathbf{A}$. To do so, we sampled the rows of $\mathbf{X}$ i.i.d. according to $\mathrm{Dir}(\alpha)$ where $\alpha = (1, 1, 1, 1, 1)^\top$, so that the true embedding dimension is $r = 5$. Our observed adjacency matrix is then obtained according to

$$\mathbb{P}(\mathbf{A} \mid \mathbf{X}) = \prod_{i<j} \left( \rho_n \mathbf{X}_i^\top \mathbf{X}_j \right)^{A_{ij}} \left( 1 - \rho_n \mathbf{X}_i^\top \mathbf{X}_j \right)^{1 - A_{ij}},$$

where, $\rho_n = (\log n)^\gamma / n$. From $\mathbf{A}$, we obtained its eigenvector decomposition and calculated the max norm of its $j = 1, 2, 5, 6, 10, 20$ eigenvectors. We conducted the above experiment for network sizes of $n = 300, 600, \dots, 7800$ and sparsity levels of $\gamma = 0.5, 1, 2, 3, 4, 5$, with 80 Monte Carlo replicates per condition. We collect the results of this experiment in Figure 11.

Examining Figure 11, we see that at the highest levels of sparsity ($\gamma = 0.5, 1, 2$; top row), eigenvectors that are associated with the signal dimensions ($j = 1, 2, 5$; red, gold, and green lines respectively) and eigenvectors that are "trailing" ($j = 6, 10, 20$; teal, blue, and magenta lines respectively) are localized: their entries do not vanish as the network size increases. When $\gamma = 3$ (bottom row, first column), aside from the first leading eigenvector ($j = 1$; red line), the eigenvectors start to delocalize but slower than an $n^{-1/2}$ rate; the first leading eigenvector delocalizes at the $n^{-1/2}$ rate. When $\gamma > 3$ (bottom row, second and third column), all eigenvectors considered in this experiment delocalize at the $n^{-1/2}$ rate we would

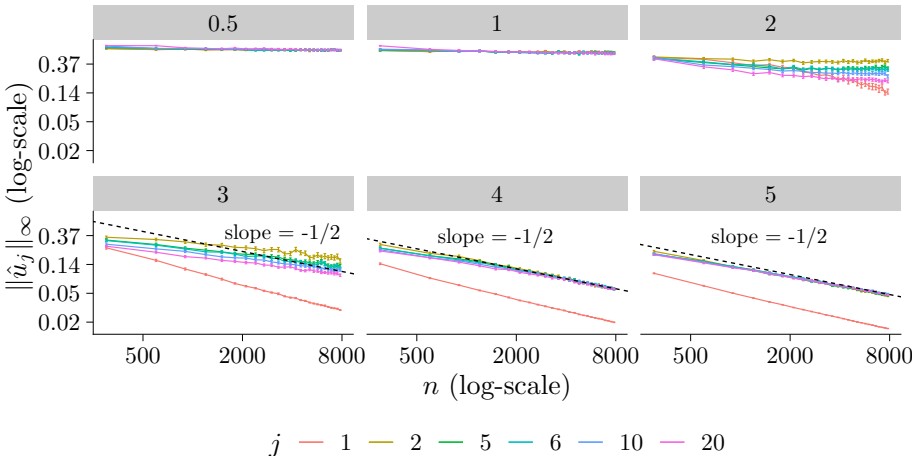

*Figure 11.* Max norm of the leading eigenvectors of a sparse, binary network $\mathbf{A}$ as a function of number of vertices $n$ for sparsity parameter $\rho_n = (\log n)^\gamma / n$. Subplots correspond to sparsity levels $\gamma = 0.5, 1, 2$ (top row) and $\gamma = 3, 4, 5$ (bottom row). We show the behavior of the $j$-th eigenvector, for $j = 1, 2, 5, 6, 10, 20$ (red, gold, green, teal, blue, and magenta lines, respectively). The black dashed line indicates an $n^{-1/2}$ convergence rate. Error bars represent two standard errors of the mean.

expect if Theorem 3.1 were generalized to the sparse binary setting (i.e., as predicted in our Conjecture 1). Based on the above experiment and the fact that $\gamma > 3$ is an assumption for the proof of delocalization given in (Erdős et al., 2013a), which covers the $r = 1$ case, we expect eigenvector delocalization to hold for sparsity levels of $\rho_n > (\log n)^3 / n$.

### A.6. Nonlinear Link Functions

As discussed in the main text after Definition 2.2, the RDPG can be generalized such that the edge probabilities are given by a kernel function applied to the latent positions. Our investigations thus far have only considered the case where this kernel function is the inner product. In our final experiment, we investigate the ability of the ASE to recover the underlying latent positions $\mathbf{X}$ when the kernel function includes a nonlinearity. As long as the kernel function is sufficiently smooth and induces an eigenstructure in $\mathbf{P}$ that is close to that of $\mathbf{X}\mathbf{X}^\top$, we expect the ASE to continue to behave similarly to the predictions of our main results. In fact, if the nonlinearity causes the rank of $\mathbf{P}$ to increase, so long as the additional non-zero eigenvalues decay sufficiently fast, we may regard these eigenvalues as noise and absorb them into the error term $\mathbf{E}$.

We investigate the impact of embedding dimension misspecification on the recovery of the latent positions $\mathbf{X}$ when the kernel function is no longer the inner product. To obtain our matrix of edge probabilities $\mathbf{P}$, we first generate latent positions $\mathbf{X}_1, \mathbf{X}_2, \ldots, \mathbf{X}_n$ by drawing i.i.d. from an $r$-dimensional multivariate normal distribution centered at the origin and with diagonal covariance matrix $0.01\mathbf{I}$. We then obtain the entries of $\mathbf{P}$ as $P_{ij} = \sigma\left(\mathbf{X}_i^\top \mathbf{X}_j\right)$ where

$$\sigma(y) = \frac{1}{1 + \exp\left(-y/h\right)} - 0.5, \quad h = n^{-1/8}.$$

Note that we include $0.5$ as a centering term to remove the bias introduced by the sigmoid function, and $h$ is a bandwidth/smoothing parameter to account for the increasing density of $\mathbf{X}_i^\top \mathbf{X}_j$ around the origin as the number of nodes $n$ in the network increases. The network is then generated according to

$$\mathbf{A} = \mathbf{P} + \mathbf{E}, \tag{20}$$

where $E_{ij} \overset{\text{i.i.d.}}{\sim} \mathcal{N}(0, 0.1^2)$ for $i > j$. From $\mathbf{A}$, we then constructed its $d$-dimensional ASE and calculated the estimation error in $(2, \infty)$-norm after solving the Procrustes problem described in Equation (13). We conducted the above experiment for network sizes $n = 1000, 2000, \ldots, 8000$ and embedding dimensions $d = 9, 10, 11$, with 40 Monte Carlo replicates per condition. We collect the results of this experiment in Figure 12.

From Figure 12, we see that the behavior of the (misspecified) ASE is similar to the behavior guaranteed by our theory when $\mathbf{P}$ has an inner product structure. The behavior of the $d$-dimensional ASE constructed from $\sigma\left(\mathbf{X}\mathbf{X}^\top\right)$ is similar to that constructed from $\mathbf{X}\mathbf{X}^\top$: when the embedding dimension is chosen too small ($d = 9$; red line) relative to the true embedding dimension ($r = 10$), the ASE is no longer consistent in estimating the latent positions. When the embedding dimension

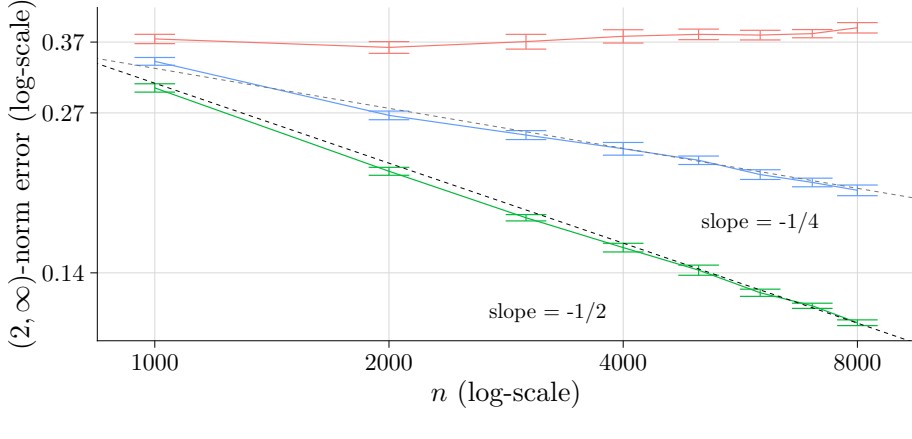

*Figure 12.* ASE estimation error in $(2, \infty)$-norm as a function of the number of nodes $n$ for three different choices of embedding dimension ($d$; line color) for weighted networks with nonlinear link functions, as in Equation (20). The black and gray dashed lines indicate convergence rates of $n^{-1/2}$ and $n^{-1/4}$ respectively. Error bars represent two standard errors of the mean.

is chosen correctly ($d = r = 10$; green line), the ASE follows the typical $n^{-1/2}$ convergence rate. When the embedding dimension is chosen too large ($d = 11$; blue line), the ASE follows an $n^{-1/4}$ convergence rate. We note that the choice of the bandwidth parameter $h = n^{-1/8}$ may not be optimal in these experiments, which impacts the estimation error in the case where the embedding dimension is correctly chosen ($d = r = 10$; green line). These experiments provide evidence that our theory may be extended to include suitably smooth nonlinear kernel functions on the latent space, a task we leave for future work.

## B. Bounds on Estimation Error

Here we provide proofs related to our RDPG estimation results, Lemma 2.4 and Theorem 3.3. We begin by establishing our basic upper- and lower-bounds on estimation error.

*Proof of Lemma 2.4.* We first consider the case where $d > r$. Define $\widetilde{\mathbf{W}} \in \mathbb{O}_d$ according to

$$\widetilde{\mathbf{W}} = \begin{bmatrix} \mathbf{W}^* & \mathbf{0} \\ \mathbf{0} & \mathbf{I}_{r+1:d} \end{bmatrix}, \tag{21}$$

where $\mathbf{W}^* \in \mathbb{O}_r$ is the matrix guaranteed by our assumption in Equation (9). Trivially,

$$\min_{\mathbf{W} \in \mathbb{O}_d} \left\| \hat{\mathbf{X}}_{1:d} \mathbf{W} - \rho_n^{1/2} \mathbf{X}_{1:d} \right\|_{2,\infty} \leq \left\| \hat{\mathbf{X}}_{1:d} \widetilde{\mathbf{W}} - \rho_n^{1/2} \mathbf{X}_{1:d} \right\|_{2,\infty}. \tag{22}$$

Now, express $\hat{\mathbf{X}}_{1:d}$ as the block matrix

$$\hat{\mathbf{X}}_{1:d} = \begin{bmatrix} \hat{\mathbf{X}}_{1:r} & \hat{\mathbf{X}}_{r+1:d} \end{bmatrix}.$$

Decomposing $\rho_n^{1/2} \mathbf{X}_{1:d}$ similarly, recalling from Equation (8) that the trailing $k$ columns of $\mathbf{X}$ are zero by construction, and using the definition of $\widetilde{\mathbf{W}}$ from Equation (21), we have

$$\hat{\mathbf{X}}_{1:d} \widetilde{\mathbf{W}} - \rho_n^{1/2} \mathbf{X}_{1:d} = \begin{bmatrix} \left( \hat{\mathbf{X}}_{1:r} \mathbf{W}^* - \rho_n^{1/2} \mathbf{X}_{1:r} \right) & \hat{\mathbf{X}}_{r+1:d} \end{bmatrix} \in \mathbb{R}^{n \times d}.$$

By the triangle inequality, it follows that

$$\left\| \hat{\mathbf{X}}_{1:d} \widetilde{\mathbf{W}} - \rho_n^{1/2} \mathbf{X}_{1:d} \right\|_{2,\infty} \leq \left\| \hat{\mathbf{X}}_{1:r} \mathbf{W}^* - \mathbf{X}_{1:r} \right\|_{2,\infty} + \left\| \hat{\mathbf{X}}_{r+1:d} \right\|_{2,\infty}.$$

By our choice of $\mathbf{W}^*$, Equation (9) allows us to bound the first right-hand term and write

$$\left\| \hat{\mathbf{X}}_{1:d} \widetilde{\mathbf{W}} - \rho_n^{1/2} \mathbf{X}_{1:d} \right\|_{2,\infty} \leq C\phi_n + \left\| \hat{\mathbf{X}}_{r+1:d} \right\|_{2,\infty}.$$

Applying this to Equation (22),

$$\min_{\mathbf{W} \in \mathbb{O}_d} \left\| \hat{\mathbf{X}}_{1:d}\mathbf{W} - \rho_n^{1/2}\mathbf{X}_{1:d} \right\|_{2,\infty} \leq C\phi_n + \left\| \hat{\mathbf{X}}_{r+1:d}\mathbf{W}^* \right\|_{2,\infty}. \tag{23}$$

Recall that by construction, $\hat{\mathbf{X}}_{r+1:d} = \hat{\mathbf{U}}_{r+1:d}|\hat{\mathbf{S}}_{r+1:d}|^{1/2}$. Thus, using basic properties of the $(2,\infty)$-norm (see, e.g., Cape et al., 2019b),

$$\left\| \hat{\mathbf{X}}_{r+1:d}\mathbf{W}^* \right\|_{2,\infty} \leq \left\| \hat{\mathbf{U}}_{r+1:d} \right\|_{2,\infty} \left\| \hat{\mathbf{S}}_{r+1:d}^{1/2}\mathbf{W}^* \right\| \leq \left\| \hat{\mathbf{U}}_{r+1:d} \right\|_{2,\infty} \|\mathbf{E}\|^{1/2}, \tag{24}$$

where the second inequality follows from unitary invariance of the spectral norm and Weyl's inequality. Substituting Equation (24) into Equation (23) completes the claim for the case of $d > r$.

For the lower-bound half of our claim, which holds when $d < r$, notice that by definition of the $(2,\infty)$-norm as a maximum, recalling the definition of $\hat{\mathbf{X}}_{1:r}^\circ$ from Equation (7), we have

$$\min_{\mathbf{W} \in \mathbb{O}_r} \left\| \hat{\mathbf{X}}_{1:r}^\circ\mathbf{W} - \rho_n^{1/2}\mathbf{X}_{1:r} \right\|_{2,\infty}^2 \geq \frac{1}{n} \min_{\mathbf{W} \in \mathbb{O}_r} \left\| \hat{\mathbf{X}}_{1:r}^\circ\mathbf{W} - \rho_n^{1/2}\mathbf{X}_{1:r} \right\|_F^2. \tag{25}$$

Expanding the Frobenius norm, for any $\mathbf{W} \in \mathbb{O}_r$,

$$\left\| \hat{\mathbf{X}}_{1:r}^\circ\mathbf{W} - \rho_n^{1/2}\mathbf{X}_{1:r} \right\|_F^2 = \left\| \hat{\mathbf{X}}_{1:r}^\circ\mathbf{W} \right\|_F^2 + \left\| \rho_n^{1/2}\mathbf{X}_{1:r} \right\|_F^2 - 2\operatorname{Tr}(\hat{\mathbf{X}}_{1:r}^\circ\mathbf{W})^\top \left( \rho_n^{1/2}\mathbf{X}_{1:r} \right).$$

Using unitary invariance of the Frobenius norm and noting that $\left\| \hat{\mathbf{X}}_{1:r}^\circ \right\|_F = \left\| \hat{\mathbf{X}}_{1:d} \right\|_F$ by construction,

$$\left\| \hat{\mathbf{X}}_{1:r}^\circ\mathbf{W} - \rho_n^{1/2}\mathbf{X}_{1:r} \right\|_F^2 = \left\| \hat{\mathbf{X}}_{1:d} \right\|_F^2 + \left\| \rho_n^{1/2}\mathbf{X}_{1:r} \right\|_F^2 - 2\operatorname{Tr}(\hat{\mathbf{X}}_{1:r}^\circ\mathbf{W})^\top \left( \rho_n^{1/2}\mathbf{X}_{1:r} \right). \tag{26}$$

Recalling that $\sqrt{\hat{s}_j}$ and $\sqrt{s_j}$ are the singular values of $\hat{\mathbf{X}}_{1:r}$ and $\rho_n^{1/2}\mathbf{X}_{1:r}$, respectively, the Von Neumann trace inequality (see, e.g., Horn & Johnson, 1985, Theorem 7.4.1.1) yields

$$\left| \operatorname{Tr}(\hat{\mathbf{X}}_{1:r}^\circ\mathbf{W})^\top(\rho_n^{1/2}\mathbf{X}_{1:r}) \right| \leq \sum_{j=1}^r \sqrt{s_j\hat{s}_j} = \sum_{j=1}^d \sqrt{s_j\hat{s}_j},$$

where we have used the fact that the singular values of $\hat{\mathbf{X}}_{1:r}^\circ$ are invariant to right-multiplication by $\mathbf{W}$ and the fact that $\hat{\mathbf{X}}_{1:d}$ has at most $d$ non-zero singular values by construction. Applying this bound in Equation (26),

$$\left\| \hat{\mathbf{X}}_{1:r}^\circ\mathbf{W} - \rho_n^{1/2}\mathbf{X}_{1:r} \right\|_F^2 \geq \sum_{j=1}^d \hat{s}_j + \sum_{j=1}^r s_j - 2\sum_{j=1}^d \sqrt{s_j\hat{s}_j}$$

$$= \sum_{j=1}^d \left( \sqrt{\hat{s}_j} - \sqrt{s_j} \right)^2 + \sum_{j=d+1}^r s_j.$$

Trivially lower-bounding the first sum by zero,

$$\left\| \hat{\mathbf{X}}_{1:r}^\circ\mathbf{W} - \rho_n^{1/2}\mathbf{X}_{1:r} \right\|_F^2 \geq \sum_{j=d+1}^r s_j.$$

Applying this lower-bound to Equation (25) and taking square roots completes the proof. □

We now turn to a proof of our general result on asymptotic behavior of the ASE, using Lemma 2.4.

*Proof of Theorem 3.3.* We first consider the case of $d > r$. Applying Lemma 2.4, there exists a sequence of $\mathbf{W} \in \mathbb{O}_d$ such that

$$\left\| \hat{\mathbf{X}}_{1:d}\mathbf{W} - \rho_n^{1/2}\mathbf{X}_{1:d} \right\|_{2,\infty} \leq C\phi_n + |\hat{s}_{r+1}|^{1/2} \left\| \hat{\mathbf{U}}_{r+1:d} \right\|_{2,\infty}. \tag{27}$$

To bound $|\hat{s}_{r+1}|$, standard matrix concentration inequalities (see, e.g., Vershynin, 2018) yield $\|\mathbf{E}\| \leq C\sigma\sqrt{n}\log n$ (see also Lemma 4.3 in Erdős et al., 2013a). Combining this with Weyl's inequality,

$$|\hat{s}_{r+1}|^{1/2} \left\|\hat{\mathbf{U}}_{r+1:d}\right\|_{2,\infty} \leq Cn^{1/4}(\sigma\log n)\max_{j\in[n]}\left(\sum_{\alpha=r+1}^{d}\hat{u}_{j\alpha}^2\right)^{1/2} \leq Cn^{1/4}(\sigma\log n)\left((d-r)\max_{j\in[n]}\max_{\alpha>r}\hat{u}_{j\alpha}^2\right)^{1/2}.$$

Then by Theorem 3.1, we have

$$|\hat{s}_{r+1}|^{1/2}\left\|\hat{\mathbf{U}}_{r+1:d}\right\|_{2,\infty} \leq Cn^{1/4}(\log n)\sqrt{(d-r)\sigma^2}\left(\frac{r^4(\log n)^{8+12\gamma}}{n}\right)^{1/2} \leq C\sqrt{(d-r)\sigma^2}\frac{r^2(\log n)^{5+6\gamma}}{n^{1/4}} \tag{28}$$

with high probability. Substituting Equation (28) back into Equation (27), we obtain

$$\min_{\mathbf{W}\in\mathbb{O}_d}\left\|\hat{\mathbf{X}}_{1:d}\mathbf{W} - \rho_n^{1/2}\mathbf{X}_{1:d}\right\|_{2,\infty} \leq C\phi_n + C\sqrt{(d-r)\sigma^2}\frac{r^2(\log n)^{5+6\gamma}}{n^{1/4}}$$

with high probability, which establishes our estimation error upper-bound.

For the the case $d < r$, by Lemma 2.4,

$$\min_{\mathbf{W}\in\mathbb{O}_r}\left\|\hat{\mathbf{X}}_{1:r}^{\circ}\mathbf{W} - \rho_n^{1/2}\mathbf{X}_{1:r}\right\|_{2,\infty} \geq \sqrt{\sum_{j=d+1}^{r}\frac{s_j}{n}}.$$

By Assumption A4, it follows that

$$\min_{\mathbf{W}\in\mathbb{O}_r}\left\|\hat{\mathbf{X}}_{1:r}^{\circ}\mathbf{W} - \rho_n^{1/2}\mathbf{X}_{1:r}\right\|_{2,\infty} \geq C\sqrt{\rho_n|d-r|},$$

as desired. $\qquad\square$

## C. Basic Results from Spectral Methods

Here we collect a handful of basic results related to matrix spectra and eigenspaces for use in Appendix D.

**Lemma C.1.** *Let $\{s_i\}$ be the eigenvalues of a rank $r$ matrix $\mathbf{P}$, and let $\{\hat{s}_i\}$ be the eigenvalues of $\mathbf{A} = \mathbf{P} + \mathbf{E}$. Suppose Assumption (A3) holds and define $\Delta = \min_{j\in[r]}\min_{i\neq j}|s_i - s_j|$. Then for $j \in [r]$ and $i \neq j$,*

$$|\hat{s}_i - s_j| \geq \Delta - \|\mathbf{E}\|. \tag{29}$$

*Proof.* Fix $j \in [r]$ and let $i \neq j$. By the reverse triangle inequality,

$$|\hat{s}_i - s_j| \geq ||\hat{s}_i - s_i| - |s_j - s_i|| \geq \Delta - |\hat{s}_i - s_i|,$$

where the second inequality follows from the definition of $\Delta$. Applying Weyl's inequality, it follows that

$$|\hat{s}_i - s_j| \geq \Delta - \|\mathbf{E}\|,$$

completing the proof. $\qquad\square$

**Lemma C.2.** *Suppose that $\mathbf{X}_1, \mathbf{X}_2, \ldots, \mathbf{X}_n$ are identically distributed $r$-dimensional random vectors with $\|\mathbf{X}_1\| \leq M < \infty$ almost surely and define $\mathbf{X} = [\mathbf{X}_1 \cdots \mathbf{X}_n]^{\top} \in \mathbb{R}^{n\times r}$. Suppose that $r$ obeys Assumption (A5). Then with high probability, $\left\|\mathbf{X}\mathbf{X}^{\top} - n\mathbb{E}[\mathbf{X}_1\mathbf{X}_1])\right\| \leq Cr\sqrt{n\log n}$.*

*Proof.* Since $\|\mathbf{X}_1\| \leq M$ almost surely, the coordinates of $\mathbf{X}_1$ are bounded random variables. By Hoeffding's inequality, we see that for any $j, k \in [r]$,

$$\mathbb{P}\left(\left|\left(\mathbf{X}^{\top}\mathbf{X}\right)_{jk} - n[\mathbb{E}[\mathbf{X}_1\mathbf{X}_1^{\top}]]_{jk}\right| \geq C\sqrt{n\log n}\right) \leq \frac{2}{n^2}.$$

From a union bound over $j, k \in [r]$, we have

$$\mathbb{P}\left( \left\| \mathbf{X}^\top \mathbf{X} - n\mathbb{E}\left[ \mathbf{X}_1 \mathbf{X}_1^\top \right] \right\|_F^2 \geq r^2 (C\sqrt{n \log n})^2 \right) \leq \frac{2r^2}{n^2}.$$

Using the fact that the Frobenius norm is an upper bound on the spectral norm, it follows that

$$\mathbb{P}\left( \left\| \mathbf{X}^\top \mathbf{X} - n\mathbb{E}\left[ \mathbf{X}_1 \mathbf{X}_1^\top \right] \right\| \geq Cr\sqrt{n \log n} \right) \leq \frac{2r^2}{n^2},$$

and the result follows after using our assumption in Equation (**A5**) to ensure that this bound converges to zero suitably quickly. $\qquad\square$

Bounds similar to the following appear in Levin et al. (2019). We give this stand-alone version for ease of reference and to incorporate the signal strength parameter $\rho_n$.

**Lemma C.3.** *Suppose that* $(\mathbf{A}, \mathbf{X}) \sim \mathrm{RDPG}(F, n)$ *with signal strength parameter* $\rho_n$, *with the condition that the latent positions are bounded almost surely. Then*

$$\|\mathbf{U}\|_{2,\infty} \lesssim \frac{\|\mathbf{X}\|_{2,\infty}}{\sqrt{n}}$$

*with high probability.*

*Proof.* Rearranging the identity $\mathbf{U}\mathbf{S}\mathbf{U}^\top = \mathbf{P} = \rho_n \mathbf{X}\mathbf{X}^\top$,

$$\mathbf{U} = \sqrt{\rho_n}\mathbf{X}\mathbf{S}^{-1/2}.$$

Applying basic properties of the $(2, \infty)$-norm (see, e.g., Cape et al., 2019b),

$$\|\mathbf{U}\|_{2,\infty} \leq \|\mathbf{X}\|_{2,\infty} \left\| \sqrt{\rho_n}\mathbf{S}^{-1/2} \right\|. \tag{30}$$

Noting that the diagonal entries of $\mathbf{S}/\rho_n$ are precisely the eigenvalues of $\mathbf{P}/\rho_n = \mathbf{X}\mathbf{X}^\top$, Lemma C.2 implies

$$\left\| \rho_n^{1/2}\mathbf{S}^{-1/2} \right\| = \frac{\rho_n^{1/2}}{\sqrt{s_r}} = O\left( \frac{1}{\sqrt{n}} \right)$$

with high probability, where we have used Assumption (**A4**). Applying this to Equation (30) completes the proof. $\qquad\square$

**Corollary C.4.** *Suppose that* $(\mathbf{A}, \mathbf{X}) \sim \mathrm{RDPG}(F, n)$ *with binary edges and sparsity parameter* $\rho_n$. *Then, with high probability,*

$$\|\mathbf{U}\|_{2,\infty} \lesssim \frac{1}{\sqrt{n}},$$

*Proof.* Applying Lemma C.3,

$$\|\mathbf{U}\|_{2,\infty} = O\left( \frac{\|\mathbf{X}\|_{2,\infty}}{\sqrt{n}} \right)$$

with high probability. The proof is complete once we note that $\|\mathbf{X}\|_{2,\infty} \leq 1$ under the binary RDPG. $\qquad\square$

## D. Delocalization Results

In this section, we prove that the eigenvectors of the weighted RDPG are delocalized, in the sense that the entries of the eigenvectors are bounded by $Cn^{-1/2} \log^c n$, for some constants $C, c > 0$. Our approach is based on earlier work by Erdős et al. (2013a) proving the local semicircle law for eigenvalues in the bulk under the Erdős–Rényi model, the expectation of which is a rank-one real symmetric matrix. We extend their approach to encompass the more general setting of a low-rank, real, and symmetric expectation. For background on the technical tools used in this section, we refer the interested reader to Erdős et al. (2013b); Benaych-Georges & Knowles (2018); Erdős & Yau (2017); Anderson et al. (2010).

The semicircle density is defined as

$$\mu_{\mathrm{sc}}(x) := \frac{1}{2\pi}\sqrt{4 - x^2}$$

for $x \in [-2, 2]$, with its associated Stieltjes transform for $\mathrm{Im}\, z > 0$,

$$m_{\mathrm{sc}}(z) := \int_{\mathbb{R}} \frac{\mu_{\mathrm{sc}}(x)}{x - z}\mathrm{d}x. \tag{31}$$

Our goal in this section is to show that for a suitably recentered and rescaled version of our adjacency matrix $\mathbf{A}$, the Stieltjes transform of its empirical eigenvalue density converges to $m_{\mathrm{sc}}(z)$. This is done in Theorem D.15 below. This result may be of independent interest to researchers working with low-rank signal-plus-noise network models like the RDPG. Most important for the present work, however, is that we may then use this result to show that the eigenvectors of $\mathbf{A}$ are delocalized, which we do in Theorem D.16.

### Notational Change: Eigenvalue Ordering and Submatrices

For this section and this section only, we make a notational change in order to agree with the convention of the vast majority of random matrix theory results (and, in particular, the convention followed in Erdős et al., 2013a, whose results we extend). In what follows, we number the eigenvalues from smallest to largest, rather than the largest-to-smallest indexing used in the main text. Thus, in this section, the eigenvalues $d_i$ of some matrix $\mathbf{D}$ will be indexed so that

$$d_1 \leq d_2 \leq \cdots \leq d_N.$$

That is, $d_N$ is the largest eigenvalue of $\mathbf{D}$. To mark this change of convention in what follows, we use $N$ rather than $n$ to denote the number of nodes in the network, but stress that $N = n$. The notational change is only to remind the reader of the altered indexing convention we are following. In this and a few other changes noted below, our notation largely follows that of Erdős et al. (2013a). This change in notation is for easier comparison between the results presented here and the results that they extend. For a matrix $\mathbf{D}$, we let $\mathbf{D}_{\cdot j}$ denote its $j$-th column. The entries of a matrix will be denoted with its non-bold variant, so that the $(i, j)$ entry of $\mathbf{D}$ is denoted as $D_{ij}$. We denote by $(\mathbf{D})^{\backslash i} \in \mathbb{R}^{(N-1) \times (N-1)}$ the principal submatrix of a matrix $\mathbf{D} \in \mathbb{R}^{N \times N}$ obtained by removing the $i$-th row and column. More generally, for some index set $\mathbb{T} \subset [N]$, we write $(\mathbf{D})^{\backslash \mathbb{T}}$ to mean the $(N - |\mathbb{T}|)$-by-$(N - |\mathbb{T}|)$ principal submatrix of $\mathbf{D}$ obtained by removing the $i$-th row and column from $\mathbf{D}$ for every $i \in \mathbb{T}$. If $\mathbf{d}$ and $d$ are used to denote the eigenvectors and eigenvalues, respectively of a matrix $\mathbf{D}$, we denote the eigenvectors and eigenvalues of $(\mathbf{D})^{\backslash \mathbb{T}}$ by $\mathbf{d}_i^{(\backslash \mathbb{T})}$ and $d_i^{(\backslash \mathbb{T})}$, respectively. The indices of eigenvectors and eigenvalues will be relabeled to be in the set $[N - |\mathbb{T}|]$, preserving their ordering described in beginning of this section: $d_1 \leq d_2 \leq \cdots \leq d_{N-|\mathbb{T}|}$. However, we keep the names of indices of $\mathbf{D}$ when defining $(\mathbf{D})^{\backslash \mathbb{T}}$.

As an example, consider the matrix

$$\mathbf{D} = \begin{bmatrix} a & b & c \\ d & e & f \\ g & h & i \end{bmatrix}.$$

Then

$$(\mathbf{D})^{\backslash 2} = \begin{bmatrix} a & c \\ g & i \end{bmatrix}.$$

We will overload the submatrix notation for vectors in the analogous way. For example, if a vector is given by $\mathbf{d} = (a, b, c)^\top$, then the vector $(\mathbf{d})^{\backslash 2} = (a, c)^\top$ has indices 1 and 3, skipping index 2. For an $N$-dimensional vector $\mathbf{d}$, let $(\mathbf{d})^{\backslash \mathbb{T}}$ denote the $(N - |\mathbb{T}|)$-dimensional vector obtained by removing every entry of $\mathbf{d}$ that is indexed by $\mathbb{T}$. Because we will often be working with submatrices and their individual entries, we adopt the following notation to indicate summing over indices:

$$\sum^{(\backslash \mathbb{T})} := \sum_{i \notin \mathbb{T}} = \sum_{i \in [N] \backslash \mathbb{T}}.$$

Furthermore, given an index set $\mathbb{T} \subset [N]$, we define $\mathbf{D}[\mathbb{T}^c] \in \mathbb{R}^{N \times N}$ to be the matrix obtained by setting all entries in the $i$-th column and row to zero for every element $i \in \mathbb{T}$. Similarly, we let $\mathbf{D}[\mathbb{T}]$ denote the matrix obtained by setting all entries

in $\mathbf{D}$ to zero *except* for the columns and rows indexed by $\mathbb{T}$. When the instance the index set consists of a single element $i$, we will simply denote the set by that element. That is, if $\mathbb{T} = \{i\}$, then $\mathbf{D}[i] := \mathbf{D}[\mathbb{T}]$. We denote the eigenvectors and eigenvalues of these "zeroed-out" matrices with the corresponding superscript. For example, if matrix $\mathbf{D}$ has eigenvector $\mathbf{d}$, then the corresponding eigenvector of $\mathbf{D}[*]$ will be denoted as $\mathbf{d}^{[*]}$. We continue use the constants $C$ and $c$ to denote positive constants whose value may change from line to line, but their value is always constant with respect to $N$.

The following definitions are largely adapted from Erdős et al. (2013a).

**Definition D.1.** An $N$-dependent event $\Omega$ holds with $(\xi, \nu)$-high probability if $\mathbb{P}(\Omega^c) \leq \exp\{-\nu(\log N)^\xi\}$ whenever $N \geq N_0(\nu, c_0, C_0)$ for $c_0, C_0$ in Equation (D.2). Moreover, for a given event $\Omega_0$, we say that $\Omega$ holds with $(\xi, \nu)$-high probability on $\Omega_0$ if $\mathbb{P}(\Omega_0 \cap \Omega^c) \leq \exp\{-\nu(\log N)^\xi\}$ for all $N \geq N_0(\nu, c_0, C_0)$.

**Definition D.2.** Let $\xi \equiv \xi(N)$ be such that

$$1 + c_0 \leq \xi \leq C_0 \log(\log N), \tag{32}$$

where $c_0 > 0$ and $C_0 \geq 10$. For $N = 1, 2, \ldots$, consider an $N \times N$ random matrix $\mathbf{H} = (H_{ij})$ whose entries are real and independent up to the symmetry constraint $H_{ij} = H_{ji}$. We assume the entries of $\mathbf{H}$ satisfy the moment conditions

$$\mathbb{E}[H_{ij}] = 0, \quad \mathbb{E}[|H_{ij}|^2] = \frac{1}{N}, \quad \mathbb{E}[|H_{ij}|^p] \leq \frac{C_1^p}{Nq^{p-2}} \tag{33}$$

where $3 \leq p \leq (\log N)^{C_0 \log \log N}$, $C_1 > 0$ and $q \equiv q(N)$ is a parameter that controls the tail behavior of the noise (Boucheron et al., 2013, Chapter 2), satisfying

$$r^2 (\log N)^{3\xi + 3\gamma} \leq q \leq C_2 N^{\frac{1}{2}} \tag{34}$$

for some $C_2 > 0$. We denote the eigenvalues of $\mathbf{H}$ as $\omega_1 \leq \omega_2 \leq \cdots \leq \omega_N$.

We next consider our non-zero expectation matrix, which is a scaled version of $\mathbf{P}$ from the main text (see Table 1 below for reference). Define an $N \times N$ real-valued, symmetric, and non-random rank $r$ matrix $\mathbf{L}$ with its spectral decomposition given as

$$\mathbf{L} = \sum_{\ell = N-r+1}^{N} \lambda_\ell \mathbf{v}_\ell \mathbf{v}_\ell^\top, \tag{35}$$

where the eigenvalues are given by

$$\lambda_1 = \lambda_2 = \cdots = \lambda_{N-r} = 0 < \lambda_{N-r+1} < \lambda_{N-r+2} < \cdots < \lambda_N.$$

Denote the sum of $\mathbf{H}$ and $\mathbf{L}$ by

$$\mathbf{B} = \mathbf{L} + \mathbf{H}, \tag{36}$$

with spectral decomposition given by

$$\mathbf{B} = \sum_{j=1}^{N} \hat{\lambda}_j \hat{\mathbf{v}}_j \hat{\mathbf{v}}_j^\top.$$

In relating the setup described here back to the main text, we note that

$$\mathbf{H} = \frac{1}{\sqrt{N}} \left( \mathbf{A} - \rho_n \mathbf{X}\mathbf{X}^\top \right) = \frac{1}{\sqrt{N}} \left( \mathbf{A} - \mathbf{P} \right).$$

In other words, $\mathbf{B}$ is a version of the observed adjacency matrix $\mathbf{A}$, scaled by a factor of $1/\sqrt{N}$. The scaling factor $1/\sqrt{N}$ is necessary in this section, because our objects of interest are related to the limiting distribution of the eigenvalues of a random matrix. Without this scaling factor, the eigenvalues of $\mathbf{A}$ would grow unbounded with $N$ for the RDPG (see, e.g., Lemma C.2), and there would be no limiting spectral distribution. The need for a limiting distribution of the eigenvalues is also why we assume the variance of the entries of $\mathbf{H}$ are $1/N$, although this assumption can be relaxed to allow for a more general doubly stochastic matrix of the variances of the entries of $\mathbf{H}$ (see Section 5 of Erdős et al., 2012). Our goal is to show that the eigenvectors of $\mathbf{B}$ (and hence also those of $\mathbf{A}$) associated with the perturbed null eigenvalues are delocalized. To do this, we study the distribution of eigenvalues of $\mathbf{B}$ and $\mathbf{H}$.

For the sake of clarity and ease of reference, we restate the Assumptions from the main text here, rewritten to match the notation in this appendix. Denote $\mathcal{S} := \{N - r + 1, \ldots, N\}$.

(**Ã1**) Assumption 1 (Delocalization): The eigenvectors associated with the non-zero eigenvalues of $\mathbf{L}$ are delocalized. That is, for some $\gamma \geq 0$ with no dependence on $N$,

$$|v_{ij}| \lesssim \frac{(\log N)^{\gamma}}{\sqrt{N}}$$

for all $i \in [N]$, and $j \in \mathcal{S}$ where $v_{ij}$ refers to the $i$-th entry of the $j$-th eigenvector of $\mathbf{L}$. To avoid confusion, this can be thought as indexing into the $(i, j)$-th entry of the matrix of eigenvectors $\mathbf{V}$.

(**Ã2**) Assumption 2: The signal strength parameter $\rho_N$ satisfies $\rho_N \sqrt{N} \gg 1$.

(**Ã3**) Assumption 3: The non-zero eigenvalues of $\mathbf{L}$ are distinct.

(**Ã4**) Assumption 4: The non-zero eigenvalues of $\mathbf{L}$ are of order $\rho_N \sqrt{N}$: $\lambda_j \asymp \rho_N \sqrt{N}$ for $j \in \mathcal{S}$.

(**Ã5**) Assumption 5: $\mathrm{rank}(\mathbf{L}) \lesssim (\log n)^{\zeta}$ for some $\zeta \geq 0$.

(**Ã6**) Assumption 6 (Eigengap): The eigengap $\Delta^{\circ} := \min_{j \in \mathcal{S}} \min_{i \neq j} |\lambda_i - \lambda_j|$ of $\mathbf{L}$ obeys $\Delta^{\circ} \gtrsim \rho_N \sqrt{N}$.

(**Ã7**) Assumption 7: The entries of $\mathbf{H}$ satisfy the moment conditions

$$\mathbb{E}\left[H_{ij}\right] = 0, \quad \mathbb{E}\left[|H_{ij}|^2\right] = \frac{1}{N}, \quad \text{and} \quad \mathbb{E}\left[|H_{ij}|^p\right] \leq \frac{C_1^p}{N q^{(p-2)}} \quad (p > 2).$$

We summarize the notational changes we have made in this appendix and their corresponding symbols from the main text in Table 1. The main differences are that in this appendix, we have rescaled the expectation matrix and random matrices by $1/\sqrt{N}$, and eigenvalue index ordering is reversed.

| Main Text | Appendix D |
|---|---|
| $n$ | $N$ |
| $\mathbf{A}$ | $\mathbf{B}$ |
| $\mathbf{P}$ | $\sqrt{N}\mathbf{L}$ |
| $\mathbf{E}$ | $\sqrt{N}\mathbf{H}$ |
| $\Delta$ | $\sqrt{N}\Delta^{\circ}$ |
| $s_1 \geq s_2 \geq \cdots \geq s_n$ | $\lambda_1 \leq \lambda_2 \leq \cdots \leq \lambda_N$ |

*Table 1.* Dictionary of the notational correspondence between the main text and this appendix.

We now introduce the empirical eigenvalue density functions for $\mathbf{B}$ and $\mathbf{H}$ respectively:

$$\tilde{\mu}(x) = \frac{1}{N} \sum_{\alpha} \delta(x - \hat{\lambda}_{\alpha}) \quad \text{and} \quad \mu(x) = \frac{1}{N} \sum_{\alpha} \delta(x - \omega_{\alpha}), \tag{37}$$

where $\delta(\cdot)$ is the Dirac measure. We define corresponding empirical cumulative distribution function (integrated empirical densities) according to

$$\tilde{n}(x) = \frac{1}{N} \left|\{\alpha : -\infty < \hat{\lambda}_{\alpha} \leq x\}\right| \tag{38}$$

and

$$n(x) = \frac{1}{N} \left|\{\alpha : -\infty < \omega_{\alpha} \leq x\}\right|. \tag{39}$$

These empirical densities are not the main focus for most of this section, as it is easier to work with their Stieltjes transforms. Showing the convergence of the Stieltjes transforms is equivalent to showing the convergence of the densities (see Lemma 2.1 in Benaych-Georges & Knowles, 2018).

As elsewhere, to study the Stieltjes transform, we consider the matrix resolvent (see, e.g., Kato, 1995). We denote the resolvents of $\mathbf{B}$ and $\mathbf{H}$ by, respectively,

$$\tilde{\mathbf{G}}(z) := (\mathbf{B} - z\mathbf{I})^{-1} \quad \text{and} \quad \mathbf{G}(z) := (\mathbf{H} - z\mathbf{I})^{-1}, \tag{40}$$

where $z \in \{x + iy : y > 0; x, y \in \mathbb{R}\}$. The trace of the resolvent is related to the Stieltjes transform of the empirical eigenvalue density function of $\mathbf{B}$ as (see, e.g., Benaych-Georges & Knowles, 2018, Equation (2.5))

$$\tilde{m}(z) := \int \frac{\tilde{\mu}(x)}{x - z} = \frac{1}{N} \operatorname{Tr} \tilde{\mathbf{G}}(z) dx \quad \text{and} \quad m(z) := \int \frac{\mu(x)}{x - z} dx = \frac{1}{N} \operatorname{Tr} \mathbf{G}(z). \tag{41}$$

In working with submatrices of $\mathbf{B}$, we denote corresponding resolvents by

$$\tilde{\mathbf{G}}^{(\backslash \mathbb{T})} := \left( (\mathbf{B})^{\backslash \mathbb{T}} - z \, (\mathbf{I})^{\backslash \mathbb{T}} \right)^{-1}.$$

Note that in defining the corresponding Stieltjes transform $\tilde{m}^{(\backslash \mathbb{T})}(z)$, we keep the $1/N$ normalization:

$$\tilde{m}^{(\backslash \mathbb{T})}(z) := \frac{1}{N} \operatorname{Tr} \tilde{\mathbf{G}}^{(\backslash \mathbb{T})}. \tag{42}$$

We are interested in understanding the error between $\tilde{m}(z)$ and $m_{\mathrm{sc}}(z)$. We denote these errors by

$$\tilde{\Lambda}(z) := |\tilde{m}(z) - m_{\mathrm{sc}}(z)|, \quad \Lambda(z) := |m(z) - m_{\mathrm{sc}}(z)|. \tag{43}$$

Below, we will consider complex numbers, parameterized by $E$ and $\eta$ as

$$z = E + i\eta,$$

where $E \in \mathbb{R}$ and $\eta > 0$ are chosen so that $z$ is near the support of the semicircle law. In particular, we consider the domain

$$D := \{z \in \mathbb{C} : |E| \leq \Sigma, 0 < \eta \leq 3\}, \tag{44}$$

and a subset $L$-dependent domain

$$D_L := \left\{ z \in \mathbb{C} : |E| \leq \Sigma, \frac{r^4 (\log N)^{L + 12\gamma}}{N} \leq \eta \leq 3 \right\} \tag{45}$$

where $\Sigma \geq 3$ is a fixed, arbitrary constant and $L \equiv L(N)$ satisfies

$$L \geq 8\xi, \tag{46}$$

where $\xi$ is given by Equation (32) in Definition D.2.

### D.1. Main Estimates

The following bounds are adapted from Section 7 of Erdős et al. (2013a). We make modifications to allow for a more general choice of $\mathbf{L}$ having rank $r \geq 1$. In particular, we allow for $\mathbf{L}$ to have finite, possibly slowly-growing rank, rather than the case of $\operatorname{rank} \mathbf{L} = 1$ considered in Erdős et al. (2013a).

**Lemma D.3.** *Let $\mathbf{B}$ satisfy Equation (36). Then for any $z \in D_L$, we have*

$$\left| \tilde{\Lambda}(z) - \Lambda(z) \right| \leq \frac{r\pi}{N\eta}$$

*Proof.* The proof largely follows that of Lemma 7.1 in Erdős et al. (2013a). The key difference is in the relation between $\tilde{n}(x)$ and $n(x)$. In our setting, $|\tilde{n}(x) - n(x)| \leq r/N$, which can be seen from Lemma D.25 and noticing that the eigenvalues of $\mathbf{H}$ and $\mathbf{B}$ are interlaced at most every $r$ indices, as we now demonstrate. Recalling the definitions of $\Lambda$ and $\tilde{\Lambda}$ from Equation (43) and applying the reverse triangle inequality,

$$|\tilde{\Lambda}(z) - \Lambda(z)| \leq |\tilde{m}(z) - m_{\mathrm{sc}}(z) - m(z) + m_{\mathrm{sc}}(z)| = |\tilde{m}(z) - m(z)|. \tag{47}$$

Recalling the definitions of $\tilde{m}(z)$ and $m(z)$ from Equation (41),

$$\tilde{m}(z) - m(z) = \int \frac{\tilde{\mu}(x) - \mu(x)}{x - z} dx = \int \frac{\tilde{n}(x) - n(x)}{(x - z)^2} dx,$$

where the second equality follows from integration by parts. Substituting this into Equation (47) and applying Jensen's inequality,

$$|\tilde{\Lambda}(z) - \Lambda(z)| \leq \int \frac{|\tilde{n}(x) - n(x)|}{|x - z|^2} dx. \tag{48}$$

Then Lemma D.25 implies the interlacing

$$\omega_i \leq \omega_{i+1} \leq \cdots \leq \omega_{i+r-1} \leq \hat{\lambda}_i \leq \cdots \leq \hat{\lambda}_{i+r-1} \leq \omega_{i+r} \leq \hat{\lambda}_{i+r}.$$

From these inequalities, we can see that there are at most $r$ eigenvalues of $\mathbf{H}$ between the $i$-th eigenvalue of $\mathbf{H}$ and the $i$-th eigenvalue of $\mathbf{B}$, since the $(i+r)$-th eigenvalue of $\mathbf{H}$ must come after the $i$-th eigenvalue of $\mathbf{B}$. Thus, for any $x$,

$$\left| |\{\alpha : -\infty < \omega_\alpha \leq x\}| - \left| \{\alpha : -\infty < \hat{\lambda}_\alpha \leq x\} \right| \right| \leq r.$$

It then follows that $|\tilde{n}(x) - n(x)| \leq r/N$. Applying this to Equation (48),

$$|\tilde{\Lambda}(z) - \Lambda(z)| \leq \frac{r}{N} \int_{-\infty}^{\infty} \frac{1}{|x - z|^2} dx = \frac{r}{N} \int_{-\infty}^{\infty} \frac{1}{(x - E)^2 + \eta^2} dx = \frac{r}{N\eta} \lim_{t \to \infty} \left[ \tan^{-1}(t) - \tan^{-1}(-t) \right] = \frac{r\pi}{N\eta},$$

as we set out to show. $\qquad\square$

**Lemma D.4.** *Let $\mathbf{B}$ satisfy Equation 36. For any $\mathbb{T} \subset [N]$ such that $0 < |\mathbb{T}| \leq \tau < N$, where $\tau$ is a constant with respect to $N$, denote $\mathbf{E}(\mathbb{T}) := \mathbf{H}[\mathbb{T}^c] - \mathbf{L}[\mathbb{T}]$. Then*

$$\|\mathbf{E}(\mathbb{T})\| \leq C(1 + r\sqrt{\tau}\rho_N (\log N)^{2\gamma}) \tag{49}$$

*with $(\xi, \nu)$-high-probability.*

*Proof.* Lemma 4.3 from Erdős et al. (2013a) states that with $(\xi, \nu)$-high-probability, we have

$$\|\mathbf{H}\| \leq 2 + \frac{(\log N)^\xi}{q^{1/2}}. \tag{50}$$

Recalling that $r^2 (\log N)^{3\xi + 3\gamma} \leq q$ by Equation (34),

$$\|\mathbf{H}\| \leq 2 + \frac{(\log N)^\xi}{(\log N)^{3\xi/2}} \leq C. \tag{51}$$

Now for any $\mathbb{T} \subset [N]$, we have

$$\|\mathbf{E}(\mathbb{T})\| = \|\mathbf{H}[\mathbb{T}^c] - \mathbf{L}[\mathbb{T}]\| \leq \|\mathbf{H}[\mathbb{T}^c]\| + \|\mathbf{L}[\mathbb{T}]\|.$$

By Theorem 4.3.17 from Horn & Johnson (1985), we have that $\|\mathbf{H}[\mathbb{T}^c]\| \leq \|\mathbf{H}\|$ for any $\mathbb{T} \subset [N]$. Thus, using Equation (51), it holds with $(\xi, \nu)$-high-probability that

$$\|\mathbf{E}(\mathbb{T})\| \leq C + \|\mathbf{L}[\mathbb{T}]\|. \tag{52}$$

To obtain an upper bound for $\|\mathbf{L}[\mathbb{T}]\|$, recall from Equation (35) that

$$\mathbf{L} = \sum_{\ell=N-r+1}^{N} \lambda_\ell \mathbf{v}_\ell \mathbf{v}_\ell^\top.$$

By Assumption ($\tilde{\mathbf{A}}$4), $c\rho_N \sqrt{N} \leq \lambda_\ell \leq C\rho_N \sqrt{N}$ for all $\ell \in \{N-r+1, N-r+2, \ldots, N\}$, and thus for each $k, j \in [N]$, we have

$$|L_{kj}| = \left| \left( \sum_{\ell=N-r+1}^{N} \lambda_\ell \mathbf{v}_\ell \mathbf{v}_\ell^\top \right)_{kj} \right| \leq C\rho_N \sqrt{N} \left| \left( \sum_{\ell=N-r+1}^{N} \mathbf{v}_\ell \mathbf{v}_\ell^\top \right)_{kj} \right|. \tag{53}$$

By Assumption ($\tilde{\mathbf{A}}$1), $\mathbf{v}_\ell \mathbf{v}_\ell^\top$ has entries of order $(\log N)^{2\gamma}/N$, whence

$$|L_{kj}| \leq C\left(\frac{r\rho_N(\log N)^{2\gamma}}{\sqrt{N}}\right).$$

It follows that, since $\mathbf{L}[\mathbb{T}]$ has $2N\tau - \tau^2$ non-zero entries by construction,

$$\|\mathbf{L}[\mathbb{T}]\| \leq \|\mathbf{L}[\mathbb{T}]\|_F \leq C(r\sqrt{\tau}\rho_N(\log N)^{2\gamma}).$$

Applying this bound to Equation (52) completes the proof. $\qquad\square$

**Lemma D.5.** *For each $\alpha = 1, 2, \ldots, N$, let $\hat{\mathbf{v}}_\alpha^{[\mathbb{T}^c]}$ be the orthonormal eigenvectors of $\mathbf{B}[\mathbb{T}^c] = \mathbf{L}[\mathbb{T}^c] + \mathbf{H}[\mathbb{T}^c]$. Denote $\mathbf{E}(\mathbb{T}) = \mathbf{H}[\mathbb{T}^c] - \mathbf{L}[\mathbb{T}]$. Then for all $j \in \{N - r + 1, N - r + 2, \ldots, N\}$ and any $\mathbb{T} \subset [N]$ where $0 < |\mathbb{T}| \leq \tau$, we have with $(\xi, \nu)$-high probability*

$$\sqrt{\sum_{\alpha \neq j} |\mathbf{v}_j^\top \hat{\mathbf{v}}_\alpha^{[\mathbb{T}^c]}|^2} \leq \frac{Cr(\log N)^{2\gamma}}{\lambda_j},$$

*where $\mathbf{v}_j$ is the eigenvector associated with the $j$-th smallest eigenvalue of $\mathbf{L}$.*

*Proof.* Following the proof of the Davis-Kahan $\sin\Theta$ theorem from Chapter 2 in (Chen et al., 2021b), for a given $j \in \{N - r + 1, \ldots, N\}$, we represent $\mathbf{L}$ via the eigendecomposition

$$\mathbf{L} = \begin{bmatrix} \mathbf{v}_j & \mathbf{U}_\perp \end{bmatrix} \begin{bmatrix} \lambda_j & 0 \\ 0 & \mathbf{M}_\perp \end{bmatrix} \begin{bmatrix} \mathbf{v}_j & \mathbf{U}_\perp \end{bmatrix}^\top, \tag{54}$$

where $\mathbf{U}_\perp$ is the $(N-1) \times (N-1)$ matrix of eigenvectors that span the orthogonal complement of $\mathbf{v}_j$, and $\mathbf{M}_\perp$ is the diagonal matrix of associated eigenvalues. Write the eigendecomposition of $\mathbf{B}[\mathbb{T}^c]$ similarly:

$$\mathbf{B}[\mathbb{T}^c] = \begin{bmatrix} \hat{\mathbf{v}}_j^{[\mathbb{T}^c]} & \widehat{\mathbf{U}}_\perp^{[\mathbb{T}^c]} \end{bmatrix} \begin{bmatrix} \hat{\lambda}_j^{[\mathbb{T}^c]} & 0 \\ 0 & \widehat{\mathbf{M}}_\perp^{[\mathbb{T}^c]} \end{bmatrix} \begin{bmatrix} \hat{\mathbf{v}}_j^{[\mathbb{T}^c]} & \widehat{\mathbf{U}}_\perp^{[\mathbb{T}^c]} \end{bmatrix}^\top, \tag{55}$$

where $\hat{\mathbf{v}}_j^{[\mathbb{T}^c]}$ is the eigenvector associated with $\hat{\lambda}_j^{[\mathbb{T}^c]}$, the $j$-th smallest eigenvalue of $\mathbf{B}[\mathbb{T}^c]$, $\widehat{\mathbf{U}}_\perp^{[\mathbb{T}^c]}$ is the matrix of orthonormal eigenvectors that span the orthogonal complement of $\hat{\mathbf{v}}_j^{[\mathbb{T}^c]}$, and $\widehat{\mathbf{M}}_\perp^{[\mathbb{T}^c]}$ is the diagonal matrix of associated eigenvalues.

By construction, $\mathbf{B}[\mathbb{T}^c] = \mathbf{L} + \mathbf{H}[\mathbb{T}^c] - \mathbf{L}[\mathbb{T}] = \mathbf{L} + \mathbf{E}(\mathbb{T})$, and thus we may view $\mathbf{E}(\mathbb{T})$ as the error between $\mathbf{L}$ and $\mathbf{B}[\mathbb{T}^c]$. That is,

$$\mathbf{E}(\mathbb{T}) = \mathbf{B}[\mathbb{T}^c] - \mathbf{L} \tag{56}$$

We will now employ a centering trick to bound the eigenvalues of $\widehat{\mathbf{M}}_\perp^{[\mathbb{T}^c]}$ away from 0, by working with the surrogate matrices

$$\tilde{\mathbf{L}} = \mathbf{L} - \lambda_j \mathbf{I} \quad \text{and} \quad \tilde{\mathbf{B}}[\mathbb{T}^c] = \mathbf{B}[\mathbb{T}^c] - \lambda_j \mathbf{I}. \tag{57}$$

From these surrogate matrices, we can see that

$$\mathbf{E}(\mathbb{T}) = \mathbf{B}[\mathbb{T}^c] - \mathbf{L} = \mathbf{B}[\mathbb{T}^c] - \lambda_j \mathbf{I} + \lambda_j \mathbf{I} - \mathbf{L} = \tilde{\mathbf{B}}[\mathbb{T}^c] - \tilde{\mathbf{L}}.$$

We observe that by Lemma C.1, the eigenvalues on the diagonal of $\widehat{\mathbf{M}}_\perp^{[\mathbb{T}^c]}$ are contained in the set $(-\infty, \lambda_j - \Delta^\circ + \|\mathbf{E}(\mathbb{T})\|] \cup [\lambda_j + \Delta^\circ - \|\mathbf{E}(\mathbb{T})\|, \infty)$. The eigenvalues of $\tilde{\mathbf{L}}$ and $\tilde{\mathbf{B}}[\mathbb{T}^c]$ are derived by applying the map $x \mapsto x - \lambda_j$ to the spectra of $\mathbf{L}$ and $\mathbf{B}[\mathbb{T}^c]$, respectively, and therefore,

$$\sigma_{\min}(\widetilde{\mathbf{M}}_\perp^{[\mathbb{T}^c]}) \geq \Delta^\circ - \|\mathbf{E}(\mathbb{T})\|. \tag{58}$$

We denote the eigendecomposition of these surrogate matrices using the same notation as in Equations (54) and (55) but with an additional tilde ($\tilde{\phantom{x}}$) above each object. Noting that the surrogate matrices share the same eigenvectors as their original matrices, we have

$$\widetilde{\mathbf{U}}_\perp^{[\mathbb{T}^c]} = \mathbf{U}_\perp \quad \text{and} \quad \widetilde{\widehat{\mathbf{U}}}_\perp^{[\mathbb{T}^c]} = \widehat{\mathbf{U}}_\perp^{[\mathbb{T}^c]}. \tag{59}$$

We then examine the quantity

$$\left(\widetilde{\widehat{\mathbf{U}}}_{\perp}^{[\mathbb{T}^c]}\right)^{\top} \mathbf{E}(\mathbb{T})\tilde{\mathbf{v}}_j = \left(\widehat{\mathbf{U}}_{\perp}^{[\mathbb{T}^c]}\right)^{\top} \mathbf{E}(\mathbb{T})\mathbf{v}_j = \left(\widehat{\mathbf{U}}_{\perp}^{[\mathbb{T}^c]}\right)^{\top} \left(\widetilde{\mathbf{B}}[\mathbb{T}^c] - \widetilde{\mathbf{L}}\right)\mathbf{v}_j = \widetilde{\widetilde{\mathbf{M}}}_{\perp}^{[\mathbb{T}^c]} \left(\widehat{\mathbf{U}}_{\perp}^{[\mathbb{T}^c]}\right)^{\top} \mathbf{v}_j,$$

where the first equality follows from Equation (59); the second equality is obtained from substituting $\mathbf{E}(\mathbb{T})$ from Equation (56); and the third equality follows from the fact that $\widetilde{\mathbf{L}}\mathbf{v}_j = 0$ by construction. Taking the Frobenius norm, we have

$$\left\|\left(\widehat{\mathbf{U}}_{\perp}^{[\mathbb{T}^c]}\right)^{\top} \mathbf{E}(\mathbb{T})\mathbf{v}_j\right\|_F = \left\|\widetilde{\widetilde{\mathbf{M}}}_{\perp}^{[\mathbb{T}^c]} \left(\widehat{\mathbf{U}}_{\perp}^{[\mathbb{T}^c]}\right)^{\top} \mathbf{v}_j\right\|_F \geq \sigma_{\min}\left(\widetilde{\widetilde{\mathbf{M}}}_{\perp}^{[\mathbb{T}^c]}\right)\left\|\left(\widehat{\mathbf{U}}_{\perp}^{[\mathbb{T}^c]}\right)^{\top} \mathbf{v}_j\right\|_F$$

$$\geq (\Delta^{\circ} - \|\mathbf{E}(\mathbb{T})\|)\left\|\left(\widehat{\mathbf{U}}_{\perp}^{[\mathbb{T}^c]}\right)^{\top} \mathbf{v}_j\right\|_F,$$

where the first inequality is from basic properties of unitarily invariant norms and the second inequality follows from Equation (58). Hence, by basic properties of the norms,

$$(\Delta^{\circ} - \|\mathbf{E}(\mathbb{T})\|)\left\|\left(\widehat{\mathbf{U}}_{\perp}^{[\mathbb{T}^c]}\right)^{\top} \mathbf{v}_j\right\|_F \leq \left\|\left(\widehat{\mathbf{U}}_{\perp}^{[\mathbb{T}^c]}\right)^{\top} \mathbf{E}(\mathbb{T})\mathbf{v}_j\right\|_F \leq \left\|\left(\widehat{\mathbf{U}}_{\perp}^{[\mathbb{T}^c]}\right)^{\top}\right\| \|\mathbf{E}(\mathbb{T})\mathbf{v}_j\|_F \leq \|\mathbf{E}(\mathbb{T})\mathbf{v}_j\|_F.$$

Lower bounding $\Delta^{\circ}$ using Assumption ($\tilde{\mathbf{A}}6$) and upper-bounding $\|\mathbf{E}(\mathbb{T})\|$ using Lemma D.4, Assumptions ($\tilde{\mathbf{A}}2$) and ($\tilde{\mathbf{A}}5$) ensure that $\Delta^{\circ} - \|\mathbf{E}(\mathbb{T})\| > 0$ for suitably large $n$ and thus

$$\left\|\left(\widehat{\mathbf{U}}_{\perp}^{[\mathbb{T}^c]}\right)^{\top} \mathbf{v}_j\right\|_F \leq \frac{\|\mathbf{E}(\mathbb{T})\mathbf{v}_j\|_F}{\Delta^{\circ} - \|\mathbf{E}(\mathbb{T})\|}.$$

Now since

$$\sqrt{\sum_{\alpha \neq j} |\mathbf{v}_j^{\top}\hat{\mathbf{v}}_{\alpha}^{[\mathbb{T}^c]}|^2} = \left\|\left(\widehat{\mathbf{U}}_{\perp}^{[\mathbb{T}^c]}\right)^{\top} \mathbf{v}_j\right\|_F,$$

we have

$$\sqrt{\sum_{\alpha \neq j} |\mathbf{v}_j^{\top}\hat{\mathbf{v}}_{\alpha}^{[\mathbb{T}^c]}|^2} \leq \frac{\|\mathbf{E}(\mathbb{T})\mathbf{v}_j\|_F}{\Delta^{\circ} - \|\mathbf{E}(\mathbb{T})\|} \leq \frac{\|\mathbf{E}(\mathbb{T})\|}{\Delta^{\circ} - \|\mathbf{E}(\mathbb{T})\|} \leq \frac{C(1 + r\rho_N(\log N)^{2\gamma})}{\Delta^{\circ} - (1 + r\rho_N(\log N)^{2\gamma})}$$

where the last inequality holds with $(\xi, \nu)$-high-probability, by Lemma D.4. Note that we have absorbed $\tau$ into the constant $C$.

Hence, by Assumption $\tilde{\mathbf{A}}6$, which states $\Delta^{\circ} = \Omega\left(\rho_N\sqrt{N}\right)$, we have shown that

$$\sqrt{\sum_{\alpha \neq j} |\mathbf{v}_j^{\top}\hat{\mathbf{v}}_{\alpha}^{[\mathbb{T}^c]}|^2} \leq \frac{C(1 + r\tau\rho_N(\log N)^{2\gamma})}{\Delta^{\circ} - (1 + r\tau\rho_N(\log N)^{2\gamma})} \leq \frac{C(1 + r\rho_N(\log N)^{2\gamma})}{\rho_N\sqrt{N} - (1 + r\rho_N(\log N)^{2\gamma})}$$

$$= \frac{C(1 + r\rho_N(\log N)^{2\gamma})}{\rho_N\sqrt{N}(1 - (1 + r\rho_N(\log N)^{2\gamma})/\rho_N\sqrt{N})}.$$

with $(\xi, \nu)$-high-probability. Then since (1) $\rho_N\sqrt{N} \gg 1$ by Assumption ($\tilde{\mathbf{A}}2$) and (2) $r(\log N)^{2\gamma} \ll \sqrt{N}$ by Assumption ($\tilde{\mathbf{A}}5$), we have that $1 + r\rho_N(\log N)^{2\gamma} \ll \rho_N\sqrt{N}$. It then follows that,

$$\sqrt{\sum_{\alpha \neq j} |\mathbf{v}_j^{\top}\hat{\mathbf{v}}_{\alpha}^{[\mathbb{T}^c]}|^2} \leq \frac{C(1 + r\rho_N(\log N)^{2\gamma})}{\rho_N\sqrt{N}}.$$

with $(\xi, \nu)$-high-probability. To simplify presentation in the following sections, we will $\rho_N$ trivially by 1 and absorb the other summand. So then

$$\sqrt{\sum_{\alpha \neq j} |\mathbf{v}_j^{\top}\hat{\mathbf{v}}_{\alpha}^{[\mathbb{T}^c]}|^2} \leq \frac{C}{\rho_N\sqrt{N}}r(\log N)^{2\gamma} \tag{60}$$

with $(\xi, \nu)$-high-probability. Recalling that $\lambda_j \asymp \rho_N \sqrt{N}$ by Assumption ($\tilde{\mathbf{A}}$4), we have that with $(\xi, \nu)$-high-probability,

$$\sqrt{\sum_{\alpha \neq j} |\mathbf{v}_j^\top \hat{\mathbf{v}}_\alpha^{[\mathbb{T}^c]}|^2} \leq \frac{C}{\lambda_j} r (\log N)^{2\gamma}. \tag{61}$$

Noting that the above argument only relied on the probability guarantee in Lemma D.4 and our choice of $\mathbb{T}$ and $j$ were otherwise arbitrary, this estimate holds for any $\mathbb{T} \subset [N]$ and $j \in \{N-r+1, N-r+2, \ldots, N\}$ with $(\xi, \nu)$-high-probability, as we set out to show. $\qquad\square$

The following lemma largely mirrors Lemma 7.2 from (Erdős et al., 2013a), but there are more terms in the matrix $\tilde{\mathcal{A}}_i(z)$ owing to the fact that we now have a rank-$r$ matrix $\mathbf{L}$, instead of the rank-one case considered by (Erdős et al., 2013a). Below, for a vector $F = (F_1, F_2, \ldots, F_N)$, we write

$$\langle F \rangle := \frac{1}{N} \sum_{i=1}^N F_i. \tag{62}$$

**Lemma D.6.** *Let $\mathbf{B}$ be as given in Equation* (36) *and denote the resolvent of $\mathbf{B}$ by $\tilde{\mathbf{G}}(z) = (\mathbf{B} - z\mathbf{I})^{-1}$ as in Equation* (40). *Then*

$$\tilde{G}_{ii}(z) = \frac{1}{-z - m_{\mathrm{sc}}(z) - \left(\langle \tilde{\psi}(z) \rangle - \tilde{Y}_i(z)\right)},$$

*where*

$$\tilde{Y}_i(z) = H_{ii} - \tilde{Z}_i(z) + \tilde{\mathcal{A}}_i(z), \tag{63}$$

*and*

$$\tilde{Z}_i(z) = Q_i\left(((\mathbf{B}_{\cdot\mathbf{i}})^{\backslash i})^\top \tilde{\mathbf{G}}^{(\backslash i)}(z)(\mathbf{B}_{\cdot\mathbf{i}})^{\backslash i}\right) \tag{64}$$

*and, for a matrix $\mathbf{D}$,*

$$Q_i(\mathbf{D}) = \mathbf{D} - \mathbb{E}\left[\mathbf{D} \mid (\mathbf{H})^{\backslash i}\right] \tag{65}$$

*and*

$$\tilde{\mathcal{A}}_i(z) = L_{ii} - \sum_{\ell=N-r+1}^N \lambda_\ell^2 (v_{i\ell}(\mathbf{v}_\ell)^{\backslash i})^\top \tilde{\mathbf{G}}^{(\backslash i)}(z) v_{i\ell}(\mathbf{v}_\ell)^{\backslash i} - \sum_{\ell_1 \neq \ell_2} \lambda_{\ell_1} \lambda_{\ell_2} (v_{i\ell_1}(\mathbf{v}_{\ell_1})^{\backslash i})^\top \tilde{\mathbf{G}}^{(\backslash i)}(z) v_{i\ell_2}(\mathbf{v}_{\ell_2})^{\backslash i}$$
$$+ \frac{1}{N} \sum_j \frac{\tilde{G}_{ij}(z)\tilde{G}_{ji}(z)}{\tilde{G}_{ii}(z)}. \tag{66}$$

*Moreover,*

$$\tilde{\psi}_i(z) := \tilde{G}_{ii}(z) - m_{\mathrm{sc}}(z). \tag{67}$$

*Proof.* By definition of $\tilde{\mathbf{G}}(z)$, taking Schur complements yields

$$\tilde{G}_{ii}(z) = \left(-z + B_{ii} - \left((\mathbf{B}_{\cdot\mathbf{i}})^{\backslash i}\right)^\top \tilde{\mathbf{G}}^{(\backslash i)}(z)(\mathbf{B}_{\cdot\mathbf{i}})^{\backslash i}\right)^{-1}. \tag{68}$$

Looking at the quadratic form, by the definition of $\mathbf{B}$ from (36), we have

$$\left((\mathbf{B}_{\cdot\mathbf{i}})^{\backslash i}\right)^\top \tilde{\mathbf{G}}^{(\backslash i)}(z)(\mathbf{B}_{\cdot\mathbf{i}})^{\backslash i} = \left[((\mathbf{H}_{\cdot\mathbf{i}})^{\backslash i})^\top + \sum_{\ell=N-r+1}^N \lambda_\ell (v_{i\ell}(\mathbf{v}_\ell)^{\backslash i})^\top\right] \tilde{\mathbf{G}}^{(\backslash i)}(z) \left[(\mathbf{H}_{\cdot\mathbf{i}})^{\backslash i} + \sum_{\ell=N-r+1}^N \lambda_\ell (v_{i\ell}(\mathbf{v}_\ell)^{\backslash i})\right],$$

where we remind the reader that $(\mathbf{H})^{\backslash i}$ denotes the $(N-1) \times (N-1)$ submatrix of $\mathbf{H}$ where the $i$-th rows and columns were removed; the notation $\mathbf{H}_{\cdot i} \in \mathbb{R}^N$ denotes the $i$-th row of $\mathbf{H}$; and hence $(\mathbf{H}_{\cdot\mathbf{i}})^{\backslash i} \in \mathbb{R}^{N-1}$ denotes the $i$-th row of $\mathbf{H}$

with its $i$-th entry removed. Expanding and taking the conditional expectation given $(\mathbf{H})^{\backslash i}$, the terms that are linear in $(\mathbf{H}_{\cdot \mathbf{i}})^{\backslash i}$ disappear, since the elements of $(\mathbf{H}_{\cdot \mathbf{i}})^{\backslash i}$ are not included in the matrix $(\mathbf{H})^{\backslash i}$, and we have

$$
\begin{aligned}
\mathbb{E}\left[((\mathbf{B}_{\cdot \mathbf{i}})^{\backslash i})^{\top} \tilde{\mathbf{G}}^{(\backslash i)}(z)(\mathbf{B}_{\cdot \mathbf{i}})^{\backslash i} \mid (\mathbf{H})^{\backslash i}\right] = {}& \mathbb{E}\left[((\mathbf{H}_{\cdot \mathbf{i}})^{\backslash i})^{\top} \tilde{\mathbf{G}}^{(\backslash i)}(z)(\mathbf{H}_{\cdot \mathbf{i}})^{\backslash i} \mid (\mathbf{H})^{\backslash i}\right] \\
& + \sum_{\ell=N-r+1}^{N} \lambda_{\ell}^2 \, (v_{i\ell}(\mathbf{v}_{\ell})^{\backslash i})^{\top} \tilde{\mathbf{G}}^{(\backslash i)}(z) v_{i\ell}(\mathbf{v}_{\ell})^{\backslash i} \\
& + \sum_{\ell_1 \neq \ell_2} \lambda_{\ell_1} \, \lambda_{\ell_2} \, (v_{i\ell_1}(\mathbf{v}_{\ell_1})^{\backslash i})^{\top} \tilde{\mathbf{G}}^{(\backslash i)}(z) v_{i\ell_2}(\mathbf{v}_{\ell_2})^{\backslash i}.
\end{aligned}
\tag{69}
$$

Recalling the definition of $\tilde{\mathcal{A}}_i(z)$ from Equation (66), it follows that

$$
\begin{aligned}
& \mathbb{E}\left[((\mathbf{B}_{\cdot \mathbf{i}})^{\backslash i})^{\top} \tilde{\mathbf{G}}^{(\backslash i)}(z)(\mathbf{B}_{\cdot \mathbf{i}})^{\backslash i} \mid (\mathbf{H})^{\backslash i}\right] \\
& \qquad = \mathbb{E}\left[((\mathbf{H}_{\cdot \mathbf{i}})^{\backslash i})^{\top} \tilde{\mathbf{G}}^{(\backslash i)}(z)(\mathbf{H}_{\cdot \mathbf{i}})^{\backslash i} \mid (\mathbf{H})^{\backslash i}\right] - \tilde{\mathcal{A}}_i(z) + L_{ii} + \frac{1}{N}\sum_j \frac{\tilde{G}_{ij}(z)\tilde{G}_{ji}(z)}{\tilde{G}_{ii}(z)}.
\end{aligned}
\tag{70}
$$

The first term in Equation (70) can be expressed as

$$
\begin{aligned}
\mathbb{E}\left[((\mathbf{H}_{\cdot \mathbf{i}})^{\backslash i})^{\top} \tilde{\mathbf{G}}^{(\backslash i)}(z)(\mathbf{H}_{\cdot \mathbf{i}})^{\backslash i} \mid (\mathbf{H}_{\cdot \mathbf{i}})^{\backslash i}\right] &= \frac{1}{N}\sum_j^{(\backslash i)} \tilde{G}_{jj}^{(\backslash i)}(z) = \frac{1}{N}\sum_j \left( \tilde{G}_{jj}(z) - \frac{\tilde{G}_{ji}(z)\tilde{G}_{ij}(z)}{\tilde{G}_{ii}(z)} \right) \\
&= \frac{1}{N}\sum_j \tilde{G}_{jj}(z) - \frac{1}{N}\sum_j \frac{\tilde{G}_{ji}(z)\tilde{G}_{ij}(z)}{\tilde{G}_{ii}(z)},
\end{aligned}
\tag{71}
$$

where in the first equality, we have used the fact the variances of $H_{ii}$ are $1/N$, along with the fact that $\mathbb{E}\left[H_{ij}\right] = 0$, and in the second equality, we have used basic properties of the entries of resolvents, given in Lemma D.17. Substituting Equation (71) into Equation (70) yields

$$
\mathbb{E}\left[((\mathbf{B}_{\cdot \mathbf{i}})^{\backslash i})^{\top} \tilde{\mathbf{G}}^{(\backslash i)}(z)(\mathbf{B}_{\cdot \mathbf{i}})^{\backslash i} \mid (\mathbf{H})^{\backslash i}\right] = \frac{1}{N}\sum_j \tilde{G}_{jj}(z) - \tilde{\mathcal{A}}_i(z) + L_{ii}
\tag{72}
$$

Recalling the definition of $Q_i(\cdot)$ from Equation (65), let us write

$$
\tilde{Z}_i(z) = Q_i\left( ((\mathbf{B}_{\cdot \mathbf{i}})^{\backslash i})^{\top} \tilde{\mathbf{G}}^{(\backslash i)}(z)(\mathbf{B}_{\cdot \mathbf{i}})^{\backslash i} \right)
$$

for ease of notation. Now, consider

$$
\begin{aligned}
B_{ii} - ((\mathbf{B}_{\cdot \mathbf{i}})^{\backslash i})^{\top} \tilde{\mathbf{G}}^{(\backslash i)}(z)(\mathbf{B}_{\cdot \mathbf{i}})^{\backslash i} = {}& B_{ii} - ((\mathbf{B}_{\cdot \mathbf{i}})^{\backslash i})^{\top} \tilde{G}^{(\backslash i)}(z)(\mathbf{B}_{\cdot \mathbf{i}})^{\backslash i} + \mathbb{E}\left[((\mathbf{B}_{\cdot \mathbf{i}})^{\backslash i})^{\top} \tilde{\mathbf{G}}^{(\backslash i)}(z)(\mathbf{B}_{\cdot \mathbf{i}})^{\backslash i} \mid (\mathbf{H})^{\backslash i}\right] \\
& - \mathbb{E}\left[((\mathbf{B}_{\cdot \mathbf{i}})^{\backslash i})^{\top} \tilde{\mathbf{G}}^{(\backslash i)}(z)(\mathbf{B}_{\cdot \mathbf{i}})^{\backslash i} \mid (\mathbf{H})^{\backslash i}\right] \\
= {}& B_{ii} - \mathbb{E}\left[((\mathbf{B}_{\cdot \mathbf{i}})^{\backslash i})^{\top} \tilde{\mathbf{G}}^{(\backslash i)}(z)(\mathbf{B}_{\cdot \mathbf{i}})^{\backslash i} \mid (\mathbf{H})^{\backslash i}\right] - \tilde{Z}_i(z) \\
= {}& B_{ii} - \left( \frac{1}{N}\sum_j \tilde{G}_{jj}(z) - \tilde{\mathcal{A}}_i(z) + L_{ii} \right) - \tilde{Z}_i(z) \\
= {}& L_{ii} + H_{ii} - \frac{1}{N}\sum_j \tilde{G}_{jj}(z) + \tilde{\mathcal{A}}_i(z) - L_{ii} - \tilde{Z}_i(z) + m_{\mathrm{sc}}(z) - m_{\mathrm{sc}}(z) \\
= {}& H_{ii} - \frac{1}{N}\sum_j \tilde{G}_{jj}(z) + \tilde{\mathcal{A}}_i(z) - \tilde{Z}_i(z) + m_{\mathrm{sc}}(z) - m_{\mathrm{sc}}(z).
\end{aligned}
$$

where in the second equality, we have substituted the definition of $\tilde{Z}_i(z)$; and in the third equality, we have substituted the expansion of the conditional expectation derived in Equation (72). Rearranging terms and recalling our bracket notation

defined in Equation (62), we find

$$B_{ii} - ((\mathbf{B}_{\cdot\mathbf{i}})^{\backslash i})^\top \tilde{G}^{(\backslash i)}(z)(\mathbf{B}_{\cdot\mathbf{i}})^{\backslash i} = -m_{\mathrm{sc}}(z) + m_{\mathrm{sc}}(z) - \frac{1}{N}\sum_j \tilde{G}_{jj}(z) + H_{ii} - \quad \tilde{Z}_i(z) + \tilde{\mathcal{A}}_i(z)$$

$$= -m_{\mathrm{sc}}(z) - \langle \tilde{\psi}(z) \rangle + \tilde{Y}_i(z),$$

where we have absorbed the $m_{\mathrm{sc}}(z)$ term into $\langle \tilde{\psi}(z) \rangle$ (see Equation (67) for the definition of $\tilde{\psi}$) and we have used the definition of $\tilde{Y}_i(z)$ from Equation (63). Substituting this into Equation (68),

$$\tilde{G}_{ii}(z) = \left( -z - m_{\mathrm{sc}}(z) - \langle \tilde{\psi}(z) \rangle + \tilde{Y}_i(z) \right)^{-1},$$

as we set out to show. $\qquad\square$

We now introduce two new quantities:

$$\tilde{\Lambda}_o(z) := \max_{i \neq j} |\tilde{G}_{ij}(z)| \quad \text{and} \quad \tilde{\Lambda}_d(z) := \max_i |\tilde{G}_{ii}(z) - m_{\mathrm{sc}}(z)|. \tag{73}$$

The error terms from Lemma D.6—$\tilde{\mathcal{A}}_i(z), \tilde{Z}_i(z),$ and $\tilde{Y}_i(z)$—are often estimated in terms of these quantities, which in turn are bounded by a deterministic $\Phi(z)$.

Next, for $\eta$ satisfying $N^{-1}r^4(\log N)^{L+12\gamma} \leq \eta \leq 3$, define the set $D(\eta) := \{z \in D_L : \mathrm{Im}\, z = \eta\}$ and the event

$$\tilde{\Omega}(\eta) := \left\{ \sup_{z \in D(\eta)} (\tilde{\Lambda}_d(z) + \tilde{\Lambda}_o(z)) \leq (\log N)^{-\xi} \right\}. \tag{74}$$

We will need Lemma 7.4 from Erdős et al. (2013a). No adaptation to the present $r > 1$ setting is necessary, since the proof only involves basic properties of resolvents (see the proof of Lemma 3.12 in Erdős et al., 2013a).

**Lemma D.7** (Erdős et al. (2013a), Lemma 7.4)**.** *Let $z = E + \mathrm{i}\eta \in D_L$, we have for any $i$ and $\mathbb{T} \subset \{1, \ldots, N\}$ satisfying $i \notin \mathbb{T}$ and $|\mathbb{T}| \leq \tau < N$, where $\tau$ is constant with respect to $N$, that*

$$\tilde{m}^{(\{i\}\cup\mathbb{T})}(z) = \tilde{m}^{(\mathbb{T})}(z) + O\left(\frac{1}{N\eta}\right) \tag{75}$$

*holds on $\tilde{\Omega}(\eta)$.*

Next, we adapt Lemma 7.5 from Erdős et al. (2013a). To do so, we first introduce a few simple concentration inequalities.

**Lemma D.8** (Erdős et al. (2013a), Lemma 3.8)**.** *Let $a_1, \ldots, a_N$ be centered and independent random variables satisfying*

$$\mathbb{E}\left[|a_i|^p\right] \leq \frac{C_1^p}{Nq^{p-2}} \tag{76}$$

*for $2 \leq p \leq (\log N)^{C_0 \log \log N}$. Then there is a $\nu > 0$, depending only on $C_1$, such that for all $\xi$, and for any $A_i, F_{ij} \in \mathbb{C}$, we have with $(\xi, \nu)$-high-probability*

$$\left| \sum_{i=1}^N A_i a_i \right| \leq (\log N)^\xi \left[ \frac{\max_i |A_i|}{q} + \left( \frac{1}{N}\sum_{i=1}^N |A_i|^2 \right)^{1/2} \right], \tag{77}$$

$$\left| \sum_{i=1}^N a_i F_{ii} a_i - \sum_{i=1}^N \sigma_i^2 F_{ii} \right| \leq (\log N)^\xi \frac{\max_i |F_{ii}|}{q}, \tag{78}$$

*and*

$$\left| \sum_{1 \leq i \neq j \leq N} a_i F_{ij} a_j \right| \leq (\log N)^{2\xi} \left( \frac{\max_{i \neq j} |F_{ij}|}{q} + \left( \frac{1}{N^2}\sum_{i \neq j} |F_{ij}|^2 \right)^{1/2} \right). \tag{79}$$

*Let $a_1, \ldots, a_N$ and $b_1, \ldots, f_N$ be mutually independent random variables, each satisfying Equation (76). Then there is a $\nu > 0$ depending only on $C$ such that for all $\xi$ and $F_{ij} \in \mathbb{C}$, we have with $(\xi, \nu)$-high-probability*

$$\left| \sum_{i,j=1}^{N} a_i F_{ij} b_j \right| \leq (\log N)^{2\xi} \left[ \frac{\max_i |F_{ii}|}{q^2} + \frac{\max_{i \neq j} |F_{ij}|}{q} + \left( \frac{1}{N^2} \sum_{i \neq j} |F_{ij}|^2 \right)^{1/2} \right] \tag{80}$$

**Lemma D.9** (Erdős et al. (2013a), Lemma 7.5, adapted)**.** *Let $\tau < N$ be constant with respect to $N$, and let $\mathbf{B}$ be as in Equation (36). Let $z = E + i\eta \in D_L$, we have for any $\mathbb{T} \subset [N]$ satisfying $|\mathbb{T}| \leq \tau < N$, as long as $i \in \mathbb{T}$, that for $j \in \{N - r + 1, N - r + 2, \ldots, N\}$,*

$$\left| \lambda_j (v_{ij}(\mathbf{v}_j)^{\backslash \mathbb{T}})^\top \tilde{\mathbf{G}}^{(\backslash \mathbb{T})}(z)(\mathbf{H}_{\cdot i})^{\backslash \mathbb{T}} \right| \leq Cr(\log N)^{\xi + 3\gamma} \left( \frac{1}{q} + \sqrt{\frac{\operatorname{Im} \tilde{m}(z)}{N\eta}} + \frac{1}{N\eta} \right) \tag{81}$$

*with $(\xi, \nu)$-high probability on the event $\tilde{\Omega}(\eta)$, where $q$ satisfies Equation (34).*

*Proof.* We note that our proof of this bound deviates slightly from Erdős et al. (2013a). Our proof is simplified because we assume that the signal eigenvalues of $\mathbf{L}$ obey $\lambda_i \asymp \rho_N \sqrt{N}$ by Assumption (**Ã4**).

First, with $\mathcal{S} = \{N - r + 1 - |\mathbb{T}|, N - r + 2 - |\mathbb{T}|, \ldots, N - |\mathbb{T}|\}$, define

$$R := \max_{|\mathbb{T}| \leq \tau} \max_{\alpha \notin \mathcal{S}} \max_j |\hat{v}_{j\alpha}^{(\backslash \mathbb{T})}|.$$

Recall the event $\tilde{\Omega}(\eta)$ from Equation (74). On $\tilde{\Omega}(\eta)$, we have by Lemma D.22,

$$c \leq |\tilde{G}_{ii}^{(\backslash \mathbb{T})}(z)| \leq C,$$

so that for $z := \hat{\lambda}_\alpha^{(\backslash \mathbb{T})} + i\eta \in D_L$ and $\alpha \notin \mathcal{S}$,

$$C \geq \operatorname{Im} \tilde{G}_{ii}^{(\backslash \mathbb{T})}(\hat{\lambda}_\alpha^{(\backslash \mathbb{T})} + i\eta) = \sum_{\beta \in [N - |\mathbb{T}|]} \frac{\eta |\hat{v}_{i\beta}^{(\backslash \mathbb{T})}|^2}{(\hat{\lambda}_\beta^{(\backslash \mathbb{T})} - \hat{\lambda}_\alpha^{(\backslash \mathbb{T})})^2 + \eta^2} = \frac{\eta |\hat{v}_{i\alpha}^{(\backslash \mathbb{T})}|^2}{\eta^2} + \sum_{\beta \neq \alpha} \frac{\eta |\hat{v}_{i\beta}^{(\backslash \mathbb{T})}|^2}{(\hat{\lambda}_\beta^{(\backslash \mathbb{T})} - \hat{\lambda}_\alpha^{(\backslash \mathbb{T})})^2 + \eta^2}$$

$$\geq \frac{|\hat{v}_{i\alpha}^{(\backslash \mathbb{T})}|^2}{\eta}.$$

Taking square roots and recalling that our choices of $\alpha \in \mathcal{S}$, $\mathbb{T} \subset [N]$, and $j \in [N]$ were arbitrary, we obtain

$$R \leq C\sqrt{\eta}. \tag{82}$$

Now, notice that the spectral decomposition of $\tilde{G}^{(\backslash \mathbb{T})}(z)$ is given by

$$\tilde{\mathbf{G}}^{(\backslash \mathbb{T})}(z) = \sum_{\alpha=1}^{N - |\mathbb{T}|} \frac{\hat{\mathbf{v}}_\alpha^{(\backslash \mathbb{T})}(\hat{\mathbf{v}}_\alpha^{(\backslash \mathbb{T})})^\top}{\hat{\lambda}_\alpha^{(\backslash \mathbb{T})} - z}.$$

Hence, applying the triangle inequality,

$$\left| \lambda_j (v_{ij}(\mathbf{v}_j)^{\backslash \mathbb{T}})^\top \tilde{\mathbf{G}}^{(\backslash \mathbb{T})}(z)(\mathbf{H}_{\cdot i})^{\backslash \mathbb{T}} \right| = \left| \lambda_j (v_{ij}(\mathbf{v}_j)^{\backslash \mathbb{T}})^\top \left( \sum_\alpha \frac{\hat{\mathbf{v}}_\alpha^{(\backslash \mathbb{T})}(\hat{\mathbf{v}}_\alpha^{(\backslash \mathbb{T})})^\top}{\hat{\lambda}_\alpha^{(\backslash \mathbb{T})} - z} \right) (\mathbf{H}_{\cdot i})^{\backslash \mathbb{T}} \right|$$

$$= \left| \sum_\alpha \frac{\lambda_j (v_{ij}(\mathbf{v}_j)^{\backslash \mathbb{T}})^\top \hat{\mathbf{v}}_\alpha^{(\backslash \mathbb{T})}(\hat{\mathbf{v}}_\alpha^{(\backslash \mathbb{T})})^\top (\mathbf{H}_{\cdot i})^{\backslash \mathbb{T}}}{\hat{\lambda}_\alpha^{(\backslash \mathbb{T})} - z} \right|$$

$$\leq \left| \frac{\lambda_j (v_{ij}(\mathbf{v}_j)^{\backslash \mathbb{T}})^\top \hat{\mathbf{v}}_{j-|\mathbb{T}|}^{(\backslash \mathbb{T})}(\hat{\mathbf{v}}_{j-|\mathbb{T}|}^{(\backslash \mathbb{T})})^\top (\mathbf{H}_{\cdot i})^{\backslash \mathbb{T}}}{\hat{\lambda}_{j-|\mathbb{T}|}^{(\backslash \mathbb{T})} - z} \right| \tag{83}$$

$$+ \left| \sum_{\alpha \neq j - |\mathbb{T}|} \frac{\lambda_j (v_{ij}(\mathbf{v}_j)^{\backslash \mathbb{T}})^\top \hat{\mathbf{v}}_\alpha^{(\backslash \mathbb{T})}(\hat{\mathbf{v}}_\alpha^{(\backslash \mathbb{T})})^\top (\mathbf{H}_{\cdot i})^{\backslash \mathbb{T}}}{\hat{\lambda}_\alpha^{(\backslash \mathbb{T})} - z} \right|.$$

Note that by Assumption ($\tilde{\mathbf{A}}$1) and the fact that the eigenvectors are normalized, we have

$$\left| (v_{ij}(\mathbf{v}_j)^{\backslash \mathbb{T}})^\top \hat{\mathbf{v}}_{j-|\mathbb{T}|}^{(\backslash \mathbb{T})} \right| = |v_{ij}| \left| ((\mathbf{v}_j)^{\backslash \mathbb{T}})^\top \hat{\mathbf{v}}_{j-|\mathbb{T}|}^{(\backslash \mathbb{T})} \right| \leq \frac{C(\log N)^\gamma}{\sqrt{N}} \left| ((\mathbf{v}_j)^{\backslash \mathbb{T}})^\top \hat{\mathbf{v}}_{j-|\mathbb{T}|}^{(\backslash \mathbb{T})} \right| \leq \frac{C(\log N)^\gamma}{\sqrt{N}}. \tag{84}$$

Applying submultiplicativity to the first term of the right-hand side of the inequality in Equation (83) and using Equation (84), we see that

$$\left| \frac{\lambda_j (v_{ij}(\mathbf{v}_j)^{\backslash \mathbb{T}})^\top \hat{\mathbf{v}}_{j-|\mathbb{T}|}^{(\backslash \mathbb{T})} (\hat{\mathbf{v}}_{j-|\mathbb{T}|}^{(\backslash \mathbb{T})})^\top (\mathbf{H}_{\cdot \mathbf{i}})^{\backslash \mathbb{T}}}{\hat{\lambda}_{j-|\mathbb{T}|}^{(\backslash \mathbb{T})} - z} \right| \leq \frac{C(\log N)^\gamma}{\sqrt{N}} \left| \frac{\lambda_j}{\hat{\lambda}_{j-|\mathbb{T}|}^{(\backslash \mathbb{T})} - z} \right| \left| (\hat{\mathbf{v}}_{j-|\mathbb{T}|}^{(\backslash \mathbb{T})})^\top (\mathbf{H}_{\cdot \mathbf{i}})^{\backslash \mathbb{T}} \right|. \tag{85}$$

Applying Lemma D.8 to control the inner product, we have with $(\xi, \nu)$-high-probability,

$$\left| \frac{\lambda_j (v_{ij}(\mathbf{v}_j)^{\backslash \mathbb{T}})^\top \hat{\mathbf{v}}_{j-|\mathbb{T}|}^{(\backslash \mathbb{T})} (\hat{\mathbf{v}}_{j-|\mathbb{T}|}^{(\backslash \mathbb{T})})^\top (\mathbf{H}_{\cdot \mathbf{i}})^{\backslash \mathbb{T}}}{\hat{\lambda}_{j-|\mathbb{T}|}^{(\backslash \mathbb{T})} - z} \right| \leq \frac{(\log N)^\gamma}{\sqrt{N}} \left| \frac{\lambda_j}{\hat{\lambda}_{j-|\mathbb{T}|}^{(\backslash \mathbb{T})} - z} \right| (C \log N)^\xi \left[ \frac{1}{q} + \frac{1}{\sqrt{N}} \right].$$

When $\lambda_j \geq \Sigma + 3$, which occurs due to Assumptions ($\tilde{\mathbf{A}}$2) and ($\tilde{\mathbf{A}}$4), we have $|\hat{\lambda}_{j-|\mathbb{T}|}^{(\mathbb{T})} - z| \geq c\lambda_j$, so with $(\xi, \nu)$-high-probability

$$\left| \frac{\lambda_j (v_{ij}(\mathbf{v}_j)^{\backslash \mathbb{T}})^\top \hat{\mathbf{v}}_{j-|\mathbb{T}|}^{(\backslash \mathbb{T})} (\hat{\mathbf{v}}_{j-|\mathbb{T}|}^{(\backslash \mathbb{T})})^\top (\mathbf{H}_{\cdot \mathbf{i}})^{\backslash \mathbb{T}}}{\hat{\lambda}_{j-|\mathbb{T}|}^{(\backslash \mathbb{T})} - z} \right| \leq \frac{C(\log N)^\gamma (\log N)^\xi}{\sqrt{N} q}. \tag{86}$$

The second term in Equation (83), consisting of eigenvalues associated with the bulk, is approached similarly.

Applying the triangle inequality,

$$\left| \sum_{\alpha \neq j-|\mathbb{T}|} \frac{\lambda_j (v_{ij}(\mathbf{v}_j)^{\backslash \mathbb{T}})^\top \hat{\mathbf{v}}_\alpha^{(\backslash \mathbb{T})} (\hat{\mathbf{v}}_\alpha^{(\backslash \mathbb{T})})^\top (\mathbf{H}_{\cdot \mathbf{i}})^{\backslash \mathbb{T}}}{\hat{\lambda}_\alpha^{(\backslash \mathbb{T})} - z} \right| \leq \lambda_j \sum_{\alpha \neq j-|\mathbb{T}|} \left| \frac{(v_{ij}(\mathbf{v}_j)^{\backslash \mathbb{T}})^\top \hat{\mathbf{v}}_\alpha^{(\backslash \mathbb{T})}}{\hat{\lambda}_\alpha^{(\backslash \mathbb{T})} - z} \right| |(\hat{\mathbf{v}}_\alpha^{(\backslash \mathbb{T})})^\top (\mathbf{H}_{\cdot \mathbf{i}})^{\backslash \mathbb{T}}|.$$

By Lemma D.8, we have with $(\xi, \nu)$-high-probability,

$$\left| \sum_{\alpha \neq j-|\mathbb{T}|} \frac{\lambda_j (v_{ij}(\mathbf{v}_j)^{\backslash \mathbb{T}})^\top \hat{\mathbf{v}}_\alpha^{(\backslash \mathbb{T})} (\hat{\mathbf{v}}_\alpha^{(\backslash \mathbb{T})})^\top (\mathbf{H}_{\cdot \mathbf{i}})^{\backslash \mathbb{T}}}{\hat{\lambda}_\alpha^{(\backslash \mathbb{T})} - z} \right| \leq \lambda_j \sum_{\alpha \neq j-|\mathbb{T}|} \left| \frac{(v_{ij}(\mathbf{v}_j)^{\backslash \mathbb{T}})^\top \hat{\mathbf{v}}_\alpha^{(\backslash \mathbb{T})}}{\hat{\lambda}_\alpha^{(\backslash \mathbb{T})} - z} \right| (\log N)^\xi \left[ \frac{C\sqrt{\eta}}{q} + \frac{1}{\sqrt{N}} \right]. \tag{87}$$

By the Cauchy-Schwarz inequality,

$$\sum_{\alpha \neq j-|\mathbb{T}|} \left| \frac{(v_{ij}(\mathbf{v}_j)^{\backslash \mathbb{T}})^\top \hat{\mathbf{v}}_\alpha^{(\backslash \mathbb{T})}}{\hat{\lambda}_\alpha^{(\backslash \mathbb{T})} - z} \right| \leq \left( \sum_{\alpha \neq j-|\mathbb{T}|} |(v_{ij}(\mathbf{v}_j)^{\backslash \mathbb{T}})^\top \hat{\mathbf{v}}_\alpha^{(\backslash \mathbb{T})}|^2 \right)^{1/2} \left( \sum_\alpha \frac{1}{|\hat{\lambda}_\alpha^{(\backslash \mathbb{T})} - z|^2} \right)^{1/2}. \tag{88}$$

By Lemma D.7, on the event $\tilde{\Omega}(\eta)$,

$$\sum_\alpha \frac{1}{|\hat{\lambda}_\alpha^{(\backslash \mathbb{T})} - z|^2} = \frac{1}{\eta} \operatorname{Im} \sum_\alpha \frac{1}{\hat{\lambda}_\alpha^{(\backslash \mathbb{T})} - z} = \frac{1}{\eta} \operatorname{Im} \operatorname{Tr} \tilde{\mathbf{G}}^{(\backslash \mathbb{T})}(z) = \frac{N}{\eta} \operatorname{Im} \tilde{m}(z) + O\left( \frac{1}{\eta^2} \right). \tag{89}$$

Moreover,

$$\sum_{\alpha \neq j-|\mathbb{T}|} |(v_{ij}(\mathbf{v}_j)^{\backslash \mathbb{T}})^\top \hat{\mathbf{v}}_\alpha^{(\backslash \mathbb{T})}|^2 \leq |v_{ij}|^2 \sum_{\alpha \neq j-|\mathbb{T}|} |((\mathbf{v}_j)^{\backslash \mathbb{T}})^\top \hat{\mathbf{v}}_\alpha^{(\backslash \mathbb{T})}|^2 \leq \frac{C(\log N)^{2\gamma}}{N} \left( \frac{r(\log N)^{2\gamma}}{\lambda_j} \right)^2$$

with $(\xi, \nu)$-high probability, where the last inequality follows from Lemma D.5 and Assumption ($\tilde{\mathbf{A}}$1) on the entries of $\mathbf{v}_j$. Taking square roots, we have with $(\xi, \nu)$-high probability,

$$\left( \sum_{\alpha \neq j-|\mathbb{T}|} |(v_{ij}(\mathbf{v}_j)^{\backslash \mathbb{T}})^\top \hat{\mathbf{v}}_\alpha^{(\backslash \mathbb{T})}|^2 \right)^{1/2} \leq \frac{C(\log N)^\gamma}{\sqrt{N}} \left( \frac{r(\log N)^{2\gamma}}{\lambda_j} \right). \tag{90}$$

Applying the bounds in Equations (89) and (90) to Equation (88),

$$\sum_{\alpha \neq j-|\mathbb{T}|} \left| \frac{(v_{ij}(\mathbf{v}_j)^{\backslash\mathbb{T}})^\top \hat{\mathbf{v}}_\alpha^{(\backslash\mathbb{T})}}{\hat{\lambda}_\alpha^{(\backslash\mathbb{T})} - z} \right| \leq \frac{C(\log N)^\gamma}{\sqrt{N}} \left( \frac{r(\log N)^{2\gamma}}{\lambda_j} \right) \sqrt{\frac{N}{\eta} \operatorname{Im} \tilde{m}(z) + O\left( \frac{1}{\eta^2} \right)}$$

with $(\xi, \nu)$-high probability on the event $\tilde{\Omega}(\eta)$. Applying this bound to Equation (87),

$$\left| \sum_{\alpha \neq j-|\mathbb{T}|} \frac{\lambda_j (v_{ij}(\mathbf{v}_j)^{\backslash\mathbb{T}})^\top \hat{\mathbf{v}}_\alpha^{(\backslash\mathbb{T})} (\hat{\mathbf{v}}_\alpha^{(\backslash\mathbb{T})})^\top (\mathbf{H}_{\cdot \mathbf{i}})^{\backslash\mathbb{T}}}{\hat{\lambda}_\alpha^{(\backslash\mathbb{T})} - z} \right|$$
$$\leq C \left( \frac{r(\log N)^{\xi+3\gamma}}{\sqrt{N}} \right) \left[ \frac{C\sqrt{\eta}}{q} + \frac{1}{\sqrt{N}} \right] \sqrt{\frac{N}{\eta} \operatorname{Im} \tilde{m}(z) + O\left( \frac{1}{\eta^2} \right)}$$

with $(\xi, \nu)$-high probability on the event $\tilde{\Omega}(\eta)$. Altogether, Equations (86) and (91) imply

$$\left| \frac{\lambda_j (v_{ij}(\mathbf{v}_j)^{\backslash\mathbb{T}})^\top \hat{\mathbf{v}}_{j-|\mathbb{T}|}^{(\backslash\mathbb{T})} (\hat{\mathbf{v}}_{j-|\mathbb{T}|}^{(\backslash\mathbb{T})})^\top (\mathbf{H}_{\cdot \mathbf{i}})^{\backslash\mathbb{T}}}{\hat{\lambda}_{j-|\mathbb{T}|}^{(\backslash\mathbb{T})} - z} \right| + \left| \sum_{\alpha \neq j-|\mathbb{T}|} \frac{\lambda_j (v_{ij}(\mathbf{v}_j)^{\backslash\mathbb{T}})^\top \hat{\mathbf{v}}_\alpha^{(\backslash\mathbb{T})} (\hat{\mathbf{v}}_\alpha^{(\backslash\mathbb{T})})^\top (\mathbf{H}_{\cdot \mathbf{i}})^{\backslash\mathbb{T}}}{\hat{\lambda}_\alpha^{(\backslash\mathbb{T})} - z} \right|$$
$$\leq \frac{C(\log N)^{\xi+\gamma}}{\sqrt{N}q} + C\left( \frac{r(\log N)^{\xi+3\gamma}}{\sqrt{N}} \right) \left[ \frac{C\sqrt{\eta}}{q} + \frac{1}{\sqrt{N}} \right] \left( \sqrt{\frac{N}{\eta} \operatorname{Im} \tilde{m}(z)} + \frac{1}{\eta} \right)$$
$$\leq Cr(\log N)^{\xi+3\gamma} \left( \frac{1}{q} + \sqrt{\frac{\operatorname{Im} \tilde{m}(z)}{N\eta}} + \frac{1}{N\eta} \right)$$

with $(\xi, \nu)$-high probability on the event $\tilde{\Omega}(\eta)$. Applying this to Equation (83),

$$\left| \lambda_j (v_{ij}(\mathbf{v}_j)^{\backslash\mathbb{T}})^\top \tilde{\mathbf{G}}^{(\backslash\mathbb{T})}(z) (\mathbf{H}_{\cdot \mathbf{i}})^{\backslash\mathbb{T}} \right| \leq Cr(\log N)^{\xi+3\gamma} \left( \frac{1}{q} + \sqrt{\frac{\operatorname{Im} \tilde{m}(z)}{N\eta}} + \frac{1}{N\eta} \right)$$

with $(\xi, \nu)$-high probability on the event $\tilde{\Omega}(\eta)$, completing the proof. $\qquad\square$

**Lemma D.10.** *Let $\tau < N$ be constant with respect to $N$ and let $z = E + \mathrm{i}\eta \in D_L$. We have, uniformly over $j \in \{N - r + 1, \ldots, N\}$ and $\mathbb{T} \subset \{1, \ldots, N\}$ satisfying $|\mathbb{T}| \leq \tau$, that with $(\xi, \nu)$-high probability,*

$$\left| \lambda_j^2 (v_{ij}(\mathbf{v}_j)^{\backslash\mathbb{T}})^\top \tilde{\mathbf{G}}^{(\backslash\mathbb{T})}(z) v_{ij}(\mathbf{v}_j)^{\backslash\mathbb{T}} \right| \leq \frac{C\lambda_j (\log N)^{2\gamma}}{N} + C\frac{r^2 (\log N)^{6\gamma}}{N\eta}. \tag{92}$$

*Proof.* Applying the triangle inequality to the spectral decomposition of $\tilde{\mathbf{G}}^{(\backslash\mathbb{T})}(z)$,

$$\left| \lambda_j^2 (v_{ij}(\mathbf{v}_j)^{\backslash\mathbb{T}})^\top \left( \sum_\alpha \frac{\hat{\mathbf{v}}_\alpha^{(\backslash\mathbb{T})} (\hat{\mathbf{v}}_\alpha^{(\backslash\mathbb{T})})^\top}{\hat{\lambda}_\alpha^{(\backslash\mathbb{T})} - z} \right) v_{ij}(\mathbf{v}_j)^{\backslash\mathbb{T}} \right| \leq \left| \frac{\lambda_j^2 (v_{ij}(\mathbf{v}_j)^{\backslash\mathbb{T}})^\top \hat{\mathbf{v}}_{j-|\mathbb{T}|}^{(\backslash\mathbb{T})} (\hat{\mathbf{v}}_{j-|\mathbb{T}|}^{(\backslash\mathbb{T})})^\top v_{ij}(\mathbf{v}_j)^{\backslash\mathbb{T}}}{\hat{\lambda}_{j-|\mathbb{T}|}^{(\backslash\mathbb{T})} - z} \right|$$
$$+ \left| \sum_{\alpha \neq j-|\mathbb{T}|} \frac{\lambda_j^2 (v_{ij}(\mathbf{v}_j)^{\backslash\mathbb{T}})^\top \hat{\mathbf{v}}_\alpha^{(\backslash\mathbb{T})} (\hat{\mathbf{v}}_\alpha^{(\backslash\mathbb{T})})^\top v_{ij}(\mathbf{v}_j)^{\backslash\mathbb{T}}}{\hat{\lambda}_\alpha^{(\backslash\mathbb{T})} - z} \right|. \tag{93}$$

Using similar arguments as in the proof of Lemma D.9 (e.g., as in Equation (85)), the first term of Equation (93) is controlled as

$$\left| \frac{\lambda_j^2 (v_{ij}(\mathbf{v}_j)^{\backslash\mathbb{T}})^\top \hat{\mathbf{v}}_{j-|\mathbb{T}|}^{(\backslash\mathbb{T})} (\hat{\mathbf{v}}_{j-|\mathbb{T}|}^{(\backslash\mathbb{T})})^\top v_{ij}(\mathbf{v}_j)^{\backslash\mathbb{T}}}{\hat{\lambda}_{j-|\mathbb{T}|}^{(\backslash\mathbb{T})} - z} \right| \leq \frac{C\lambda_j^2}{\lambda_j} \frac{(\log N)^{2\gamma}}{N} = \frac{C\lambda_j (\log N)^{2\gamma}}{N} \tag{94}$$

with $(\xi, \nu)$-high probability. Turning our attention to the second term on the right-hand side of Equation (93), noticing that $|\hat{\lambda}_{j-|\mathbb{T}|}^{(\backslash \mathbb{T})} - z| \geq \eta$ since $\hat{\lambda}_{j-|\mathbb{T}|}^{(\backslash \mathbb{T})}$ is real,

$$\left| \sum_{\alpha \neq j - |\mathbb{T}|} \frac{\lambda_j^2 (v_{ij}(\mathbf{v}_j)^{\backslash \mathbb{T}})^\top \hat{\mathbf{v}}_\alpha^{(\backslash \mathbb{T})} (\hat{\mathbf{v}}_\alpha^{(\backslash \mathbb{T})})^\top v_{ij}(\mathbf{v}_j)^{\backslash \mathbb{T}}}{\hat{\lambda}_\alpha^{(\backslash \mathbb{T})} - z} \right| \leq \left( \frac{C\lambda_j^2 (\log N)^{2\gamma}}{N\eta} \right) \sum_{\alpha \neq j - |\mathbb{T}|} \left| ((\mathbf{v}_j)^{\backslash \mathbb{T}})^\top \hat{\mathbf{v}}_{j-|\mathbb{T}|}^{(\backslash \mathbb{T})} \right|^2$$
$$\leq C\frac{r^2(\log N)^{6\gamma}}{N\eta}, \tag{95}$$

where the last inequality holds with $(\xi, \nu)$-high probability by Lemma D.5. Applying Equations (94) and (95) to the right-hand side of Equation (93), we obtain

$$\left| \lambda_j^2 (v_{ij}(\mathbf{v}_j)^{\backslash \mathbb{T}})^\top \tilde{\mathbf{G}}^{(\backslash \mathbb{T})}(z) v_{ij}(\mathbf{v}_j)^{\backslash \mathbb{T}} \right| \leq \frac{C\lambda_j(\log N)^{2\gamma}}{N} + C\frac{r^2(\log N)^{6\gamma}}{N\eta}$$

with $(\xi, \nu)$-high probability. The choice of $j \in [N]$ was arbitrary, so this bound holds for all $j \in [N]$ uniformly with $(\xi, \nu)$-high probability, as we set out to show. $\square$

The following lemma has no analogue in Erdős et al. (2013a), as it is required only when we consider a signal matrix of rank $r > 1$.

**Lemma D.11.** *Let* $\mathbf{B}$ *satisfy Equation* (36)*, and let* $\tau < N$ *be constant with respect to* $N$ *Let* $z = E + \mathrm{i}\eta \in D_L$*, we have uniformly for any* $\mathbb{T} \subset \{1, \ldots, N\}$ *satisfying* $|\mathbb{T}| \leq \tau < N$*,* $\ell_1, \ell_2 \in \{N - r + 1, \ldots, N\}$ *and* $\ell_1 \neq \ell_2$*, and for possibly* $i = j \in [N] \setminus \mathbb{T}$*, that with* $(\xi, \nu)$*-high probability,*

$$\left| \lambda_{\ell_2}\lambda_{\ell_1}(v_{i\ell_1}(\mathbf{v}_{\ell_1})^{\backslash \mathbb{T}})^\top \tilde{\mathbf{G}}^{(\backslash \mathbb{T})}(z) v_{j\ell_2}(\mathbf{v}_{\ell_2})^{\backslash \mathbb{T}} \right| \leq Cr^2 \frac{(\log N)^{6\gamma}}{N\eta}. \tag{96}$$

*Proof.* We follow an argument similar to that used in Lemmas D.9 and D.10. Let $\mathbb{T} \subset [N]$ and $\ell_1, \ell_2 \in \{N - r + 1, \ldots, N\}$ distinct. By the spectral decomposition of $\tilde{\mathbf{G}}^{(\backslash \mathbb{T})}(z)$ and Lemma C.2, we have

$$\left| \lambda_{\ell_2}\lambda_{\ell_1}(v_{i\ell_1}(\mathbf{v}_{\ell_1})^{\backslash \mathbb{T}})^\top \tilde{\mathbf{G}}^{(\backslash \mathbb{T})}(z) v_{j\ell_2}(\mathbf{v}_{\ell_2})^{\backslash \mathbb{T}} \right| \leq C\lambda_N^2 \left| (v_{i\ell_1}(\mathbf{v}_{\ell_1})^{\backslash \mathbb{T}})^\top \left( \sum_\alpha \frac{\hat{\mathbf{v}}_\alpha^{(\backslash \mathbb{T})}(\hat{\mathbf{v}}_\alpha^{(\backslash \mathbb{T})})^\top}{\hat{\lambda}_\alpha^{(\backslash \mathbb{T})} - z} \right) v_{j\ell_2}(\mathbf{v}_{\ell_2})^{\backslash \mathbb{T}} \right|$$
$$\leq C\lambda_N^2 |v_{i\ell_1}| |v_{j\ell_2}| \left| ((\mathbf{v}_{\ell_1})^{\backslash \mathbb{T}})^\top \left( \sum_\alpha \frac{\hat{\mathbf{v}}_\alpha^{(\backslash \mathbb{T})}(\hat{\mathbf{v}}_\alpha^{(\backslash \mathbb{T})})^\top}{\hat{\lambda}_\alpha^{(\backslash \mathbb{T})} - z} \right) (\mathbf{v}_{\ell_2})^{\backslash \mathbb{T}} \right|. \tag{97}$$

with $(\xi, \nu)$-high probability. Recalling our assumption on the entries of $\mathbf{v}_k$, for $k \in \{N - r + 1, \ldots, N\}$ from Assumption (Ã1), we have $N |v_{i\ell_1}| |v_{j\ell_2}| \leq C(\log N)^{2\gamma}$. Applying this inequality to Equation (97) yields

$$\left| \lambda_{\ell_2}\lambda_{\ell_1}(v_{i\ell_1}(\mathbf{v}_{\ell_1})^{\backslash \mathbb{T}})^\top \tilde{\mathbf{G}}^{(\backslash \mathbb{T})}(z) v_{j\ell_2}(\mathbf{v}_{\ell_2})^{\backslash \mathbb{T}} \right| \leq \frac{C\lambda_N^2(\log N)^{2\gamma}}{N} \left| ((\mathbf{v}_{\ell_1})^{\backslash \mathbb{T}})^\top \left( \sum_\alpha \frac{\hat{\mathbf{v}}_\alpha^{(\backslash \mathbb{T})}(\hat{\mathbf{v}}_\alpha^{(\backslash \mathbb{T})})^\top}{\hat{\lambda}_\alpha^{(\backslash \mathbb{T})} - z} \right) (\mathbf{v}_{\ell_2})^{\backslash \mathbb{T}} \right|$$

with $(\xi, \nu)$-high probability. Distributing and applying the triangle inequality,

$$\left| \lambda_{\ell_2}\lambda_{\ell_1}(v_{i\ell_1}(\mathbf{v}_{\ell_1})^{\backslash \mathbb{T}})^\top \tilde{\mathbf{G}}^{(\backslash \mathbb{T})}(z) v_{j\ell_2}(\mathbf{v}_{\ell_2})^{\backslash \mathbb{T}} \right|$$
$$\leq \frac{C\lambda_N^2(\log N)^{2\gamma}}{N} \left( \left| \frac{((\mathbf{v}_{\ell_1})^{\backslash \mathbb{T}})^\top \hat{\mathbf{v}}_{\ell_1-|\mathbb{T}|}^{(\backslash \mathbb{T})}(\hat{\mathbf{v}}_{\ell_1-|\mathbb{T}|}^{(\backslash \mathbb{T})})^\top (\mathbf{v}_{\ell_2})^{\backslash \mathbb{T}}}{\hat{\lambda}_{\ell_1-|\mathbb{T}|}^{(\mathbb{T})} - z} \right| + \left| \frac{((\mathbf{v}_{\ell_1})^{\backslash \mathbb{T}})^\top \hat{\mathbf{v}}_{\ell_2-|\mathbb{T}|}^{(\backslash \mathbb{T})}(\hat{\mathbf{v}}_{\ell_2-|\mathbb{T}|}^{(\backslash \mathbb{T})})^\top (\mathbf{v}_{\ell_2})^{\backslash \mathbb{T}}}{\hat{\lambda}_{\ell_2-|\mathbb{T}|}^{(\mathbb{T})} - z} \right| \right.$$
$$\left. + \sum_{\alpha \notin \mathcal{S}} \left| \frac{((\mathbf{v}_{\ell_1})^{\backslash \mathbb{T}})^\top \hat{\mathbf{v}}_\alpha^{(\backslash \mathbb{T})}(\hat{\mathbf{v}}_\alpha^{(\backslash \mathbb{T})})^\top (\mathbf{v}_{\ell_2})^{\backslash \mathbb{T}}}{\hat{\lambda}_\alpha^{(\backslash \mathbb{T})} - z} \right| \right) \tag{98}$$

where $\mathcal{S} = \{\ell_1 - |\mathbb{T}|, \ell_2 - |\mathbb{T}|\}$.

The first two terms of Equation (98) are controlled similarly. We will examine the first term as an example. Using the fact that both $\mathbf{v}_{\ell_1}$ and $\hat{\mathbf{v}}_{\ell_1-|\mathbb{T}|}^{(\backslash\mathbb{T})}$ are normalized eigenvectors, we have $\left|((\mathbf{v}_{\ell_1})^{\backslash\mathbb{T}})^\top \hat{\mathbf{v}}_{\ell_1-|\mathbb{T}|}^{(\backslash\mathbb{T})}\right| \leq 1$. Moreover, $|\hat{\lambda}_{j-|\mathbb{T}|}^{(\mathbb{T})} - z| \geq c\lambda_j$ by Assumption ($\tilde{\mathbf{A}}$4). These two bounds together imply that with $(\xi, \nu)$-high probability,

$$\left|\frac{((\mathbf{v}_{\ell_1})^{\backslash\mathbb{T}})^\top \hat{\mathbf{v}}_{\ell_1-|\mathbb{T}|}^{(\backslash\mathbb{T})}(\hat{\mathbf{v}}_{\ell_1-|\mathbb{T}|}^{(\backslash\mathbb{T})})^\top (\mathbf{v}_{\ell_2})^{\backslash\mathbb{T}}}{\hat{\lambda}_{\ell_1-|\mathbb{T}|}^{(\mathbb{T})} - z}\right| \leq \frac{C}{\lambda_{\ell_1}}\left|(\hat{\mathbf{v}}_{\ell_1-|\mathbb{T}|}^{(\backslash\mathbb{T})})^\top(\mathbf{v}_{\ell_2})^{\backslash\mathbb{T}}\right| \leq \frac{C}{\lambda_{\ell_1}}\frac{r(\log N)^{2\gamma}}{\lambda_{\ell_2}}, \tag{99}$$

where the second inequality follows from Lemma D.5 and Assumption ($\tilde{\mathbf{A}}$1). Using the fact that $\lambda_{\ell_1} \geq \lambda_{N-r+1}$ and substituting Equation (99) into (98) yields that with $(\xi, \nu)$-high probability,

$$\left|\lambda_{\ell_2}\lambda_{\ell_1}(v_{i\ell_1}(\mathbf{v}_{\ell_1})^{\backslash\mathbb{T}})^\top \tilde{\mathbf{G}}^{(\backslash\mathbb{T})}(z)v_{j\ell_2}(\mathbf{v}_{\ell_2})^{\backslash\mathbb{T}}\right|$$
$$\leq \frac{C\lambda_N^2(\log N)^{2\gamma}}{N}\left(\frac{Cr(\log N)^{2\gamma}}{\lambda_{N-r+1}^2} + \sum_{\alpha\notin\mathcal{S}}\left|\frac{((\mathbf{v}_{\ell_1})^{\backslash\mathbb{T}})^\top \hat{\mathbf{v}}_\alpha^{(\backslash\mathbb{T})}(\hat{\mathbf{v}}_\alpha^{(\backslash\mathbb{T})})^\top(\mathbf{v}_{\ell_2})^{\backslash\mathbb{T}}}{\hat{\lambda}_\alpha^{(\backslash\mathbb{T})} - z}\right|\right). \tag{100}$$

Since $\left|\hat{\lambda}_\alpha^{(\backslash\mathbb{T})} - z\right| \geq \eta$, the triangle inequality yields

$$\sum_{\alpha\notin\mathcal{S}}\left|\frac{((\mathbf{v}_{\ell_1})^{\backslash\mathbb{T}})^\top\hat{\mathbf{v}}_\alpha^{(\backslash\mathbb{T})}(\hat{\mathbf{v}}_\alpha^{(\backslash\mathbb{T})})^\top(\mathbf{v}_{\ell_2})^{\backslash\mathbb{T}}}{\hat{\lambda}_\alpha^{(\backslash\mathbb{T})} - z}\right| \leq \frac{1}{\eta}\sum_{\alpha\notin\mathcal{S}}|((\mathbf{v}_{\ell_1})^{\backslash\mathbb{T}})^\top\hat{\mathbf{v}}_\alpha^{(\backslash\mathbb{T})}||(\hat{\mathbf{v}}_\alpha^{(\backslash\mathbb{T})})^\top(\mathbf{v}_{\ell_2})^{\backslash\mathbb{T}}|$$
$$\leq \frac{1}{\eta}\left(\sum_{\alpha\notin\mathcal{S}}|((\mathbf{v}_{\ell_1})^{\backslash\mathbb{T}})^\top\hat{\mathbf{v}}_\alpha^{(\backslash\mathbb{T})}|^2\right)^{1/2}\left(\sum_{\alpha\notin\mathcal{S}}|(\hat{\mathbf{v}}_\alpha^{(\backslash\mathbb{T})})^\top(\mathbf{v}_{\ell_2})^{\backslash\mathbb{T}}|^2\right)^{1/2}$$
$$\leq \frac{C}{\eta}\left(\frac{r(\log N)^{2\gamma}}{\lambda_{N-r+1}}\right)^2,$$

where the second inequality follows from the Cauchy-Schwarz inequality and the last inequality holds with $(\xi, \nu)$-high probability by Lemma D.5. Applying this to Equation (100) and using Assumption $\tilde{\mathbf{A}}$4, we obtain

$$\left|\lambda_{\ell_2}\lambda_{\ell_1}(v_{i\ell_1}(\mathbf{v}_{\ell_1})^{\backslash\mathbb{T}})^\top\tilde{\mathbf{G}}^{(\backslash\mathbb{T})}(z)v_{j\ell_2}(\mathbf{v}_{\ell_2})^{\backslash\mathbb{T}}\right| \leq Cr^2\frac{(\log N)^{6\gamma}}{N\eta},$$

with $(\xi, \nu)$-high probability. The choice of $\ell_1, \ell_2 \in \{N-r+1, \ldots, N\}$ and $\mathbb{T} \subset [N]$ were arbitrary, so this bound holds for any choice of $\ell_1, \ell_2 \in \{N-r+1, \ldots, N\}$ and $\mathbb{T}$ such that $|\mathbb{T}| \leq \tau$ with $(\xi, \nu)$-high probability, completing the proof. $\qquad\square$

Following Erdős et al. (2013a), we will introduce a control parameter,

$$\Phi(z) := \frac{r^2(\log N)^{\xi+3\gamma}}{q} + r^2(\log N)^{2\xi+6\gamma}\left(\sqrt{\frac{\operatorname{Im}\tilde{m}(z)}{N\eta}} + \frac{r^2}{N\eta}\right) \tag{101}$$

where $q$ satisfies Equation (34).

**Lemma D.12** (Erdős et al. (2013a) Proposition 7.6, adapted)**.** *Let* $\mathbf{B}$ *satisfy Equation* (36)*. For* $z = E + \mathrm{i}\eta \in D_L$*, we have that on on* $\tilde{\Omega}(\eta)$ *(see Equation* (74)*) with* $(\xi, \nu)$*-high probability,*

$$\tilde{\Lambda}_o(z) \leq C\Phi(z), \tag{102}$$

$$\max_{i\in[N]}|\tilde{Z}_i(z)| \leq C\Phi(z) \tag{103}$$

*and*

$$\max_{i\in[N]}|\tilde{\mathcal{A}}_i(z)| \leq \frac{Cr(\log N)^{2\gamma}}{\sqrt{N}} + \frac{Cr^4(\log N)^{6\gamma}}{N\eta}, \tag{104}$$

*where* $\tilde{Z}$ *and* $\tilde{\mathcal{A}}$ *are as in Equations* (64) *and* (66)*, respectively, and* $\tilde{\Lambda}_o(z)$ *is as in Equation* (73)*.*

*Proof.* Starting with the estimate for $\tilde{\Lambda}_o(z)$, we see that from Lemma D.18, we have

$$\tilde{G}_{ij}(z) = \tilde{G}_{ii}(z)\tilde{G}_{jj}^{(\backslash i)}(z)\left(((\mathbf{B}_{\cdot i})^{\backslash ij})^{\top}\tilde{\mathbf{G}}^{(\backslash ij)}(z)\,(\mathbf{B}_{\cdot j})^{\backslash ij}\right) - \tilde{G}_{ii}(z)\tilde{G}_{jj}^{(\backslash i)}(z)B_{ij}.$$

By Lemma D.22, we have $\left|\tilde{G}_{ii}(z)\right| \leq C$ and $|\tilde{G}_{jj}^{(\backslash i)}(z)| \leq C$ for all $i,j \in [N]$. Therefore,

$$\left|\tilde{G}_{ij}(z)\right| \leq C\left|((\mathbf{B}_{\cdot i})^{\backslash ij})^{\top}\tilde{\mathbf{G}}^{(\backslash ij)}(z)\,(\mathbf{B}_{\cdot j})^{\backslash ij}\right| + C\,|B_{ij}| = C\left|\sum_{k,l}^{(\backslash ij)} B_{ik}\tilde{G}_{kl}^{(\backslash ij)}(z)B_{lj}\right| + C|B_{ij}|. \tag{105}$$

By Lemma D.24, we have that with $(\xi,\nu)$-high probability, it holds uniformly over for all $i,j \in [N]$ that

$$|B_{ij}| \leq C\frac{r\rho_N(\log N)^{2\gamma}}{\sqrt{N}} + \frac{C}{q}.$$

Substituting this into Equation (105), we have

$$|\tilde{G}_{ij}(z)| \leq C\left|\sum_{k,l}^{(\backslash ij)} B_{ik}\tilde{G}_{kl}^{(\backslash ij)}(z)B_{lj}\right| + C\frac{r\rho_N(\log N)^{2\gamma}}{\sqrt{N}} + \frac{C}{q}. \tag{106}$$

Recalling the definition of $\mathbf{B}$ from Equation (36),

$$\left|\sum_{k,l}^{(\backslash ij)} B_{ik}\tilde{G}_{kl}^{(\backslash ij)}(z)B_{lj}\right| = \left|\sum_{k,l}^{(\backslash ij)}\left(h_{ik} + \sum_{\alpha=N-r+1}^{N}\lambda_\alpha v_{i\alpha}v_{k\alpha}\right)\tilde{G}_{kl}^{(\backslash ij)}(z)\left(h_{lj} + \sum_{\alpha=N-r+1}^{N}\lambda_\alpha v_{l\alpha}v_{j\alpha}\right)\right|. \tag{107}$$

After applying triangle inequality to the right-hand side of Equation (107), there will be a term controllable by Lemma D.21 according to

$$\left|\sum_{k,l}^{(\backslash ij)} h_{ik}\tilde{G}_{kl}^{(\backslash ij)}(z)h_{lj}\right| \leq \frac{C(\log N)^{\xi}}{q} + C(\log N)^{2\xi}\left(\sqrt{\frac{\operatorname{Im}\tilde{m}(z)}{N\eta}} + \frac{1}{N\eta}\right). \tag{108}$$

on the event $\tilde{\Omega}(\eta)$ with $(\xi,\nu)$-high probability.

After applying triangle inequality in Equation (107), there will also be $2r$ terms that can be controlled by Lemma D.9 as, for $\ell = i, j$,

$$\left|\lambda_{\alpha_1}(v_{i\alpha_1}(\mathbf{v}_{\alpha_1})^{\backslash ij})^{\top}\tilde{\mathbf{G}}^{(\backslash ij)}(z)\,(\mathbf{H}_{\cdot\ell})^{\backslash ij}\right| \leq Cr(\log N)^{\xi+3\gamma}\left(\frac{1}{q} + \sqrt{\frac{\operatorname{Im}\tilde{m}(z)}{N\eta}} + \frac{1}{N\eta}\right) \tag{109}$$

with $(\xi,\nu)$-high probability. By comparing the quantity on the right-hand side of Equation (109) to the definition of $\Phi(z)$ in Equation (101), we see that on $\tilde{\Omega}(\eta)$ with $(\xi,\nu)$-high probability,

$$\left|\lambda_{\alpha_1}(v_{i\alpha_1}(\mathbf{v}_{\alpha_1})^{\backslash ij})^{\top}\tilde{\mathbf{G}}^{(\backslash ij)}(z)\,(\mathbf{H}_{\cdot\ell})^{\backslash ij}\right| \leq Cr(\log N)^{\xi+3\gamma}\left(\frac{1}{q} + \sqrt{\frac{\operatorname{Im}\tilde{m}(z)}{N\eta}} + \frac{1}{N\eta}\right) \leq C\Phi(z). \tag{110}$$

There will be $r$ terms in Equation (107) controllable by Lemma D.10 according to

$$\left|\lambda_{\alpha_1}^2(v_{i\alpha_1}(\mathbf{v}_{\alpha_1})^{\backslash ij})^{\top}\tilde{\mathbf{G}}^{(\backslash ij)}(z)v_{j\alpha_1}(\mathbf{v}_{\alpha_1})^{\backslash ij}\right| \leq \left(\frac{C\lambda_{\alpha_1}(\log N)^{2\gamma}}{N} + C\frac{r^2(\log N)^{6\gamma}}{N\eta}\right) \tag{111}$$

with $(\xi,\nu)$-high probability.

There will be $\binom{r}{2}$ terms in Equation (107) controllable by Lemma D.11 as

$$\left|\lambda_{\alpha_2}\lambda_{\alpha_1}(v_{i\alpha_1}(\mathbf{v}_{\alpha_1})^{\backslash ij})^{\top}\tilde{\mathbf{G}}^{(\backslash ij)}(z)v_{j\alpha_2}(\mathbf{v}_{\alpha_2})^{\backslash ij}\right| \leq C\frac{r^2(\log N)^{6\gamma}}{N\eta} \tag{112}$$

with $(\xi, \nu)$-high probability. The upper bounds provided by Lemmas D.10 and D.11 are dominated by $\Phi(z)$ since for any $j$, looking at Equation (101),

$$C\frac{r^2(\log N)^{6\gamma}}{N\eta} \leq C\frac{r^2(\log N)^{2\xi+6\gamma}}{N\eta} \leq \frac{(\log N)^{\xi+3\gamma}}{q} + (\log N)^{2\xi+6\gamma}\left(\sqrt{\frac{\operatorname{Im}\tilde{m}(z)}{N\eta}} + \frac{1}{N\eta}\right) \leq \frac{C\Phi(z)}{r^2}$$

and

$$\frac{C\lambda_{\alpha_1}(\log N)^{2\gamma}}{N} \leq \frac{C(\log N)^{2\gamma}}{\sqrt{N}} \leq \frac{r(\log N)^{\xi+3\gamma}}{q} + r(\log N)^{2\xi+6\gamma}\left(\sqrt{\frac{\operatorname{Im}\tilde{m}(z)}{N\eta}} + \frac{r}{N\eta}\right) \leq \frac{C\Phi(z)}{r}.$$

Applying the above two bounds to Equations (111) and (112),

$$r\left|\lambda_{\alpha_1}^2(v_{i\alpha_1}(\mathbf{v}_{\alpha_1})^{\backslash ij})^\top\tilde{\mathbf{G}}^{(\backslash ij)}(z)v_{j\alpha_1}(\mathbf{v}_{\alpha_1})^{\backslash ij}\right| \leq C\Phi(z) \tag{113}$$

and

$$\binom{r}{2}\left|\lambda_{\alpha_2}\lambda_{\alpha_1}(v_{i\alpha_1}(\mathbf{v}_{\alpha_1})^{\backslash ij})^\top\tilde{\mathbf{G}}^{(\backslash ij)}(z)v_{j\alpha_2}(\mathbf{v}_{\alpha_2})^{\backslash ij}\right| \leq C\Phi(z). \tag{114}$$

Applying Equations (108), (110), (113), and (114) to (106), and taking the maximum over $i \neq j$, we have that on the event $\tilde{\Omega}(\eta)$ with $(\xi, \nu)$-high-probability,

$$\tilde{\Lambda}_o(z) := \max_{i\neq j}|\tilde{G}_{ij}(z)| \leq C\Phi(z),$$

which establishes Equation (102). Note that to accommodate the union bound over $i, j \in [N]$, we increase the value of $\nu$ in the definition of $(\xi, \nu)$-high-probability by a constant factor.

Recalling the definition of $\tilde{Z}_i(z)$ from Equation (64),

$$\tilde{Z}_i(z) = Q_i\left(((\mathbf{B}_{.i})^{\backslash i})^\top\tilde{\mathbf{G}}^{(\backslash i)}(z)(\mathbf{B}_{.i})^{\backslash i}\right) = ((\mathbf{B}_{.i})^{\backslash i})^\top\tilde{\mathbf{G}}^{(\backslash i)}(z)(\mathbf{B}_{.i})^{\backslash i} - \mathbb{E}\left[((\mathbf{B}_{.i})^{\backslash i})^\top\tilde{\mathbf{G}}^{(\backslash i)}(z)(\mathbf{B}_{.i})^{\backslash i} \mid (\mathbf{H})^{\backslash i}\right],$$

where we have used the definition of $Q_i(\cdot)$ from Equation (65). Adding and subtracting appropriate quantities and using properties of the conditional expectation,

$$\tilde{Z}_i(z) = \sum_{\alpha=N-r+1}((\mathbf{H}_{.i})^{\backslash i})^\top\tilde{\mathbf{G}}^{(\backslash i)}(z)\lambda_\alpha v_{i\alpha}(\mathbf{v}_\alpha)^{\backslash i} + \sum_{\alpha=N-r+1}\lambda_w(v_{i\alpha}(\mathbf{v}_\alpha)^{\backslash i})^\top\tilde{\mathbf{G}}^{(\backslash i)}(z)(\mathbf{H}_{.i})^{\backslash i}$$
$$+ ((\mathbf{H}_{.i})^{\backslash i})^\top\tilde{\mathbf{G}}^{(\backslash i)}(z)(\mathbf{H}_{.i})^{\backslash i} - \mathbb{E}\left[((\mathbf{H}_{.i})^{\backslash i})^\top\tilde{\mathbf{G}}^{(\backslash i)}(z)(\mathbf{H}_{.i})^{\backslash i} \mid (\mathbf{H})^{\backslash i}\right] \tag{115}$$

The first two terms are controlled using Lemma D.9:

$$\left|\sum_{\alpha=N-r+1}((\mathbf{H}_{.i})^{\backslash i})^\top\tilde{\mathbf{G}}^{(\backslash i)}(z)\lambda_\alpha v_{i\alpha}(\mathbf{v}_\alpha)^{\backslash i} + \sum_{\alpha=N-r+1}\lambda_w(v_{i\alpha}(\mathbf{v}_\alpha)^{\backslash i})^\top\tilde{\mathbf{G}}^{(\backslash i)}(z)(\mathbf{H}_{.i})^{\backslash i}\right|$$
$$\leq 2Cr(\log N)^{\xi+3\gamma}\left(\frac{1}{q} + \sqrt{\frac{\operatorname{Im}\tilde{m}(z)}{N\eta}} + \frac{1}{N\eta}\right). \tag{116}$$

To control the latter two terms in Equation (115), we follow the argument first given in Lemma 3.13 from (Erdős et al., 2013a). We have from Lemma D.8, with $(\xi, \nu)$-high probability:

$$\left|((\mathbf{H}_{.i})^{\backslash i})^\top\tilde{\mathbf{G}}^{(\backslash i)}(z)(\mathbf{H}_{.i})^{\backslash i} - \mathbb{E}\left[((\mathbf{H}_{.i})^{\backslash i})^\top\tilde{\mathbf{G}}^{(\backslash i)}(z)(\mathbf{H}_{.i})^{\backslash i} \mid (\mathbf{H})^{\backslash i}\right]\right|$$
$$\leq \left|\sum_{k\neq l}h_{kl}\tilde{G}_{kl}^{(\backslash i)}(z)h_{lk}\right| + \left|\sum_k^{(\backslash i)}|h_{kk}|^2\,\tilde{G}_{kk}^{(\backslash i)}(z) - \frac{1}{N}\tilde{G}_{kk}^{(\backslash i)}(z)\right|, \tag{117}$$

where we have used the fact that

$$\mathbb{E}\left[((\mathbf{H}_{.i})^{\backslash i})^\top\tilde{\mathbf{G}}^{(\backslash i)}(z)(\mathbf{H}_{.i})^{\backslash i} \mid (\mathbf{H})^{\backslash i}\right] = \mathbb{E}\left[\sum_{k,l\neq i}h_{ki}\tilde{G}_{kl}^{(\backslash i)}(z)h_{il} \,\Big|\, (\mathbf{H})^{\backslash i}\right] = \sum_k^{(\backslash i)}\frac{1}{N}\tilde{G}_{kk}(z).$$

The right-hand side of Equation (117) is estimated using Lemma D.8 for both terms, yielding

$$\left| ((\mathbf{H}_{.i})^{\backslash i})^{\top} \tilde{\mathbf{G}}^{(\backslash i)}(z)(\mathbf{H}_{.i})^{\backslash i} - \mathbb{E}\left[ ((\mathbf{H}_{.i})^{\backslash i})^{\top} \tilde{G}^{(\backslash i)}(z)(\mathbf{H}_{.i})^{\backslash i} \mid (\mathbf{H})^{\backslash i} \right] \right|$$

$$\leq (\log N)^{2\xi} \left[ \frac{\max_{k \neq l} \left| \tilde{G}_{kl}^{(\backslash i)}(z) \right|}{q} + \left( \frac{1}{N^2} \sum_{k \neq l} \left| \tilde{G}_{kl}^{(\backslash i)}(z) \right|^2 \right)^{1/2} \right] + (\log N)^{\xi} \frac{\max_k \left| \tilde{G}_{kk}^{(\backslash i)}(z) \right|}{q}.$$

with $(\xi, \nu)$-high probability. By Lemma D.22, on the event $\tilde{\Omega}(\eta)$, the $\tilde{G}_{kk}^{(\backslash i)}(z)$ and $\tilde{G}_{kl}^{(\backslash i)}(z)$ terms are controlled by constants, whence

$$\left| ((\mathbf{H}_{.i})^{\backslash i})^{\top} \tilde{\mathbf{G}}^{(\backslash i)}(z)(\mathbf{H}_{.i})^{\backslash i} - \mathbb{E}\left[ ((\mathbf{H}_{.i})^{\backslash i})^{\top} \tilde{G}^{(\backslash i)}(z)(\mathbf{H}_{.i})^{\backslash i} \mid (\mathbf{H})^{\backslash i} \right] \right|$$

$$\leq (\log N)^{2\xi} \left[ \frac{C(\log N)^{-\xi}}{q} + \left( \frac{1}{N^2} \sum_{k \neq l} \left| \tilde{G}_{kl}^{(\backslash i)}(z) \right|^2 \right)^{1/2} \right] + (\log N)^{\xi} \frac{C}{q} \tag{118}$$

on the event $\tilde{\Omega}(\eta)$ with $(\xi, \nu)$-high probability. From Ward's Identity (see Equation (3.6) in Benaych-Georges & Knowles, 2018), we have

$$\sum_{j=1}^{N} |\tilde{G}_{ij}(z)|^2 = \frac{\operatorname{Im} \tilde{G}_{ii}(z)}{\eta}. \tag{119}$$

So then on the event $\tilde{\Omega}(\eta)$,

$$\frac{1}{N^2} \sum_{k,l}^{(\backslash i)} \left| \tilde{G}_{kl}^{(\backslash i)}(z) \right|^2 = \frac{1}{N^2 \eta} \sum_{k}^{(\backslash i)} \operatorname{Im} \tilde{G}_{kk}^{(\backslash i)}(z) \leq \frac{C}{N\eta} \left( \operatorname{Im} \tilde{m}(z) + \tilde{\Lambda}_o^2(z) \right), \tag{120}$$

where in the inequality we have used the fact that $\tilde{G}_{kk}^{(\backslash i)}(z) = \tilde{G}_{kk}(z) + C\tilde{\Lambda}_o^2(z)$ on $\tilde{\Omega}(\eta)$ by Lemma D.17 and the fact that $\tilde{m}(z) = \frac{1}{N} \operatorname{Tr} \tilde{\mathbf{G}}(z)$. Equation (120) then implies that on $\tilde{\Omega}(\eta)$ with $(\xi, \nu)$-high probability,

$$\left( \frac{1}{N^2} \sum_{k,l}^{(\backslash i)} \left| \tilde{G}_{kl}^{(\backslash i)}(z) \right|^2 \right)^{1/2} \leq C \left( \sqrt{\frac{\operatorname{Im} \tilde{m}(z)}{N\eta}} + \frac{\tilde{\Lambda}_o}{\sqrt{N\eta}} \right) \leq C \left( \sqrt{\frac{\operatorname{Im} \tilde{m}(z)}{N\eta}} + \frac{\Phi(z)}{\sqrt{N\eta}} \right), \tag{121}$$

where the second inequality follows from Equation (102). Substituting Equation (121) into Equation (118) and collecting terms, we have

$$\left| ((\mathbf{H}_{.i})^{\backslash i})^{\top} \tilde{\mathbf{G}}^{(\backslash i)}(z)(\mathbf{H}_{.i})^{\backslash i} - \mathbb{E}\left[ ((\mathbf{H}_{.i})^{\backslash i})^{\top} \tilde{G}^{(\backslash i)}(z)(\mathbf{H}_{.i})^{\backslash i} \mid (\mathbf{H})^{\backslash i} \right] \right|$$

$$\leq C \left( \frac{(\log N)^{\xi}}{q} + (\log N)^{2\xi} \left( \sqrt{\frac{\operatorname{Im} \tilde{m}(z)}{N\eta}} + \frac{\Phi(z)}{\sqrt{N\eta}} \right) \right)$$

on $\tilde{\Omega}(\eta)$ with $(\xi, \nu)$-high probability. Hence, recalling the definition of $D_L$ from Equation (45) and our bounds on $L$ from Equation (46), we have

$$\left| ((\mathbf{H}_{.i})^{\backslash i})^{\top} \tilde{\mathbf{G}}^{(\backslash i)}(z)(\mathbf{H}_{.i})^{\backslash i} - \mathbb{E}\left[ ((\mathbf{H}_{.i})^{\backslash i})^{\top} \tilde{G}^{(\backslash i)}(z)(\mathbf{H}_{.i})^{\backslash i} \mid (\mathbf{H})^{\backslash i} \right] \right| \leq C\Phi(z). \tag{122}$$

on $\tilde{\Omega}(\eta)$ with $(\xi, \nu)$-high probability. Applying the triangle inequality in Equation (115) and using Equations (122) and (116),

$$\left| \tilde{Z}_i(z) \right| \leq C\Phi(z) + Cr^2 (\log N)^{\xi + 3\gamma} \left( \frac{1}{q} + \sqrt{\frac{\operatorname{Im} \tilde{m}(z)}{N\eta}} + \frac{1}{N\eta} \right) \leq C\Phi(z) \tag{123}$$

on $\tilde{\Omega}(\eta)$ with $(\xi, \nu)$-high probability. Hence, taking the maximum over $i \in [N]$ we have, after increasing the value of $\nu$ by a constant factor, on the event $\tilde{\Omega}(\eta)$ with $(\xi, \nu)$-high-probability,

$$\max_i |\tilde{Z}_i(z)| \le C\Phi(z),$$

yielding Equation (103).

Finally, to bound $\tilde{\mathcal{A}}_i(z)$, recall its definition from Equation (66),

$$\tilde{\mathcal{A}}_i(z) = L_{ii} - \sum_{\alpha=N-r+1}^{N} \lambda_\alpha^2 (v_{i\alpha}(\mathbf{v}_\alpha)^{\backslash i})^\top \tilde{G}^{(\backslash i)}(z) v_{i\alpha}(\mathbf{v}_\alpha)^{\backslash i}$$
$$- \sum_{\alpha_1 \ne \alpha_2} \lambda_{\alpha_1} \lambda_{\alpha_2} (v_{i\alpha_1}(\mathbf{v}_{\alpha_1})^{\backslash i})^\top \tilde{G}^{(\backslash i)}(z) v_{i\alpha_2}(\mathbf{v}_{\alpha_2})^{\backslash i} + \frac{1}{N} \sum_j \frac{\tilde{G}_{ij}(z)\tilde{G}_{ji}(z)}{\tilde{G}_{ii}(z)} \tag{124}$$

For the first term, Assumption ($\tilde{\mathbf{A}}$1) ensures that

$$L_{ii} = C\left(\frac{r\rho_N(\log N)^{2\gamma}}{\sqrt{N}}\right). \tag{125}$$

There are $r$ terms in Equation (124) that can be controlled using Lemma D.10, providing the bounds of the form

$$\left|\lambda_{\alpha_1}^2 (v_{i\alpha_1}(\mathbf{v}_{\alpha_1})^{\backslash i})^\top \tilde{\mathbf{G}}^{(\backslash i)}(z) v_{i\alpha_1}(\mathbf{v}_{\alpha_1})^{\backslash i}\right| \le \left(\frac{C\lambda_j(\log N)^{2\gamma}}{N} + C\frac{r^2(\log N)^{6\gamma}}{N\eta}\right) \tag{126}$$

with $(\xi, \nu)$-high probability. There are $\binom{r}{2}$ cross-terms in Equation (124) that can be controlled using Lemma D.11, providing the bounds for $\alpha_1 \ne \alpha_2$ of the form

$$\left|\lambda_{\alpha_2} \lambda_{\alpha_1} (v_{i\alpha_1}(\mathbf{v}_{\alpha_1})^{\backslash i})^\top \tilde{\mathbf{G}}^{(\backslash i)}(z) v_{i\alpha_2}(\mathbf{v}_{\alpha_2})^{\backslash i}\right| \le C\frac{r^2(\log N)^{6\gamma}}{N\eta} \tag{127}$$

with $(\xi, \nu)$-high probability. The last term in Equation (124) is controlled using Lemma D.22 and Ward's Identity given in Equation (119). Altogether, applying Equations (125), (126) and (127), Lemma ($\tilde{\mathbf{A}}$1) and Ward's Identity to Equation (124), we have

$$|\tilde{\mathcal{A}}_i(z)| \le C\frac{r(\log N)^{2\gamma}}{\sqrt{N}} + C\frac{r^4(\log N)^{6\gamma}}{N\eta}$$

on $\tilde{\Omega}(\eta)$ with $(\xi, \nu)$-high probability. Taking the maximum over $i \in [N]$, and reducing the value of $\nu$ by a constant factor yields that on the event $\tilde{\Omega}(\eta)$ with $(\xi, \nu)$-high probability,

$$\max_i |\tilde{\mathcal{A}}_i(z)| \le \frac{Cr(\log N)^{2\gamma}}{\sqrt{N}} + \frac{Cr^4(\log N)^{6\gamma}}{N\eta},$$

establishing Equation (104) and completing the proof. $\qquad\square$

The following is a straightforward adaptation of Lemma 7.7 from Erdős et al. (2013a). Details are included for the sake of completeness.

**Lemma D.13.** *For $z = E + \mathrm{i}\eta \in D_L$, we have on the event $\tilde{\Omega}(\eta)$ with $(\xi, \nu)$-high probability,*

$$\max_i |\tilde{G}_{ii}(z) - \tilde{m}(z)| \le C\Phi(z) \tag{128}$$

*and*

$$\tilde{\Lambda}_d(z) \le \tilde{\Lambda}(z) + C\Phi(z). \tag{129}$$

*Proof.* Recall the definition of $\tilde{Y}_i(z)$ in Equation (63). Then by Lemma D.12,

$$\max_i |\tilde{Y}_i(z)| = \max_i \left|H_{ii} - \tilde{Z}_i(z) + \tilde{\mathcal{A}}_i(z)\right| \le \frac{C}{q} + C\Phi(z) + \frac{Cr(\log N)^{2\gamma}}{\sqrt{N}} + \frac{Cr^4(\log N)^{6\gamma}}{N\eta}$$

on $\tilde{\Omega}(\eta)$ with $(\xi, \nu)$-high probability. Then we see that

$$|\tilde{G}_{ii}(z) - \tilde{G}_{jj}(z)| = \left| \frac{\tilde{G}_{ii}(z)\tilde{G}_{jj}(z)}{\tilde{G}_{jj}(z)} - \frac{\tilde{G}_{jj}(z)\tilde{G}_{ii}(z)}{\tilde{G}_{ii}(z)} \right| = |\tilde{G}_{jj}(z)||\tilde{G}_{ii}(z)| \left| \frac{1}{\tilde{G}_{jj}(z)} - \frac{1}{\tilde{G}_{ii}(z)} \right|$$

$$= |\tilde{G}_{jj}(z)||\tilde{G}_{ii}(z)| \left| \tilde{Y}_j(z) - \tilde{Y}_i(z) \right|.$$

The last equality follows from substituting the identity for $\tilde{G}_{ii}(z)$, found in Lemma D.6. Then since on $\tilde{\Omega}(\eta)$, for any $i \in [N]$ uniformly, $c \le |\tilde{G}_{ii}(z)| \le C$, we have by triangle inequality with $(\xi, \nu)$-high probability,

$$|\tilde{G}_{ii}(z) - \tilde{G}_{jj}(z)| \le C \left( \frac{C}{q} + C\Phi(z) + \frac{Cr(\log N)^{2\gamma}}{\sqrt{N}} + \frac{Cr^4(\log N)^{6\gamma}}{N\eta} \right)$$

Hence,

$$\left| \frac{1}{N} \sum_j (\tilde{G}_{ii}(z) - \tilde{G}_{jj}(z)) \right| \le C\Phi(z)$$

on $\tilde{\Omega}(\eta)$ with $(\xi, \nu)$-high probability. Taking the maximum over $i \in [N]$ and recalling that

$$\tilde{m}(z) = \frac{1}{N} \operatorname{Tr} \tilde{\mathbf{G}}(z)$$

yields Equation (128).

Recalling the definition of $\tilde{\Lambda}_d(z)$ from Equation (73) and applying the triangle inequality,

$$\tilde{\Lambda}_d(z) = \max_i |\tilde{G}_{ii}(z) - m_{\mathrm{sc}}(z)| \le |\tilde{m}(z) - m_{\mathrm{sc}}(z)| + \max_i |\tilde{G}_{ii}(z) - \tilde{m}(z)| \le \tilde{\Lambda}(z) + \max_i |\tilde{G}_{ii}(z) - \tilde{m}(z)|,$$

where the second inequality follows from the definition of $\tilde{\Lambda}(z)$ in Equation (43). Applying Equation (128) then yields Equation (129), completing the proof. $\square$

**Lemma D.14** (Erdős et al. (2013a) Lemma 7.8, adapted)**.** *If $\eta \ge 2$, then $\tilde{\Omega}(\eta)$ holds with $(\xi, \nu)$-high probability, where $\tilde{\Omega}(\eta)$ is defined in Equation* (74)*.*

*Proof.* Let $z = E + i\eta$. To estimate $\tilde{\Lambda}_o$, we follow the approach from Lemma D.12. Again, we apply Lemma D.18 to obtain

$$\tilde{G}_{ij}(z) = \tilde{G}_{ii}(z)\tilde{G}_{jj}^{(\backslash i)}(z) \left( ((\mathbf{B}_{\cdot i})^{\backslash ij})^\top \tilde{\mathbf{G}}^{(\backslash ij)}(z) (\mathbf{B}_{\cdot j})^{\backslash ij} \right) - \tilde{G}_{ii}(z)\tilde{G}_{jj}^{(\backslash i)}(z)B_{ij}.$$

Using Lemma D.23, we have with $(\xi, \nu)$-high probability,

$$\left| \tilde{G}_{ij}(z) \right| \le C \left| ((\mathbf{B}_{\cdot i})^{\backslash ij})^\top \tilde{\mathbf{G}}^{(\backslash ij)}(z) (\mathbf{B}_{\cdot j})^{\backslash ij} \right| + C |B_{ij}| = C \left| \sum_{k,l}^{(\backslash ij)} B_{ik} \tilde{G}_{kl}^{(\backslash ij)}(z) B_{lj} \right| + C |B_{ij}|.$$

Applying Lemma D.20 and Lemma (D.24),

$$\left| \tilde{G}_{ij}(z) \right| \le C \left| \sum_{k,l}^{(\backslash ij)} B_{ik} \tilde{G}_{kl}^{(\backslash ij)}(z) B_{lj} \right| + C \frac{r\rho_N (\log N)^{2\gamma}}{\sqrt{N}} + \frac{C}{q} \tag{130}$$

To estimate $\left| \sum_{k,l}^{(\backslash ij)} B_{ik} \tilde{G}_{kl}^{(\backslash ij)}(z) B_{lj} \right|$, we again have to estimate $\left| \sum_{k,l}^{(\backslash ij)} h_{ik} \tilde{G}_{kl}^{(\backslash ij)}(z) h_{lj} \right|$. So then by Equation (80) from Lemma D.8, we have

$$\left| \sum_{k,l}^{(\backslash ij)} h_{ik} \tilde{G}_{kl}^{(\backslash ij)}(z) h_{lj} \right| \le (\log N)^{2\xi} \left[ \frac{\max_k \left| \tilde{G}_{kk}^{(\backslash ij)}(z) \right|}{q^2} + \frac{\max_{i \ne j} \left| \tilde{G}_{kl}^{(\backslash ij)}(z) \right|}{q} + \left( \sum_{k,l}^{(\backslash ij)} \left| \frac{\tilde{G}_{kl}^{(\backslash ij)}(z)}{N} \right|^2 \right)^{1/2} \right] \tag{131}$$

with $(\xi, \nu)$-high probability. Examining the rightmost term, we see that

$$\frac{1}{N^2} \sum_{k,l}^{(\backslash ij)} \left| \tilde{G}_{kl}^{(\backslash ij)}(z) \right|^2 = \frac{1}{N^2 \eta} \sum_{k}^{(\backslash ij)} \operatorname{Im} \tilde{G}_{kk}^{(\backslash ij)}(z) = \frac{1}{N\eta} \operatorname{Im} \tilde{m}^{(\backslash ij)}(z) \le \frac{C}{N} \tag{132}$$

with $(\xi, \nu)$-high probability, where the first equality is by Ward's Identity from Equation (119), the second equality is by the definition of $\tilde{m}^{(\backslash ij)}(z)$ given in Equation (42), and the inequality follows from $\eta \ge 2$ and Lemma D.23. Substituting Equation (132) into (131) yields

$$\left| \sum_{k,l}^{(\backslash ij)} h_{ik} \tilde{G}_{kl}^{(\backslash ij)}(z) h_{lj} \right| \le (\log N)^{2\xi} \left[ \frac{\max_i \left| \tilde{G}_{kk}^{(\backslash ij)}(z) \right|}{q^2} + \frac{\max_{i \ne j} \left| \tilde{G}_{kl}^{(\backslash ij)}(z) \right|}{q} + \frac{C}{\sqrt{N}} \right]$$

with $(\xi, \nu)$-high probability. Distributing and using Lemma D.23, we have

$$\left| \sum_{k,l}^{(\backslash ij)} h_{ik} \tilde{G}_{kl}^{(\backslash ij)}(z) h_{lj} \right| \le \frac{C(\log N)^{2\xi}}{q^2} + \frac{C(\log N)^{2\xi} \tilde{\Lambda}_o(z)}{q} + \frac{C(\log N)^{2\xi}}{\sqrt{N}} \tag{133}$$

with $(\xi, \nu)$-high probability. For the terms of $\left| \sum_{k,l}^{(\backslash ij)} B_{ik} \tilde{G}_{kl}^{(\backslash ij)}(z) B_{lj} \right|$ controlled by Lemma D.9 (i.e., terms that are linear in $(\mathbf{H}_{\cdot i})^{\backslash i}$, we modify the argument in Lemma D.9 in places where the bound $R \le C\eta$ was used, such as Equation (91), by replacing the bound with the trivial $R \le 1$. In doing so, we find that, as in the result of Lemma D.9, for any $\alpha_1 \in \{N - r + 1, \ldots, N\}$, when $\eta \ge 2$,

$$\left| \lambda_{\alpha_1} (v_{i\alpha_1} (\mathbf{v}_{\alpha_1})^{\backslash ij})^\top \tilde{\mathbf{G}}^{(\backslash ij)}(z) (\mathbf{H}_{\cdot i})^{\backslash ij} \right| \le Cr(\log N)^{\xi + 3\gamma} \left( \frac{1}{q} + \sqrt{\frac{\operatorname{Im} \tilde{m}(z)}{N\eta}} + \frac{1}{N\eta} \right)$$

with $(\xi, \nu)$-high probability. Then using Lemma D.23 and the assumption that $\eta \ge 2$ here, we have that with $(\xi, \nu)$-high probability,

$$\left| \lambda_{\alpha_1} (v_{i\alpha_1} (\mathbf{v}_{\alpha_1})^{\backslash ij})^\top \tilde{\mathbf{G}}^{(\backslash ij)}(z) (\mathbf{H}_{\cdot i})^{\backslash ij} \right| \le Cr(\log N)^{\xi + 3\gamma} \left( \frac{1}{q} + \frac{1}{\sqrt{N}} \right). \tag{134}$$

Applying the triangle inequality to $\left| \sum_{k,l}^{(\backslash ij)} B_{ik} \tilde{G}_{kl}^{(\backslash ij)}(z) B_{lj} \right|$ yields $2r$ terms that, by Lemma D.10, are bounded by $C((\log N)^{2\gamma} / \sqrt{N})$; and $\binom{r}{2}$ terms that, by Lemma D.11, are bounded by $C(r^2 (\log N)^{6\gamma} / N)$, both with $(\xi, \nu)$-high probability. That is,

$$\left| \sum_{k,l}^{(\backslash ij)} B_{ik} \tilde{G}_{kl}^{(\backslash ij)}(z) B_{lj} \right| \le 2r \frac{C(\log N)^{2\gamma}}{\sqrt{N}} + C \binom{r}{2} \frac{r^2 (\log N)^{6\gamma}}{N} \tag{135}$$

with $(\xi, \nu)$-high probability. Altogether, from Lemmas D.10, D.11, and Equation (134), we have that with $(\xi, \nu)$-high probability,

$$\begin{aligned} |\tilde{G}_{ij}(z)| &\le \frac{C(\log N)^{2\xi}}{q^2} + \frac{C(\log N)^{2\xi} \tilde{\Lambda}_o(z)}{q} + \frac{C(\log N)^{2\xi}}{\sqrt{N}} \\ &\quad + Cr^2 (\log N)^{\xi + 3\gamma} \left( \frac{1}{q} + \frac{1}{\sqrt{N}} \right) + \frac{Cr^4 (\log N)^{6\gamma}}{N}. \end{aligned} \tag{136}$$

Taking the maximum over $i \ne j$ (adjusting $\nu$ by a constant factor as needed), using the fact that $1/q \le 1/(r^2 (\log N)^{3\xi + 3\gamma})$ from Equation (34), and Assumption (Ã5), we have, with $(\xi, \nu)$-high-probability,

$$\tilde{\Lambda}_o(z) \le C(\log N)^{-2\xi}. \tag{137}$$

To bound $|\tilde{Z}_i(z)|$, we apply Equation (134) $2r$ times to obtain, from Equation (115),

$$\begin{aligned} \left| \tilde{Z}_i(z) \right| &\le Cr^2 (\log N)^{\xi + 3\gamma} \left( \frac{1}{q} + \frac{1}{\sqrt{N}} \right) + \left| ((\mathbf{H}_{\cdot i})^{\backslash i})^\top \tilde{\mathbf{G}}^{(\backslash i)}(z) (\mathbf{H}_{\cdot i})^{\backslash i} \right. \\ &\quad \left. - \mathbb{E} \left[ ((\mathbf{H}_{\cdot i})^{\backslash i})^\top \tilde{\mathbf{G}}^{(\backslash i)}(z) (\mathbf{H}_{\cdot i})^{\backslash i} \mid (\mathbf{H})^{\backslash i} \right] \right| \end{aligned} \tag{138}$$

with $(\xi, \nu)$-high probability. As in the proof of D.12, the right two terms of Equation (138) are bounded as

$$
\left| ((\mathbf{H_{.i}})^{\backslash i})^\top \tilde{\mathbf{G}}^{(\backslash i)}(z)(\mathbf{H_{.i}})^{\backslash i} - \mathbb{E}\left[ ((\mathbf{H_{.i}})^{\backslash i})^\top \tilde{\mathbf{G}}^{(\backslash i)}(z)(\mathbf{H_{.i}})^{\backslash i} \mid (\mathbf{H})^{\backslash i} \right] \right|
$$

$$
\leq (\log N)^{2\xi} \left[ \frac{\max_{k \neq l} \left| \tilde{G}_{kl}^{(\backslash i)}(z) \right|}{q} + \left( \frac{1}{N^2} \sum_{k \neq l} \left| \tilde{G}_{kl}^{(\backslash i)}(z) \right|^2 \right)^{1/2} \right] + (\log N)^\xi \frac{\max_k \left| \tilde{G}_{kk}^{(\backslash i)}(z) \right|}{q}
$$

with $(\xi, \nu)$-high probability. Using Lemma D.23 and Equation (137), we have

$$
\left| ((\mathbf{H_{.i}})^{\backslash i})^\top \tilde{\mathbf{G}}^{(\backslash i)}(z)(\mathbf{H_{.i}})^{\backslash i} - \mathbb{E}\left[ ((\mathbf{H_{.i}})^{\backslash i})^\top \tilde{\mathbf{G}}^{(\backslash i)}(z)(\mathbf{H_{.i}})^{\backslash i} \mid (\mathbf{H})^{\backslash i} \right] \right|
$$

$$
\leq \frac{C}{q} + \frac{C(\log N)^\xi}{q} + (\log N)^{2\xi} \left( \frac{1}{N^2} \sum_{k \neq l} \left| \tilde{G}_{kl}^{(\backslash i)}(z) \right|^2 \right)^{1/2} \tag{139}
$$

with $(\xi, \nu)$-high probability. Following the arguments leading up to Equation (132), using Ward's identity, we obtain

$$
\left( \frac{1}{N^2} \sum_{k \neq l} \left| \tilde{G}_{kl}^{(\backslash i)}(z) \right|^2 \right)^{1/2} \leq \frac{C}{\sqrt{N}}
$$

with $(\xi, \nu)$-high probability. Hence we have, applying this bound to Equation (139),

$$
\left| ((\mathbf{H_{.i}})^{\backslash i})^\top \tilde{G}^{(\backslash i)}(z)(\mathbf{H_{.i}})^{\backslash i} - \mathbb{E}\left[ ((\mathbf{H_{.i}})^{\backslash i})^\top \tilde{G}^{(\backslash i)}(z)(\mathbf{H_{.i}})^{\backslash i} \mid (\mathbf{H})^{\backslash i} \right] \right| \leq \frac{C(\log N)^\xi}{q} + \frac{C(\log N)^{2\xi}}{\sqrt{N}}
$$

with $(\xi, \nu)$-high probability. Applying this bound to Equation (138) yields

$$
\left| \tilde{Z}_i(z) \right| \leq Cr^2 (\log N)^{\xi + 3\gamma} \left( \frac{1}{q} + \frac{1}{\sqrt{N}} \right) + \frac{C(\log N)^{2\xi}}{\sqrt{N}} \leq C(\log N)^{-2\xi}
$$

with $(\xi, \nu)$-high probability, where in the second inequality we applied Equation (34) to upper bound $q$. Taking the maximum over $i$ and reducing the value of $\nu$ yields the bound for $\tilde{Z}_i(z)$ with $(\xi, \nu)$-high-probability.

To estimate $\tilde{\mathcal{A}}_i(z)$, we follow the approach taken in Lemma D.12. After applying the triangle inequality, one application of Assumption (**Ã1**), $r$ applications of D.10, and $\binom{r}{2}$ applications of Lemma D.11 yield that with $(\xi, \nu)$-high probability,

$$
\left| \tilde{\mathcal{A}}_i(z) \right| \leq \frac{Cr\rho_N (\log N)^{2\gamma}}{\sqrt{N}} + C\left( \frac{r\sqrt{N}(\log N)^{2\gamma}}{N} + \frac{r^3(\log N)^{6\gamma}}{N\eta} \right) + C\frac{r^4(\log N)^{6\gamma}}{N\eta} + \left| \frac{1}{N}\sum_{j=1}^N \frac{\tilde{G}_{ij}(z)\tilde{G}_{ji}(z)}{\tilde{G}_{ii}(z)} \right|.
$$

Substituting our choice of $\eta \geq 2$ and applying Assumption (**Ã5**), we obtain

$$
\left| \tilde{\mathcal{A}}_i(z) \right| \leq C\frac{(\log N)^{2\gamma + \zeta}}{\sqrt{N}} + C\frac{(\log N)^{6\gamma + 4\zeta}}{N} + \left| \frac{1}{N}\sum_{j=1}^N \frac{\tilde{G}_{ij}(z)\tilde{G}_{ji}(z)}{\tilde{G}_{ii}(z)} \right| \tag{140}
$$

with $(\xi, \nu)$-high probability. From Lemma D.18, we have

$$
\frac{\tilde{G}_{ij}(z)}{\tilde{G}_{ii}(z)}\tilde{G}_{ji}(z) = -\tilde{G}_{jj}^{(\backslash i)}(z)\left(B_{ij} - \sum_{k,l}^{(\backslash ij)} B_{ik}\tilde{G}_{kj}^{(\backslash ij)}(z)B_{lj}\right)\tilde{G}_{ji}(z)
$$

Hence by Lemma D.23, we have

$$
\frac{1}{N}\sum_{j=1}^N \left| \frac{\tilde{G}_{ij}(z)}{\tilde{G}_{ii}(z)}\tilde{G}_{ji}(z) \right| \leq \left| C\tilde{\Lambda}_o(z)\left( B_{ij} - \sum_{k,l}^{(\backslash ij)} B_{ik}\tilde{G}_{kj}^{(\backslash ij)}(z)B_{lj} \right) \right|
$$

$$
\leq C\tilde{\Lambda}_o(z)\left( |B_{ij}| + \left| \sum_{k,l}^{(\backslash ij)} B_{ik}\tilde{G}_{kj}^{(\backslash ij)}(z)B_{lj} \right| \right) \tag{141}
$$

with $(\xi, \nu)$-high probability. To bound the absolute value of the right-hand side of the last inequality, we apply Equations (135) and (136) to obtain

$$\left| \sum_{k,l} B_{ik} \tilde{G}_{kj}^{(\backslash ij)}(z) B_{lj} \right| \leq C \frac{r^2 (\log N)^{\xi + 3\gamma}}{q} \leq C (\log N)^{-2\xi},$$

where we have used $q \geq r^2 (\log N)^{3\xi + 3\gamma}$ from Equation (34). Applying this to Equation (141), controlling $|B_{ij}|$ with Lemma D.24, and bounding $\tilde{\Lambda}_o \leq C$ by Lemma D.23, we have

$$\frac{1}{N} \sum_{j=1}^{N} \left| \frac{\tilde{G}_{ij}(z)}{\tilde{G}_{ii}(z)} \tilde{G}_{ji}(z) \right| \leq C (\log N)^{-2\xi}$$

with $(\xi, \nu)$-high probability. Substituting this bound into Equation (140),

$$\left| \tilde{\mathcal{A}}_i(z) \right| \leq C \frac{(\log N)^{2\gamma + \zeta}}{\sqrt{N}} + C \frac{(\log N)^{6\gamma + 4\zeta}}{N} + C (\log N)^{-2\xi} \leq C (\log N)^{-2\xi}$$

for sufficiently large $N$ with $(\xi, \nu)$-high probability. Taking the maximum over $i$ (reducing the value of $\nu$ by a constant factor, if needed) yields the result with $(\xi, \nu)$-high-probability. All of these rates are as in Lemma 7.8 of Erdős et al. (2013a). The remainder of the proof then follows from the arguments contained in Lemma 3.16 of Erdős et al. (2013a). □

The following result, which establishes the global and strong local semicircle laws for the eigenvalues of $\mathbf{B}$, might be of independent interest. Note that the conditions on $\xi$ and $q$ are slightly stronger than those made elsewhere in this appendix.

**Theorem D.15.** *Let $z = E + i\eta \in D$ and define $\kappa_E = ||E| - 2|$. Suppose $\mathbf{B}$ satisfies Equation (36), and Assumptions ($\tilde{\mathbf{A}}1$) – ($\tilde{\mathbf{A}}7$) hold, and suppose that*

$$\xi = \frac{C_0}{2} \log(\log N) \quad and \quad r^2 (\log N)^{C_3 + 3\gamma} \leq q.$$

*Then there are universal constants $C_3, C_4 > 0$ such that for some $\nu > 0$ that depends on $C_0, C_1$, and $C_2$ in Equations (D.2), (34), and Definition D.2, such that the event*

$$|\tilde{m}(z) - m_{\text{sc}}(z)| \leq \frac{r\pi}{N\eta} + (\log N)^{C_4 \xi} \left( \min \left( \frac{1}{q^2 \sqrt{\kappa_E + \eta}}, \frac{1}{q} \right) + \frac{1}{N\eta} \right) \tag{142}$$

*holds with $(\xi, \nu)$-high probability uniformly over all $z \in D$. Moreover, we have with $(\xi, \nu)$-high probability uniformly over all $z \in D$*

$$\max_{1 \leq i,j \leq N} \left| \tilde{G}_{ij}(z) - I(i = j) m_{\text{sc}}(z) \right| \leq \frac{r^2 (\log N)^{C_3 \xi + 3\gamma}}{q} + r^2 (\log N)^{C_3 \xi + 6\gamma} \left( \sqrt{\frac{\text{Im}\, m_{\text{sc}}(z)}{N\eta}} + \frac{r^2}{N\eta} \right). \tag{143}$$

*Proof.* To show Equation (142), we have from Lemma D.3,

$$\left| \tilde{\Lambda}(z) - \Lambda \right| \leq \frac{r\pi}{N\eta}.$$

From which, observe that

$$\tilde{\Lambda}(z) \leq \frac{r\pi}{N\eta} + \Lambda(z).$$

Applying Theorem 2.8 from Erdős et al. (2013a), specifically Equation (2.16), and recalling the definition of $\tilde{\Lambda}(z)$ and $\Lambda(z)$ from Equation (43) we have uniformly over $z \in D$ with $(\xi, \nu)$-high probability, the event

$$|\tilde{m}(z) - m_{\text{sc}}(z)| \leq \frac{r\pi}{N\eta} + (\log N)^{C_2 \xi} \left( \min \left( \frac{1}{q^2 \sqrt{\kappa_E + \eta}}, \frac{1}{q} \right) + \frac{1}{N\eta} \right),$$

as desired.

To show Equation (143), we first extend Lemma D.14 uniformly for $z \in D_L$ using a continuity argument. Substituting Lemmas D.12 and D.13 and Equation 142 for Equations (7.14), (7.22), and (2.20) from Erdős et al. (2013a) respectively, the argument follows exactly as given for Lemma 7.9 from Erdős et al. (2013a), and details are omitted.

To extend the bound uniformly for $z \in D$, follow the arguments given in (Erdős et al., 2013a) after Equation (4.28) until the end of section 4.1. □

## D.2. Proof of Delocalization

We are now prepared to prove our main result on the delocalization of the eigenvectors of $\mathbf{B}$. We remark that only the weak local law is needed to show delocalization.

**Theorem D.16.** *Let $\mathbf{B}$ satisfy Equation* (36) *and let $\mathcal{S} = \{N - r + 1, \dots, N\}$. Under Assumptions* ($\tilde{\mathbf{A}}1$) – ($\tilde{\mathbf{A}}7$), *for some $\nu > 0$ that depends on $C_0, C_1,$ and $C_2$ in Definition D.2, Equations* (32) *and* (34)*, we have*

$$\max_{\alpha \in \mathcal{S}} \max_{j} |\hat{v}_{j\alpha}| \leq C \frac{r^2 (\log N)^{4\xi + 6\gamma}}{\sqrt{N}} \tag{144}$$

*with $(\xi, \nu)$-high probability.*

*Proof.* Following Erdős et al. (2013a), set

$$L = 8\xi, \tag{145}$$

and set

$$\eta = \frac{r^4 (\log N)^{L + 12\gamma}}{N}. \tag{146}$$

By Equation (129) in Lemma D.13,

$$\tilde{\Lambda}_d(z) \leq \tilde{\Lambda}(z) + C\Phi(z). \tag{147}$$

We examine $\tilde{\Lambda}(z)$ first. By Lemma D.3 and Theorem 3.1 from Erdős et al. (2013a), we have

$$\tilde{\Lambda}(z) \leq |\tilde{\Lambda}(z) - \Lambda(z)| + \Lambda(z) \leq \frac{r\pi}{N\eta} + \frac{C(\log N)^{\xi}}{\sqrt{q}} + \frac{C(\log N)^{2\xi}}{(N\eta)^{1/3}}. \tag{148}$$

Applying this to Equation (147) and recalling the definition of the spectral control parameter $\Phi(z)$ from Equation (101),

$$\tilde{\Lambda}_d(z) \leq \frac{r\pi}{N\eta} + \frac{C(\log N)^{\xi}}{\sqrt{q}} + \frac{C(\log N)^{2\xi}}{(N\eta)^{1/3}} + \frac{r^2 (\log N)^{\xi + 3\gamma}}{q} + r^2 (\log N)^{2\xi + 6\gamma} \left( \sqrt{\frac{\operatorname{Im} \tilde{m}(z)}{N\eta}} + \frac{r^2}{N\eta} \right). \tag{149}$$

By the triangle inequality,

$$\operatorname{Im} \tilde{m}(z) \leq |\operatorname{Im} \tilde{m}(z) - \operatorname{Im} m_{\mathrm{sc}}(z)| + \operatorname{Im} m_{\mathrm{sc}}(z) \leq |\tilde{m}(z) - m_{\mathrm{sc}}| + \operatorname{Im} m_{\mathrm{sc}}(z) = \tilde{\Lambda}(z) + \operatorname{Im} m_{\mathrm{sc}}(z)$$

where in the equality, we have substituted the definition of $\tilde{\Lambda}(z)$ from Equation (43).

Now since $C^{-1} \leq |m_{\mathrm{sc}}(z)| \leq C$ for $z \in D_L$ by Lemma 3.2 in Erdős et al. (2013a),
and using Equation (148), we have

$$\operatorname{Im} \tilde{m}(z) \leq \frac{r\pi}{N\eta} + \frac{C(\log N)^{\xi}}{\sqrt{q}} + \frac{C(\log N)^{2\xi}}{(N\eta)^{1/3}} + C$$

Substituting our value for $\eta$ from Equation (146), recalling the bounds on $q$ from Equation (34) and simplifying, we have

$$\operatorname{Im} \tilde{m}(z) \leq \frac{r\pi}{r^4 (\log N)^{8\xi + 12\gamma}} + \frac{C(\log N)^{\xi}}{r (\log N)^{(3\xi + 3\gamma)/2}} + \frac{C(\log N)^{2\xi}}{(r^4 (\log N)^{8\xi + 12\gamma})^{1/3}} + C \leq C \tag{150}$$

Hence, we find that by substituting Equation (150) into Equation (149),

$$\tilde{\Lambda}_d(z) \leq \frac{r\pi}{r^4 (\log N)^{8\xi + 12\gamma}} + \frac{C(\log N)^{\xi}}{r (\log N)^{(3\xi + 3\gamma)/2}} + \frac{C(\log N)^{2\xi}}{(r^4 (\log N)^{8\xi + 12\gamma})^{1/3}} + \frac{r^2 (\log N)^{\xi + 3\gamma}}{r^2 (\log N)^{3\xi + 3\gamma}}$$
$$+ r^2 (\log N)^{2\xi + 6\gamma} \left( \sqrt{\frac{C}{r^4 (\log N)^{8\xi + 12\gamma}}} + \frac{r^2}{r^4 (\log N)^{8\xi + 12\gamma}} \right).$$

Bounding decaying terms of the form $(\log n)^{-c}$ by a constant,

$$\tilde{\Lambda}_d(z) \leq C.$$

Recalling the definition of $\tilde{\Lambda}_d(z)$ from (73), we have for any $j \in [N]$

$$|\tilde{G}_{jj}(z) - m_{\mathrm{sc}}(z)| \leq C.$$

Recalling again that $|m_{\mathrm{sc}}(z)|$ is bounded by constants from Lemma 3.2 in (Erdős et al., 2013a), we have that

$$|\tilde{G}_{jj}(z)| \leq C,$$

which then implies that for $\alpha \notin \{N - r + 1, N - r + 2, \ldots, N\}$,

$$C \geq \mathrm{Im}\, \tilde{G}_{jj}(\hat{\lambda}_\alpha + i\eta) = \sum_{\beta=1}^{N} \frac{\eta|\hat{v}_{j\beta}|^2}{(\hat{\lambda}_\beta - \hat{\lambda}_\alpha)^2 + \eta^2} = \frac{\eta\,|\hat{v}_{j\alpha}|^2}{\eta^2} + \sum_{\beta \neq \alpha} \frac{\eta|\hat{v}_{j\beta}|^2}{(\hat{\lambda}_\beta - \hat{\lambda}_\alpha)^2 + \eta^2} \geq \frac{|\hat{v}_{j\alpha}|^2}{\eta}.$$

Rearranging and recalling our choice of $\eta$ from Equation (146) and $L$ from Equation (145),

$$|\hat{v}_{j\alpha}| \leq C\frac{r^2(\log N)^{4\xi+6\gamma}}{\sqrt{N}},$$

as we set out to show. $\square$

### D.3. Auxiliary Estimates

Here we collect a handful of useful matrix results, mostly related to resolvents.

**Lemma D.17** (Resolvent Identities; Benaych-Georges & Knowles (2018) Lemma 3.5). *For any Hermitian matrix* $\mathbf{B}$ *and* $\mathbb{T} \subset [N]$, *we have for any* $i, j, \notin \mathbb{T}$ *and* $i \neq j$

$$G_{ij}^{(\backslash\mathbb{T})}(z) = -G_{ii}^{(\backslash\mathbb{T})}(z) \sum_{k}^{(\backslash\mathbb{T}i)} B_{ik} G_{kj}^{(\backslash\mathbb{T}i)}(z) = -G_{jj}^{(\backslash\mathbb{T})}(z) \sum_{k}^{(\backslash\mathbb{T}j)} G_{ik}^{(\backslash\mathbb{T}j)}(z) B_{kj} \tag{151}$$

*Moreover, for any* $i, j,$ *and* $k \notin \mathbb{T}$ *and* $i, j \neq k$

$$G_{ij}^{(\backslash\mathbb{T})}(z) = G_{ij}^{(\backslash\mathbb{T}k)}(z) + \frac{G_{ik}^{(\backslash\mathbb{T})}(z)G_{kj}^{(\backslash\mathbb{T})}(z)}{G_{kk}^{(\backslash\mathbb{T})}(z)} \tag{152}$$

*Proof.* A proof is given in Appendix A of Benaych-Georges & Knowles (2018). $\square$

**Lemma D.18.** *For a resolvent* $\mathbf{G} = (\mathbf{F} - z\mathbf{I})^{-1}$, *where* $\mathbf{F}$ *is Hermitian, we have for any* $i \neq j \in [N]$

$$G_{ij}(z) = G_{ii}(z)G_{jj}^{(\backslash i)}(z) \sum_{k,l}^{(\backslash ij)} F_{ik} G_{kj}^{(\backslash ij)}(z) F_{lj} - G_{ii}(z)G_{jj}^{(\backslash i)}(z)H_{ij}.$$

*Proof.* This follows from Equation (151) in Lemma D.17. In particular, setting $\mathbb{T} = \emptyset$ in that lemma, we have

$$G_{ij}(z) = -G_{ii}(z) \sum_{k}^{(\backslash i)} F_{ik} G_{kj}^{(\backslash i)}(z) = -G_{ii}(z) \sum_{k}^{(\backslash ij)} F_{ik} G_{kj}^{(\backslash i)}(z) - G_{ii}(z)G_{jj}^{(\backslash i)}(z)F_{ij}. \tag{153}$$

Using Equation (151) again with $\mathbb{T} = \{i\}$, we have

$$G_{kj}^{(\backslash i)}(z) = -G_{jj}^{(\backslash i)}(z) \sum_{l}^{(\backslash ij)} G_{kl}^{(\backslash ij)}(z) F_{lj}.$$

Substituting this into (153) completes the proof. $\square$

**Lemma D.19.** *Let $\tau < N$ be constant with respect to $N$. For $|\mathbb{T}| \leq \tau$, where $\mathbb{T} \subset [N]$ we have that on the event $\tilde{\Omega}(\eta)$ defined in Equation (74),*

$$\max_{i \neq j} \left| \tilde{G}_{ij}^{(\backslash \mathbb{T})}(z) \right| \leq C \max_{i \neq j} \left| \tilde{G}_{ij}(z) \right| \leq (\log N)^{-\xi}$$

*Proof.* This is an application of Lemma D.17, from which, for $k \in \mathbb{T}$

$$\tilde{G}_{ij}(z) = \tilde{G}_{ij}^{(\backslash k)}(z) + \frac{\tilde{G}_{ik}(z)\tilde{G}_{kj}(z)}{\tilde{G}_{kk}(z)}.$$

Applying the reverse triangle inequality,

$$\left| \left| \tilde{G}_{ij}^{(\backslash k)}(z) \right| - \left| \frac{\tilde{G}_{ik}(z)\tilde{G}_{kj}(z)}{\tilde{G}_{kk}(z)} \right| \right| \leq \left| \tilde{G}_{ij}(z) \right|,$$

from which the triangle inequality yields

$$\left| \tilde{G}_{ij}^{(\backslash k)}(z) \right| \leq \left| \tilde{G}_{ij}(z) \right| + \frac{\left| \tilde{G}_{ik}(z) \right| \left| \tilde{G}_{kj} \right|}{\left| \tilde{G}_{kk}(z) \right|}. \tag{154}$$

By Lemma D.22, we have that on $\tilde{\Omega}(\eta)$, $c \leq \left| \tilde{G}_{ii}(z) \right| \leq C$, which then implies

$$\frac{\left| \tilde{G}_{ik}(z) \right| \left| \tilde{G}_{kj}(z) \right|}{\left| \tilde{G}_{kk}(z) \right|} \leq C \max_{i \neq j} \left| \tilde{G}_{ij}(z) \right|^2.$$

Applying this to Equation (154) and taking the maximum over $i \neq j$ yields

$$\max_{i \neq j} \left| \tilde{G}_{ij}^{(\backslash k)}(z) \right| \leq \max_{i \neq j} \left| \tilde{G}_{ij}(z) \right| + C \max_{i \neq j} \left| \tilde{G}_{ij}(z) \right|^2 \leq \max_{i \neq j} \left| \tilde{G}_{ij}(z) \right| \left( 1 + C \max_{i \neq j} \left| \tilde{G}_{ij}(z) \right| \right) \leq \max_{i \neq j} \left| \tilde{G}_{ij}(z) \right|.$$

where we used in the last inequality that on $\tilde{\Omega}(\eta)$, we have $\max_{i \neq j} \left| \tilde{G}_{ij}(z) \right| \leq (\log N)^{-\xi} \leq C$ by construction of $\tilde{\Omega}(\eta)$. Recalling the definition of $\tilde{\Lambda}_o(z)$ from Equation (73) yields that on $\tilde{\Omega}(\eta)$,

$$\max_{i \neq j} \left| \tilde{G}_{ij}^{(\backslash k)}(z) \right| \leq C \tilde{\Lambda}_o(z) \leq C(\log N)^{-\xi}. \tag{155}$$

where the second inequality follows from the definition of $\tilde{\Omega}(\eta)$. To see that $\max_{i \neq j} \left| \tilde{G}_{ij}^{(\backslash \mathbb{T})}(z) \right| \leq C \tilde{\Lambda}_o(z)$, we repeat the above for each $k \in \mathbb{T}$. In particular, in the next iteration, set $\tilde{\mathbf{G}} := \tilde{\mathbf{G}}^{(\backslash k)}$ and repeat the above for $k' \in \mathbb{T} \setminus \{k\}$. $\square$

**Lemma D.20.** *Let $\mathbf{H}$ be matrix whose entries, $H_{ij}$ are independent up to symmetry and satisfy Equation (33). For $C > 0$ sufficiently large, we have with $(\xi, \nu)$-high probability,*

$$\max_{i,j} |H_{ij}| \leq \frac{C}{q}. \tag{156}$$

*Proof.* Set $p = \nu(\log N)^\xi$ in Equation (33). Then by Markov's inequality,

$$\mathbb{P}\left( |H_{ij}| \geq \frac{C}{q} \right) = \mathbb{P}\left( |H_{ij}|^p \geq \frac{C^p}{q^p} \right) \leq \frac{\mathbb{E}\left[ |H_{ij}|^p \right] q^p}{C^p} \leq \frac{C_1^p q^p}{N q^{p-2} C^p} = \left( \frac{C_1}{C} \right)^p \frac{q^2}{N} \leq \left( \frac{C_1}{C} \right)^p,$$

where the second inequality follows from Equation (33) and the last inequality follows from Equation (34). Taking a union bound over all $i, j \in [N]$ yields

$$\mathbb{P}\left( \max_{i,j} |H_{ij}| \geq \frac{C}{q} \right) \leq N^2 \left( \frac{C_1}{C} \right)^p = e^{2(\log N)} \left( \frac{C_1}{C} \right)^{\nu(\log N)^\xi}.$$

So then for sufficiently large $C > 0$, and using the fact that $1 < \xi$ by Definition (32),

$$\mathbb{P}\left(\max_{i,j}|H_{ij}| \geq \frac{C}{q}\right) \leq e^{-\nu(\log N)^\xi},$$

completing the proof. $\qquad\qquad\qquad\qquad\qquad\qquad\qquad\qquad\qquad\qquad\qquad\qquad\qquad\qquad\qquad\qquad\qquad\qquad\square$

**Lemma D.21.** *Let $\mathbf{H}$ be as in Equation (33) and $z \in D_L$, then on $\tilde{\Omega}(\eta)$ defined in Equation (74), we have for all $i \neq j \in [N]$ that*

$$\left|\sum_{k,l}^{(\backslash ij)} h_{ik}\tilde{G}_{kl}^{(\backslash ij)}(z)h_{lj}\right| \leq \frac{C(\log N)^\xi}{q} + C(\log N)^{2\xi}\left(\sqrt{\frac{\operatorname{Im}\tilde{m}(z)}{N\eta}} + \frac{1}{N\eta}\right) \tag{157}$$

*with $(\xi, \nu)$-high probability.*

*Proof.* This is an application of Lemma D.8. By Equation (80) in that lemma, we have that with $(\xi, \nu)$-high probability

$$\left|\sum_{k,l}^{(\backslash ij)} h_{ik}\tilde{G}_{kl}^{(\backslash ij)}(z)h_{lj}\right| \leq (\log N)^{2\xi}\left[\frac{\max_k\left|\tilde{G}_{kk}^{(\backslash ij)}(z)\right|}{q^2} + \frac{\max_{i\neq j}\left|\tilde{G}_{kl}^{(\backslash ij)}(z)\right|}{q} + \left(\frac{1}{N^2}\sum_{k,l}^{(\backslash ij)}\left|\tilde{G}_{kl}^{(\backslash ij)}\right|^2\right)^{1/2}\right]. \tag{158}$$

Note that on $\tilde{\Omega}(\eta)$, we have $\max_{i\neq j}\left|G_{kl}^{(\backslash ij)}\right| \leq C\max_{i\neq j}|G_{kl}|$ by Lemma D.19. By the definition of $q$ in Equation (34), we have $q \geq (\log N)^{3\xi}$. Hence, on $\tilde{\Omega}(\eta)$, substituting this bound for $q$ into Equation (158), using the fact that $\max_i|G_{ii}| \leq C$ on $\tilde{\Omega}(\eta)$, and distributing,

$$\left|\sum_{k,l}^{(\backslash ij)} h_{ik}\tilde{G}_{kl}^{(\backslash ij)}(z)h_{lj}\right| \leq \frac{C(\log N)^{2\xi}}{q(\log N)^{3\xi}} + \frac{C(\log N)^{2\xi}\max_{i\neq j}|G_{ij}|}{q} + (\log N)^{2\xi}\left(\frac{1}{N^2}\sum_{i\neq j}\left|\tilde{G}_{ij}(z)\right|^2\right)^{1/2} \tag{159}$$

with $(\xi, \nu)$-high probability. Notice by Ward's Lemma (Equation (119)), the summation in Equation (159) can be expressed as

$$\sum_{k,l}^{(\backslash ij)}\left|\tilde{G}_{kl}^{(\backslash ij)}(z)\right|^2 = \frac{1}{\eta}\sum_k^{(\backslash ij)}\operatorname{Im}\tilde{G}_{kk}^{(\backslash ij)}(z). \tag{160}$$

Now notice that by using Equation (152) Lemma D.17 twice and using Lemma D.22, we have

$$\tilde{G}_{kk}^{(\backslash ij)}(z) = \tilde{G}_{kk}(z) + \frac{\left|\tilde{G}_{ik}(z)\right|^2}{\tilde{G}_{ii}(z)} + \frac{\left|\tilde{G}_{jk}(z)\right|^2}{\tilde{G}_{jj}(z)} = \tilde{G}_{kk}(z) + O(\tilde{\Lambda}_o^2(z)). \tag{161}$$

Substituting Equation (161) into (160) and dividing by $N^2$ yields

$$\frac{1}{N^2}\sum_{k,l}^{(\backslash ij)}\left|\tilde{G}_{kl}^{(\backslash ij)}(z)\right|^2 = \frac{1}{N^2\eta}\sum_k^{(\backslash ij)}\operatorname{Im}\tilde{G}_{kk}^{(\backslash ij)}(z) \leq \frac{1}{N\eta}\left(\operatorname{Im}\tilde{m}(z) + C\tilde{\Lambda}_o^2(z)\right), \tag{162}$$

where the inequality follows from the fact that

$$\tilde{m}(z) = \frac{1}{N}\operatorname{Tr}\tilde{\mathbf{G}}(z).$$

Substituting Equation (162) into Equation (159) yields that with $(\xi, \nu)$-high probability

$$\left|\sum_{k,l}^{(\backslash ij)} h_{ik}\tilde{G}_{kl}^{(\backslash ij)}(z)h_{lj}\right| \leq \frac{C}{q(\log N)^\xi} + \frac{(\log N)^{2\xi}\max_{i\neq j}\left|\tilde{G}_{ij}\right|}{q} + (\log N)^{2\xi}\left(\frac{1}{N\eta}\left(\operatorname{Im}\tilde{m}(z) + C\tilde{\Lambda}_o^2(z)\right)\right)^{1/2}.$$

Since $\sqrt{a+b} \le \sqrt{a} + \sqrt{b}$, it follows that

$$\left| \sum_{k,l}^{(\backslash ij)} h_{ik} \tilde{G}_{kl}^{(\backslash ij)}(z) h_{lj} \right| \le \frac{C}{q(\log N)^\xi} + \frac{C(\log N)^{2\xi} \tilde{\Lambda}_o(z)}{q} + (\log N)^{2\xi} \left( \sqrt{\frac{\operatorname{Im} \tilde{m}(z)}{N\eta}} + \frac{C\tilde{\Lambda}_o(z)}{\sqrt{N\eta}} \right) \tag{163}$$

with $(\xi, \nu)$-high probability. Using that $(\log N)^{4\xi} \le \sqrt{N\eta}$ on $D_L$, we have following Equation (163)

$$\left| \sum_{k,l}^{(\backslash ij)} h_{ik} \tilde{G}_{kl}^{(\backslash ij)}(z) h_{lj} \right| \le \frac{C}{q(\log N)^\xi} + \frac{C(\log N)^{2\xi} \tilde{\Lambda}_o(z)}{q} + \sqrt{\frac{\operatorname{Im} \tilde{m}(z)}{N\eta}} + \frac{C\tilde{\Lambda}_o(z)}{(\log N)^{4\xi}}.$$

Hence, by increasing the value of $\nu$ by a constant factor, we have that with $(\xi, \nu)$-high probability, it holds uniformly over all $i \ne j \in [N]$ that

$$\left| \sum_{k,l}^{(\backslash ij)} h_{ik} \tilde{G}_{kl}^{(\backslash ij)}(z) h_{lj} \right| \le \frac{C}{q(\log N)^\xi} + \frac{C(\log N)^{2\xi} \tilde{\Lambda}_o(z)}{q} + \sqrt{\frac{\operatorname{Im} \tilde{m}(z)}{N\eta}},$$

as we set out to show. $\qquad \square$

**Lemma D.22.** *Recall the definitions of $\tilde{\Lambda}_o(z)$ and $\tilde{\Lambda}_d(z)$ from Equation* (73)*,*

$$\tilde{\Lambda}_o(z) := \max_{i \ne j} |\tilde{G}_{ij}(z)|, \quad \tilde{\Lambda}_d(z) := \max_i |\tilde{G}_{ii}(z) - m_{\mathrm{sc}}(z)|.$$

*On the event $\tilde{\Omega}(\eta)$ given in Equation* (74)*, we have for any $i \in [N]$ and $z \in D_L$*

$$c \le \left| \tilde{G}_{ii}(z) \right| \le C. \tag{164}$$

*Further, for any $\mathbb{T} \subset [N]$ with $|\mathbb{T}| \le \tau < N$, where $\tau$ is constant with respect to $N$,*

$$c \le \left| \tilde{G}_{ii}^{(\backslash \mathbb{T})}(z) \right| \le C. \tag{165}$$

*Proof.* Notice that on $\tilde{\Omega}(\eta)$, since $\tilde{\Lambda}_d(z) \ge 0$, we have

$$\tilde{\Lambda}_o(z) \le \tilde{\Lambda}_d(z) + \tilde{\Lambda}_o(z) \le (\log N)^{-\xi},$$

so that for any $i \in [N]$,

$$-(\log N)^{-\xi} \le \tilde{G}_{ii}(z) - m_{\mathrm{sc}}(z) \le (\log N)^{-\xi}.$$

Adding $m_{\mathrm{sc}}(z)$ to both sides, we obtain

$$m_{\mathrm{sc}}(z) - (\log N)^{-\xi} \le \tilde{G}_{ii}(z) \le (\log N)^{-\xi} + m_{\mathrm{sc}}(z). \tag{166}$$

For $z \in D_L$, Lemma 3.2 in Erdős et al. (2013a) asserts that $C^{-1} \le |m_{\mathrm{sc}}(z)| \le C$. Using this fact in Equation (166) yields

$$c \le \left| \tilde{G}_{ii}(z) \right| \le C, \tag{167}$$

proving the claim in Equation (164).

For the claim in Equation (165), suppose we have $k \in \mathbb{T}$. We apply Equation (152), from which we obtain

$$\tilde{G}_{ii}(z) - \frac{\left| \tilde{G}_{ik}(z) \right|^2}{\tilde{G}_{kk}(z)} = \tilde{G}_{ii}^{(k)}(z). \tag{168}$$

Repeatedly applying Equation (152) to (168) for the remaining indices in $\mathbb{T}$ yields

$$\tilde{G}_{ii}(z) - \sum_{k \in \mathbb{T}} \frac{\left|\tilde{G}_{ik}(z)\right|^2}{\tilde{G}_{kk}(z)} = \tilde{G}_{ii}^{(\backslash \mathbb{T})}(z). \tag{169}$$

Applying Equation (169) to Equation (166) and recalling $m_{\mathrm{sc}}(z)$ is lower and upper bounded by constants from Lemma 3.2 in Erdős et al. (2013a), we see

$$m_{\mathrm{sc}}(z) - (\log N)^{-\xi} - \sum_{k \in \mathbb{T}} \frac{\left|\tilde{G}_{ik}(z)\right|^2}{\tilde{G}_{kk}(z)} \leq \tilde{G}_{ii}^{(k)}(z) \leq (\log N)^{-\xi} + m_{\mathrm{sc}}(z) - \sum_{k \in \mathbb{T}} \frac{\left|\tilde{G}_{ik}(z)\right|^2}{\tilde{G}_{kk}(z)}. \tag{170}$$

Notice that on $\tilde{\Omega}(\eta)$ and $z \in D_L$, we have from Equation (167) that $c \leq \left|\tilde{G}_{kk}(z)\right| \leq C$, so then on $\tilde{\Omega}(\eta)$ ,

$$\frac{\left|\tilde{G}_{ik}(z)\right|^2}{\tilde{G}_{kk}(z)} = O((\log N)^{-2\xi}). \tag{171}$$

It then follows from applying this bound to Equation (170) that

$$c \leq \left|\tilde{G}_{ii}^{(\backslash \mathbb{T})}(z)\right| \leq C, \tag{172}$$

proving Equation (165). $\qquad\square$

The following lemma provides initial bounds for when $z$ is far from the real line ($3 \geq \eta \geq 2$) and the resolvent entries are easily controlled. In particular, they will be used to show that the event $\tilde{\Omega}(\eta)$ occurs with high probability and hence our main resolvent estimates are bounded by the function $\Phi(z)$. Hence, using that $\Phi(z), \tilde{G}_{ij}(z)$, and $\tilde{m}(z)$ are all Lipschitz-continuous, the resolvent entries can then be bounded for arbitrarily small $\eta$.

**Lemma D.23.** *Let $\tau < N$ be constant with respect to $N$. For $z = E + \mathrm{i}\eta \in D_L$ and $2 \leq \eta \leq 3$, suppose that $\mathbb{T} \subset [N]$ satisfies $|\mathbb{T}| \leq \tau$. Then with $(\xi, \nu)$-high probability,*

$$c \leq \left|\tilde{G}_{kk}^{(\backslash \mathbb{T})}(z)\right| \leq \frac{1}{\eta}, \qquad \left|\tilde{m}^{(\backslash \mathbb{T})}(z)\right| \leq \frac{1}{\eta}, \qquad \left|\tilde{G}_{ij}^{(\backslash \mathbb{T})}(z)\right| \leq \frac{1}{\eta}, \qquad \text{and} \qquad \max_{i \neq j} \left|\tilde{G}_{ij}^{(\backslash \mathbb{T})}(z)\right| \leq C \tilde{\Lambda}_o(z).$$

*Proof.* We first lower bound $\left|\tilde{G}_{kk}^{(\backslash \mathbb{T})}(z)\right|$. Notice that by taking the imaginary part of the spectral decomposition,

$$\mathrm{Im}\, \tilde{G}_{kk}^{(\backslash \mathbb{T})}(z) = \eta \sum_{\alpha=1}^{N-|\mathbb{T}|} \frac{\left|\hat{v}_\alpha^{(\backslash \mathbb{T})}(k)\right|^2}{(\hat{\lambda}_\alpha - E)^2 + \eta^2} \geq \eta \sum_{\alpha \notin \mathcal{S}} \frac{\left|\hat{v}_\alpha^{(\backslash \mathbb{T})}(k)\right|^2}{(\hat{\lambda}_\alpha - E)^2 + \eta^2} \tag{173}$$

where $\mathcal{S} = \{N - r + 1 - |\mathbb{T}|, \dots, N - |\mathbb{T}|\}$. Now, by Lemma 4.3 from Erdős et al. (2013a) and Lemma D.25, with $(\xi, \nu)$-high probability, we have $\hat{\lambda}_\alpha \leq \|\mathbf{H}\| \leq C$ for all $\alpha \in \mathcal{S}$. Moreover, $E \leq 3$ by definition of $D_L$, so that $(\hat{\lambda}_\alpha - E)^2 \leq C$ for all $\alpha \notin \mathcal{S}$ with $(\xi, \nu)$-high probability. Then since $\eta$ is bounded above and below by constants, we have for some $c > 0$,

$$\mathrm{Im}\, \tilde{G}_{kk}^{(\backslash \mathbb{T})}(z) \geq \eta \sum_{\alpha \notin \mathcal{S}} \frac{\left|\hat{v}_\alpha^{(\backslash \mathbb{T})}(k)\right|^2}{(\hat{\lambda}_\alpha - E)^2 + \eta^2} \geq c \sum_{\alpha \notin \mathcal{S}} \left|\hat{v}_\alpha^{(\backslash \mathbb{T})}(k)\right|^2 \tag{174}$$

with $(\xi, \nu)$-high probability. To lower bound $\sum_{\alpha \notin \mathcal{S}} \left|\hat{v}_\alpha^{(\backslash \mathbb{T})}(k)\right|^2$, notice that since $(\mathbf{B})^{\backslash \mathbb{T}}$ is a real, symmetric matrix, the eigenvectors of $(\mathbf{B})^{\backslash \mathbb{T}}$ form an orthonormal basis. Thus, $\sum_{\alpha=1}^{N-|\mathbb{T}|} \left|\hat{v}_\alpha^{(\backslash \mathbb{T})}(k)\right|^2 = 1$. We note that the eigenvectors of $(\mathbf{B})^{\backslash \mathbb{T}}$

can be used to obtain $N - |\mathbb{T}|$ eigenvectors of $\mathbf{B}^{[\mathbb{T}^c]}$ by filling in zeros at the indices given by $\mathbb{T}$. It follows that for each $\alpha \in \mathcal{S}$, corresponding to the leading eigenvalues of $\mathbf{B}$,

$$\left\| \hat{\mathbf{v}}_{\alpha-|\mathbb{T}|}^{(\backslash\mathbb{T})} (\hat{\mathbf{v}}_{\alpha-|\mathbb{T}|}^{(\backslash\mathbb{T})})^\top - (\mathbf{v}_\alpha)^{\backslash\mathbb{T}} ((\mathbf{v}_\alpha)^{\backslash\mathbb{T}})^\top \right\|_F^2 \le \left\| \hat{\mathbf{v}}_{\alpha-|\mathbb{T}|}^{(\backslash\mathbb{T})} (\hat{\mathbf{v}}_{\alpha-|\mathbb{T}|}^{(\backslash\mathbb{T})})^\top - (\mathbf{v}_\alpha)^{\backslash\mathbb{T}} ((\mathbf{v}_\alpha)^{\backslash\mathbb{T}})^\top \right\|_F^2 + \left\| \mathbf{v}_\alpha^{[\mathbb{T}]} (\mathbf{v}_\alpha^{[\mathbb{T}]})^\top \right\|_F^2$$

$$= \left\| \hat{\mathbf{v}}_\alpha^{[\mathbb{T}^c]} (\hat{\mathbf{v}}_\alpha^{[\mathbb{T}^c]})^\top - \mathbf{v}_\alpha \mathbf{v}_\alpha^\top \right\|_F^2 \tag{175}$$

By Lemma D.5, with $(\xi, \nu)$-high probability,

$$\left\| \hat{\mathbf{v}}_\alpha^{[\mathbb{T}^c]} (\hat{\mathbf{v}}_\alpha^{[\mathbb{T}^c]})^\top - \mathbf{v}_\alpha \mathbf{v}_\alpha^\top \right\|_F^2 = 2 \left\| \left( \mathbf{I} - \hat{\mathbf{v}}_\alpha^{[\mathbb{T}^c]} (\hat{\mathbf{v}}_\alpha^{[\mathbb{T}^c]})^\top \right)^\top \mathbf{v}_\alpha \right\|_F^2 \le \frac{C}{N}.$$

Applying this to Equation (175),

$$\left\| \hat{\mathbf{v}}_{\alpha-|\mathbb{T}|}^{(\backslash\mathbb{T})} (\hat{\mathbf{v}}_{\alpha-|\mathbb{T}|}^{(\backslash\mathbb{T})})^\top - (\mathbf{v}_\alpha)^{\backslash\mathbb{T}} ((\mathbf{v}_\alpha)^{\backslash\mathbb{T}})^\top \right\|_F^2 \le \frac{C}{N} \tag{176}$$

with $(\xi, \nu)$-high probability. In particular, the $(i, j) \in ([N] \setminus \mathbb{T})^2$ entries of $\hat{\mathbf{v}}_\alpha^{(\backslash\mathbb{T})} (\hat{\mathbf{v}}_\alpha^{(\backslash\mathbb{T})})^\top - (\mathbf{v}_\alpha)^{\backslash\mathbb{T}} ((\mathbf{v}_\alpha)^{\backslash\mathbb{T}})^\top$ are given by $\hat{v}_{i\alpha} \hat{v}_{j\alpha} - v_{i\alpha} v_{j\alpha}$, and thus

$$\sum_{i,j\in[N]\setminus\mathbb{T}} (\hat{v}_{i\alpha} \hat{v}_{j\alpha} - v_{i\alpha} v_{j\alpha})^2 = \left\| \hat{\mathbf{v}}_\alpha^{(\backslash\mathbb{T})} (\hat{\mathbf{v}}_\alpha^{(\backslash\mathbb{T})})^\top - (\mathbf{v}_\alpha)^{\backslash\mathbb{T}} ((\mathbf{v}_\alpha)^{\backslash\mathbb{T}})^\top \right\|_F^2.$$

Using the trivial upper bound

$$\sum_{j\in[N]\setminus\mathbb{T}} (\hat{v}_{j\alpha} \hat{v}_{j\alpha} - v_{j\alpha} v_{j\alpha})^2 \le \sum_{i,j\in[N]\setminus\mathbb{T}} (\hat{v}_{i\alpha} \hat{v}_{j\alpha} - v_{i\alpha} v_{j\alpha})^2,$$

we have for $k \in [N] \setminus \mathbb{T}$

$$(\hat{v}_{k\alpha}^2 - v_{k\alpha}^2)^2 \le \sum_{i,j\in[N]\setminus\mathbb{T}} (\hat{v}_{i\alpha} \hat{v}_{j\alpha} - v_{i\alpha} v_{j\alpha})^2.$$

Hence, applying this bound to Equation (176) we have with $(\xi, \nu)$-high probability

$$(\hat{v}_{k\alpha}^2 - v_{k\alpha}^2)^2 \le \frac{C}{N}.$$

Taking square roots, we have that with $(\xi, \nu)$-high probability,

$$\left| \hat{v}_{k\alpha}^2 - v_{k\alpha}^2 \right| \le \frac{C}{\sqrt{N}}. \tag{177}$$

By Assumption (Ã1),

$$\hat{v}_{k\alpha}^2 \le \frac{C}{\sqrt{N}} + v_{k\alpha}^2 \le \frac{C}{\sqrt{N}} + \frac{C(\log N)^{2\gamma}}{N},$$

from which

$$\hat{v}_{k\alpha}^2 \le \frac{C}{\sqrt{N}} + v_{k\alpha}^2 \le \frac{C}{\sqrt{N}} + \frac{C(\log N)^{2\gamma}}{N} \le \frac{C}{\sqrt{N}},$$

and after taking square roots, with $(\xi, \nu)$-high probability,

$$|\hat{v}_{k\alpha}| \le \frac{C}{N^{1/4}}. \tag{178}$$

So since the eigenvectors of $\tilde{\mathbf{G}}^{(\backslash\mathbb{T})}(z)$ form an orthogonal matrix,

$$\sum_{\alpha=1}^{N-|\mathbb{T}|} |\hat{v}_{k\alpha}|^2 = 1, \tag{179}$$

we have with $(\xi, \nu)$-high probability,

$$\sum_{\alpha \notin \mathcal{S}} |\hat{v}_{k\alpha}|^2 = 1 - \sum_{\alpha \in \mathcal{S}} |\hat{v}_{k\alpha}|^2 \geq 1 - \frac{Cr}{\sqrt{N}} \geq c \tag{180}$$

for some $c > 0$, sufficiently large $N$, and Assumption ($\tilde{\mathbf{A}}$5). It then follows from Equation (174) that

$$c \leq \operatorname{Im} \tilde{G}_{kk}^{(\backslash \mathbb{T})}(z) \leq \left| \tilde{G}_{kk}^{(\backslash \mathbb{T})}(z) \right|.$$

with $(\xi, \nu)$-high probability.

To upper bound $\left| \tilde{G}_{kk}^{(\backslash \mathbb{T})}(z) \right|$, note that $(\mathbf{B} - z\mathbf{I})^{\backslash \mathbb{T}}$ is normal and its singular values are given by

$$\left\{ \sqrt{(\hat{\lambda}_\alpha^{(\backslash \mathbb{T})} - E)^2 + \eta^2} : \alpha = 1, 2, \ldots, N - |\mathbb{T}| \right\}.$$

Hence, the singular values of $\tilde{\mathbf{G}}^{(\backslash \mathbb{T})} = ((\mathbf{B} - z\mathbf{I})^{\backslash \mathbb{T}})^{-1}$ can be bounded according to

$$\frac{1}{\sqrt{(\hat{\lambda}_\alpha^{(\backslash \mathbb{T})} - E)^2 + \eta^2}} \leq \frac{1}{\eta}.$$

It then follows that

$$\max_{i,j} \left| \tilde{G}_{ij}^{(\backslash \mathbb{T})}(z) \right| \leq \left\| \tilde{\mathbf{G}}^{(\backslash \mathbb{T})}(z) \right\| \leq \frac{1}{\eta}. \tag{181}$$

For the upper bound on $\tilde{m}^{(\backslash \mathbb{T})}(z)$, notice that

$$\left| \tilde{m}^{(\backslash \mathbb{T})}(z) \right| = \left| \frac{1}{N} \sum_i \tilde{G}_{ii}^{(\backslash \mathbb{T})}(z) \right| \leq \frac{1}{N} \sum_i \left| \tilde{G}_{ii}^{(\backslash \mathbb{T})}(z) \right| \leq \frac{1}{\eta}$$

where the inequality follows from Equation (181).

To show the last claim that $\max_{i \neq j} \left| \tilde{G}_{ij}^{(\backslash \mathbb{T})}(z) \right| \leq C \tilde{\Lambda}_o(z)$, observe that from Equation (154) in the proof of Lemma D.19 and Equation (181) and recalling $1/\eta \leq 1/2$, we have

$$\max_{i \neq j} \left| \tilde{G}_{ij}^{(\backslash k)} \right| (z) = \max_{i \neq j} \left| -\tilde{G}_{ij}(z) + \frac{\tilde{G}_{ik}(z)\tilde{G}_{kj}(z)}{\tilde{G}_{kk}(z)} \right| \leq \max_{i \neq j} \left| \tilde{G}_{ij}(z) \right| + \max_{i \neq j} \frac{\left| \tilde{G}_{ik}(z) \right| \left| \tilde{G}_{kj}(z) \right|}{\left| \tilde{G}_{kk}(z) \right|}$$
$$\leq \max_{i \neq j} \left| \tilde{G}_{ij}(z) \right| + C \max_{i \neq j} \left| \tilde{G}_{ij}(z) \right|^2. \tag{182}$$

Then take $\mathbb{T} = \emptyset$ in Equation (181), so we have

$$\max_{i \neq j} \left| \tilde{G}_{ij}(z) \right| + C \max_{i \neq j} \left| \tilde{G}_{ij}(z) \right|^2 = \max_{i \neq j} \left| \tilde{G}_{ij}(z) \right| (1 + C \max_{i \neq j} \left| \tilde{G}_{ij}(z) \right|) \leq \max_{i \neq j} \left| \tilde{G}_{ij}(z) \right| \left( 1 + \frac{C}{\eta} \right), \tag{183}$$

it follows that

$$\max_{i \neq j} \left| \tilde{G}_{ij}^{(\backslash k)}(z) \right| \leq C \max_{i \neq j} \left| \tilde{G}_{ij}(z) \right|. \tag{184}$$

The claim then follows by the same argument as given in the proof of Lemma D.19 for the remaining $k' \in \mathbb{T} \setminus \{k\}$ by replacing $\tilde{\mathbf{G}}$ with $\tilde{\mathbf{G}}^{(\backslash k)}$ in Equation (182) and repeating the above. $\qquad \square$

**Lemma D.24.** *Let $\mathbf{B}$ be as in Equation (36). We have with $(\xi, \nu)$-high-probability,*

$$\max_{i,j} |B_{ij}| \leq C \frac{r \rho_N (\log N)^{2\gamma}}{\sqrt{N}} + \frac{C}{q}.$$

*Proof.* Note we can decompose any entry $B_{ij}$ as

$$B_{ij} = L_{ij} + H_{ij}. \tag{185}$$

We can further decompose $L_{ij}$ by recalling Equation (53)

$$\mathbf{L} = \sum_{\alpha=N-r+1}^{N} \lambda_\alpha \mathbf{v}_\alpha \mathbf{v}_\alpha^\top.$$

By Assumption ($\tilde{\mathbf{A}}4$), $c\sqrt{N} \leq \lambda_\alpha \leq C\sqrt{N}$ for all $\alpha \in \{N-r+1, N-r+2, \ldots, N\}$. So then for each $i, j \in [N]$, we have

$$L_{ij} = \left( \sum_{\alpha=N-r+1}^{N} \lambda_\alpha \mathbf{v}_\alpha \mathbf{v}_\alpha^\top \right)_{ij} \leq C\sqrt{N} \left( \sum_{\alpha=N-r+1}^{N} \mathbf{v}_\alpha \mathbf{v}_\alpha^\top \right)_{ij}.$$

By Assumption ($\tilde{\mathbf{A}}1$), $\mathbf{v}_i \mathbf{v}_i^\top$ has entries of order $1/N$, whence

$$L_{ij} = O\left( \frac{r\rho_N (\log N)^{2\gamma}}{\sqrt{N}} \right).$$

Using this bound for $L_{ij}$ and controlling the entries of $\mathbf{H}$ with Lemma D.20, applying the triangle inequality to Equation (185) yields

$$\max_{i,j} |B_{ij}| \leq \max_{i,j} |L_{ij}| + \max_{i,j} |H_{ij}| \leq C \frac{r\rho_N (\log N)^{2\gamma}}{\sqrt{N}} + \frac{C}{q}$$

with $(\xi, \nu)$-high-probability, as we set out to show. $\qquad\square$

**Lemma D.25** (Weyl's Inequality; Theorem 4.3.1 in Horn & Johnson (1985))**.** *Let* $\mathbf{A}, \mathbf{B}$ *be Hermitian whose eigenvalues are ordered as* $\lambda_{\min} = \lambda_1 \leq \lambda_2 \leq \cdots \leq \lambda_N = \lambda_{\max}$. *Then*

$$\lambda_i(\mathbf{A} + \mathbf{B}) \leq \lambda_{i+j}(\mathbf{A}) + \lambda_{N-j}(\mathbf{B}) \tag{186}$$

*where* $j = 0, \ldots N - i$ *for each* $i \in [N]$. *Similarly,*

$$\lambda_{i-j+1}(\mathbf{A}) + \lambda_j(\mathbf{B}) \leq \lambda_i(\mathbf{A} + \mathbf{B}) \tag{187}$$

*where* $j = 1, 2, \ldots, i$ *for each* $i \in [N]$.

*Further, if* $\mathbf{A}, \mathbf{B}$ *are Hermitian matrices, and* $\mathbf{B}$ *is rank* $r$. *Then*

$$\lambda_i(\mathbf{A}) \leq \lambda_i(\mathbf{A} + \mathbf{B}) \leq \lambda_{i+r}(\mathbf{A}) \quad \text{for } 1 \leq i \leq N - r.$$

