# OpenReview forum: "On the Effect of Misspecifying the Embedding Dimension in Low-rank Network Models"
_ICML.cc/2026/Conference — ICML 2026 regular_

### Official Review · Reviewer_Z8ps · 2026-03-10

**Soundness:** 4
**Presentation:** 3
**Significance:** 3
**Originality:** 3
**Overall Recommendation:** 5
**Confidence:** 2

**Summary:**

The paper provides new results for the consistency of spectral embedding of random dot product graphs (RDPGs). In particular, their main result (Theorem 3.3) is that, under several assumptions, spectral embedding with a mismatched dimension, that is higher than the true dimension of the RDPG, is bounded above in error by a term that scales positively with both the extent of the mismatch and the true dimension, and inversely with the number of nodes. The proof technique involves showing that the adjacency eigenvectors are delocalized (its entries are bounded), and the means of showing this requires a strong assumption regarding the variance of the noise matrix. Besides this main result, the paper also shows that the error when the used dimension is lower than the true dimension is bounded below based on the mismatch. Further, it provides some experiments on a certain class of RDPGs doing spectral embedding with a range of used dimensionalities, to validate the theoretical results.

**Compliance With Llm Reviewing Policy:**

Affirmed.

**Final Justification:**

The paper offers a clear, novel proof for a theoretical result that is core to graph modeling. The strong assumptions required for this proof notwithstanding, this work seems of sufficient significance to maintain an accept rating.

**Key Questions For Authors:**

1. Can A3 be relaxed with just some hairier notation, or is there any other obstacle?
2. Would it help validate the results empirically to run experiments with an embedding distribution that produces more heterogeneous degree distributions?
3. Models often use a nonlinear link function $p_{ij} = \sigma(x_i \cdot x_j)$, e.g., a logistic function, which can make the model more powerful. Can the techniques here help provide similar results for nonlinear models?

**Limitations:**

Yes, the authors clearly define the scope of the work and discuss the assumptions required for the proof.

**Strengths And Weaknesses:**

# Strengths
- The core result is quite fundamental and of high interest to the node embedding area.
- The proof technique involves what seems to be a novel application of random matrix theory techniques to network modeling.
- The writing is clear and easy to follow, with well-specified notation.

# Weaknesses
- As the authors acknowledge, the core result requires some strong assumptions, including A7 that is not typical to network modeling work but apparently common in random matrix theory. The assumptions are mostly justified in the writing, but difficult to interpret for someone outside the area.
- This seems like a weak reason for not pursuing the proof without the assumption in this work: "We expect that Assumption A3 can be relaxed at the expense of increased notational complexity in tracking the eigenspaces corresponding to repeated eigenvalues in the proof of Theorem 3.1, a task we leave to future work."
- The thrust of the empirical section seems to be mostly to validate the theoretical results, but it would help to use experiments to poke beyond the assumptions. E.g., A7 is perhaps most clearly violated in graphs with heterogeneous degrees, but the experiments only test a distribution that produces concentrated node degrees. One other limitation is that only the ground truth dimension 5 is tested.

---

> ### Author Rebuttal · Authors · 2026-03-30
>
> 1) Assumption (A3) can be relaxed with strictly hairier notation. Specifically, Lemmas D.9 through D.11 would need to be modified. Currently, the distinct eigenvalue assumption allows us to assume that the $j$-th signal eigenvector is close to the $j$-th sample eigenvector. This is no longer true when the eigenvalues are repeated. Instead, we can only say that the $j$-th population eigenspace is close to the $j$-th sample eigenspace; however, one would need to keep track of the multiplicity of the repeated eigenvalues when working with the spectral decomposition of the resolvent $G$. To further validate the relaxation of Assumption (A3), we conducted an experiment in which the expectation matrix $P$ is an $n\times n$ low-rank matrix with repeated eigenvalues. In particular, we obtained $P = USU^\top$, where $S$ is a $r\times r$ diagonal matrix with $n/10$ along its diagonals, and $U \in \mathbb{R}^{n \times r}$ is an orthogonal matrix sampled from the Haar distribution on the Stiefel manifold. We then obtain $A = P + E$ with $E_{ij} \sim \mathcal{N}(0, 0.1^2)$ and computed its d-dimensional ASE and the resulting estimation error of the latent positions in $(2,\infty)$-norm. Despite the repeated eigenvalues, our results are still concordant behavior implied by our Theorem 3.3 and similar experiments (see Figure 1 in the manuscript). Should our work be accepted, we will include discussion of these proof details and results of the above experiment in the camera-ready version.
>
> 2) We agree with the reviewer that an experiment in the heterogeneous degree setting would improve the paper. To this end, we considered estimating the latent positions of a degree-corrected stochastic blockmodel with $10$ communities whose degree-correction parameters, $\theta_i$, were sampled i.i.d from a $\operatorname{Pareto}(\alpha = 5, \beta = 4/5)$ distribution. The intra-community connectivity probability was set to $p=0.6$, and the between-community probability set to $q=(0.6)*(3/10)$. Thus, the probability of an edge between vertices $i$ and $j$ belonging to the same community is $\theta_i \theta_j p$, while the probability of an edge when $i$ and $j$ belong to different communities is $\theta_i \theta_j q$.
> The results are in-line with the SBM experiment presented in Appendix A.2: the convergence rate of the estimation error is quickest, at $n^{-1/2}$, when the embedding dimension is correctly chosen; it exhibits the slower $n^{-1/4}$ rate when the dimension is chosen too large ($d = 11$), and it is inconsistent when chosen too small ($d = 9$). Should the paper be accepted, we will include the details and findings of this experiment in the final publication.
>
> 3) The techniques used in this paper to show delocalization are applicable as long as the nonlinear link function induces a signal matrix $P = \mathbb{E}[ A \mid X]$ that satisfies the assumptions introduced in Section 2 (i.e., low-rank, delocalized leading eigenvectors, etc.). In light of this, the nonlinear link function would need to be sufficiently smooth and induce an approximately low-rank structure in $P$ for our results to hold. We would need either that $P$ still have a low-rank structure (unlikely unless $X$ and the link function are very carefully chosen) or that $P$ have an approximately low-rank structure, in which its trailing eigenvalues are decaying fast enough that we can absorb them into the error term (note that similar ideas have been deployed successfully in "Network representation using graph root distributions" by Jing Lei, 2021 in Annals of Statistics). We will include a discussion of this idea in the camera-ready version of the paper. We also investigated this idea specifically in the setting of the logistic link function suggested by the reviewer in a small preliminary experiment. We generated latent positions i.i.d.~from a Dirichlet distribution, and constructed expected adjacency matrix $P = [P_{ij}] = \sigma(x_i^\top x_j)$, where $\sigma(z) = 1/(1 + \exp\\{-(z - 0.5)/h)\\})$ (note that the offset term $0.5$ is necessary to correctly center the argument). We then computed the estimation error in $(2, \infty)$-norm between the $d$-dimensional ASE of $P$ and the matrix of latent positions. We find that provided that the bandwidth scales at the correct rate, the behavior of the (misspecified) ASE is similar to the behavior guaranteed by our existing theory (i.e., our theory for when the link function is the identity). That is, even when a nonlinear transformation is applied to the entires of $XX^\top$, the $d$-dimensional ASEs of $XX^\top$ and $\sigma(XX^\top)$ behave similarly. Provided our work is accepted for publication, we will include a remark on the extension to nonlinear link functions and a larger-scale version of the experiment described above.

---

> > ### Author Rebuttal · Reviewer_Z8ps · 2026-04-04
> >
> > I thank the authors for their response. The experiment in (3) is indeed of interest, and could be a good addition to the experimental section, along the same lines of toy experiments going beyond proved results. I have no further concerns.

---

### Official Review · Reviewer_MLX6 · 2026-03-12

**Soundness:** 3
**Presentation:** 3
**Significance:** 3
**Originality:** 3
**Overall Recommendation:** 5
**Confidence:** 4

**Summary:**

This paper studies the effect of embedding-dimension misspecification in adjacency spectral embedding (ASE) for low-rank latent network models, with an emphasis on nodewise uniform error measured by the $(2,\infty)$-norm. While much prior theory assumes the embedding dimension matches the true latent rank $r$, the authors analyze both under-embedding and over-embedding regimes. The main message is a clear bias--variance tradeoff: when $d<r$, ASE can be inconsistent due to irreducible truncation bias; when $d\ge r$, ASE remains consistent, but choosing $d>r$ induces a slower convergence rate. A central technical contribution is an eigenvector delocalization result for the ``non-signal'' eigenvectors of the observed matrix, obtained via a local-law approach, which underpins the $d>r$ bounds. The experiments (weighted and binary settings) are well aligned with the theoretical predictions: $d=r$ exhibits approximately $n^{-1/2}$ behavior, $d>r$ shows approximately $n^{-1/4}$ behavior, and $d<r$ trends toward an error floor; fixed-$n$ sweeps over $d$ also qualitatively match the predicted $\sqrt{d-r}$-type degradation.

**Compliance With Llm Reviewing Policy:**

Affirmed.

**Final Justification:**

The paper provides a rigorous theoretical analysis of embedding-dimension misspecification in adjacency spectral embedding, revealing a clear bias–variance tradeoff. Overall, the work is sound, original, and significant for practitioners using spectral methods. We therefore maintain a score of 5 (accept).

**Key Questions For Authors:**

(1) Binary networks (proved vs conjectured). Please clearly delineate which results are rigorously proved for the weighted/continuous-noise setting versus conjectured for Bernoulli adjacency. Are any partial binary results provable under additional regularity conditions (e.g., near-constant variance profile), even if the most general case remains open?

(2) Conditions for binary RDPG. For the conjectured binary extension, can the authors provide concrete, checkable sufficient conditions (e.g., constraints on $P_{ij}(1-P_{ij})$, sparsity scaling $\rho_n$, and allowable degree heterogeneity) under which eigenvector delocalization is expected to hold, and how these relate to modern local-law results for general variance profiles (e.g., Ajanki et al.)?

(3) Dependence on $d-r$ and tightness. The theory suggests a $\sqrt{d-r}$-type degradation for fixed $n$ when $d>r$, while the experiments in Figure~2 appear slightly slower. Is this discrepancy primarily due to constants/proof looseness, the Procrustes alignment step, or finite-sample effects? Any brief diagnostic clarifying this would help interpret the tightness of the $d-r$ dependence.

(4) Under-embedding in sparse regimes. In the binary experiments, inconsistency for $d<r$ is clear in denser regimes, while behavior at higher sparsity levels is less conclusive at the largest simulated $n$. Could the authors provide a lightweight additional check (e.g., a few larger-$n$ points on a subset of sparsity levels, or a quantitative extrapolation) to better support the claim that the error levels off for larger sparsity as well?

(5) Practical guidance for choosing $d$. Given the derived tradeoff (irreducible bias for $d<r$ versus slower rates for $d>r$), can the authors provide a brief rule-of-thumb for choosing $d$ in practice (even heuristic), especially as a function of sample size and sparsity?

**Limitations:**

yes

**Strengths And Weaknesses:**

Strengths:

(1) The paper addresses an important and practically relevant question: the consequences of choosing an embedding dimension $d$ that does not match the true latent rank $r$ in spectral embeddings.

(2) The technical contribution is strong: trailing-eigenvector delocalization via a local-law approach, which is central to controlling over-embedding ($d>r$) in $(2,\infty)$-norm.

(3) The empirical section is carefully designed to validate the theory, covering both rate-versus-$n$ behavior under correct/over/under embedding, as well as fixed-$n$ dependence on $d$.

(4) The results provide an actionable qualitative takeaway: under-embedding can be fundamentally biased, while over-embedding is comparatively safe but slower.

Weaknesses:

(1) The main delocalization proof requires structural conditions on the noise variance profile (e.g., a doubly stochastic variance matrix) that Bernoulli adjacency matrices in binary RDPG settings do not generally satisfy. The binary extension is therefore framed as a conjecture supported by simulations.

(2) Several spectral assumptions (incoherence/delocalization of the signal eigenvectors, signal strength, and eigengap conditions) may be restrictive in very sparse or highly degree-heterogeneous regimes; a clearer discussion of when these assumptions hold in common RDPG/SBM settings would strengthen applicability.

---

> ### Author Rebuttal · Authors · 2026-03-30
>
> 1) No binary results (even partial) are currently proved in this paper, even with near-constant variance profile. The only papers of which we are aware that provide results for this setting apply only to the bulk of the spectrum, which is not enough for our purposes here. The random matrix theory machinery used here assumes that the diagonal entries of the resolvent approach the Stieltjes transform of the semicircle law, which typically relies on the matrix of entry-wise variances of $A$ being doubly stochastic. Conjecture 1 stated in Section 3.1 is entirely a conjecture (though supported by experiments; see our answer to Question 2 below). We will restructure the last parts of Section 3 to make this point more clear. See our reponse to Question 2 below for discussion of the challenges in a potential proof for binary networks.
>
> 2)  As stated in Conjecture 1, we suspect that as long as the expected probability matrix, $P$, satisfies assumptions A1 -- A6, and the noise distribution still satisfies assumption A7 (with the doubly stochastic condition relaxed), we expect delocalization to hold. In preliminary experiments, we found that eigenvector delocalization for binary matrices appeared to hold for quite sparse settings. Indeed, in experiments that were not included in the submitted version, when the sparsity parameter was set to $\rho_n = (\log n)^{\gamma}/n$, delocalization held for $\gamma > 3$. Unfortunately, the theory for delocalization for general variance profiles has not yet allowed for sparse binary matrices whose (expected) rank is greater than one. The work by Ajanki et al. for Wigner-type matrices assumes that the noise distribution obeys certain moment bounds, which sparse binary matrices do not satisfy in general. We will include the delocalization experiments for binary matrices described above, along with a more in-depth discussion of how Ajanki et al.'s work relates to Conjecture 1.
>
> 3) As mentioned at the bottom of the first column on page 7, we suspect the observed slower growth is due to a combination of Procrustes alignment and finite-sample effects. This is borne out, first and foremost, by the fact that these under-estimation issues do not appear in the other plots, where the embedding dimension is not growing. More generally, when $d>r$, the alignment of $\hat X_{1:d}$ to $X_{1:d}$ involves aligning the trailing $d-r$ columns of $\hat X_{1:d}$ to the $d-r$ columns of zeros that were padded to the end of $X_{1:r}$ to form $X_{1:d}$. The alignment matrix $Q$ in Equation (13) is the minimizer of a Frobenius norm, rather than minimizing the $(2,\infty)$-norm directly. As the embedding dimension grows, the $d-r$ extra dimensions of $\hat{X}$ are essentially noise, and it is comparatively easy to align that noise with the zero vector so that some of the error disappears, at least in $(2,\infty)$-norm (but not in Frobenius norm). This discrepancy between Frobenius norm and $(2,\infty)$-norm has been discussed elsewhere. See, for example, "Orthogonal Procrustes and norm-dependent optimality" by Joshua Cape in Electronic Journal of Linear Algebra. We will add a more substantial discussion of this point to the first column of page 7.
>
> 4) By "leveling out", we meant that the curves "stabilize" for larger values of $n$. The slope of the curves for differing sparsity levels appear to be in flux at smaller values of $n$, until they reach their asymptotic rate for some $n$ that appears to depend on the sparsity value. Should the paper be accepted, we will clarify our language in the discussion of Figure 4 and we will include additional experiments with larger values of $n$ for sparse binary matrices.
>
> 5) As mentioned in the last paragraph of column 1 on page 2 and again in Remark 3.2, the focus of this paper is on characterizing the consequences of model misspecification, rather than developing new model selection methods per se. It is unlikely that a heuristic in terms of sample size and sparsity alone will be sufficient for model selection: existing model selection methods that we are aware of typically require more involved approaches based on sequential hypothesis testing, cross-validation and other methods that we alluded to in the middle paragraph in first column of page 2. That being said, our results do have implications for how practitioners should choose the embedding dimension, regardless of what specific model selection method they are using. In short, our results suggest erring on the side of larger embedding dimensions. This may mean, for example, that when a model selection method identifies more than one choice of $d$ to be comparably good, practitioners should choose the larger value of $d$. To this end, should the paper be accepted, we will include in the discussion section further treatment of the implications for model selection of the ASE's behavior under misspecification.

---

> > ### Author Rebuttal · Reviewer_MLX6 · 2026-04-02
> >
> > I thank the authors for addressing the concerns well; I have no further comments.

---

### Official Review · Reviewer_ANZL · 2026-03-12

**Soundness:** 3
**Presentation:** 3
**Significance:** 2
**Originality:** 3
**Overall Recommendation:** 5
**Confidence:** 4

**Summary:**

This paper provides a theoretical treatment of dimension misspecification in Adjacency Spectral Embedding (ASE) for Random Dot Product Graphs (RDPG). Specifically, it investigates the behavior of the estimated latent positions when the chosen embedding dimension d differs from the true rank of the underlying probability matrix r

**Compliance With Llm Reviewing Policy:**

Affirmed.

**Final Justification:**

The rebuttal addressed my concerns adequately ad including these points will strengthen the paper, in that they emphasize the utility of the results beyond the specific technical setup considered.
Hence, I have updated and increased my score.

**Key Questions For Authors:**

Addressing any of the three identified weaknesses above would strengthen the paper and lead me to increase the score.

**Limitations:**

yes

**Strengths And Weaknesses:**

**Strengths**

* The mathematical analysis is detailed and sound as far as I can tell
* The result is interesting and a useful as a "sanity check" for practitioners. (it could be argued, however, that the result that over-estimating the dimension is safer than under-estimating it, is quite unsurprising).
* The paper is written in a clear and convincing way

**Weaknesses**
1. The paper describes the consequences of misspecification but offers no new methodology to avoid it. It does not provide a new estimator for r nor does it propose a more robust embedding procedure. The lack of an algorithmic contribution or a "fix" for the slower convergence rates noted in the d>r case makes the paper feel somewhat descriptive. The paper could be strenghten significantly if the authors could expand on this further
2. The analysis is specific to the RDPG framework in conjunction with adjacency matrix based embeddings. While RDPGs are flexible, they assume a very specific dot-product structure. Can the authors comment on how these results could be extended to other random graph models? Even some numerical experiments in this direction could be of value and would strenghten the paper.
3. The authors note that over-specification leads to a "possibly slower rate" of convergence. However, the paper lacks a characterization of this trade-off. Without knowing how much slower the rate is, it is difficult to judge the cost of over-specification in finite-sample regimes. In practice, a "consistent" estimator that converges very slowly may be indistinguishable from an inconsistent estimator. Can the authors expand on these aspects?

---

> ### Author Rebuttal · Authors · 2026-03-30
>
> 1) The reviewer is correct that our paper does not provide a new estimator for $r$. As mentioned in the last paragraph of col. 1 on page 2 and in Remark 3.2, our focus is on the consequences of model misspecification, rather than developing new model selection methods per se. That is, our results are concerned with the behavior of an estimator already in wide use, for which there exist many model selection methods. As stated in the Introduction, none of these existing methods have guarantees about what happens when they make the wrong choice of dimension, which occurs with non-zero probability for any finite-sized network. Our results *do* have implications for how to choose the embedding dimension, regardless of what specific model selection method is used. Theorem 3.3 serves as an addendum to *any* model selection method. If a method identifies two or more embedding dimensions as comparably good, our result provides concrete guidance as to how to "break the tie": err on the side of choosing the larger dimension. The slower convergence rate when $d>r$ cannot be "fixed" except via better model selection: this slow rate is inherent to the ASE when the dimension is chosen too large. We will add more discussion of these points in the final paper, if it is accepted.
>
> 2) We discuss the RDPG in Sections 2 and 3 because we can prove that it satisfies the assumptions needed for the random matrix theory machinery to work (i.e., the assumptions listed at the end of Section 2). Our results are applicable to a wider class of models: any model that induces a signal matrix that satisfies Assumptions A1 through A6 (i.e., low-rank, delocalized leading eigenvectors, etc.) and any noise matrix whose entries exhibit tail behavior satisfying A7. Extensions beyond low-rank models is interesting. One possible avenue, suggested by reviewer Z8ps, is to consider models like $P = g(X_i,X_j)$, where $g$ is a kernel function (reviewer Z8ps suggests the choice $g(y,z)=\sigma(y^\top z)$, the logistic link). The techniques used to show delocalization in this paper apply so long as $g$ induces a signal matrix $P = \mathbb{E}[A\mid X]$ satisfying A1 through A7 in Section 2 (i.e., low-rank, delocalized leading eigenvectors, etc.). We expect that our results can be extended to this setting so long as the kernel $g$ is sufficiently smooth. We would need either that $P$ still have low-rank structure (unlikely unless $X$ and $g$ are very carefully chosen) or that $P$ have an approximately low-rank structure, in which its trailing eigenvalues decay fast enough that we can absorb them into the error term (similar ideas have been used in "Network representation using graph root distributions" by J. Lei, 2021 in Annals of Statistics). We will add a discussion of this to the paper. In a small preliminary experiment, we investigated this idea under the logistic link suggested by reviewer Z8ps. We generated latent positions i.i.d. from a Dirichlet distribution, and constructed $P=[P_{ij}]=\sigma(x_i^\top x_j)$, where $\sigma(z)=1/(1+\exp\\{-(z-0.5)/h\\})$ (note that the offset $0.5$ is necessary to correctly center the argument). We then computed the estimation error in $(2,\infty)$-norm between the $d$-dimensional ASE of $P$ and $X$ (after Procrustes alignment). We find that, if the bandwidth $h$ grows at the correct rate, the behavior of the (misspecified) ASE is similar to the behavior guaranteed by our theory (i.e., when the link function is the identity). That is, even when a nonlinear transformation is applied to the entires of $XX^\top$, the $d$-dimensional ASEs of $XX^\top$ and $\sigma(XX^\top)$ behave similarly. If the paper is accepted, we will include a remark on the extension to nonlinear link functions and a larger-scale version of the experiment described above. Extending our results further to, say, the preferential attachment model or configuration models is far more challenging, since they do not typically have a low-rank expected adjacency matrix. We will add discussion of these broader model classes to the final version if the paper is accepted.
>
> 3) Our results *do* characterize the slower rate of convergence: Equation (11) in Theorem 3.3 states our upper bound, and includes dependence on the edge noise parameter $\sigma^2$, the number of nodes $n$, the true rank $r$ and the "amount of misspecification" $d-r$. In particular, our result characterizes the slower rate when $d>r$ as growing with the square root of $d-r$ and with the reciprocal of the fourth root of the number of nodes $n$. So long as $d$ is not too much larger than $r$, we may still expect convergence, albeit at a rate of $n^{-1/4}$ rather than the parametric rate $n^{-1/2}$. Our experiments show that this convergence is distinguishable from inconsistency as $n$ grows. For example, compare the slopes in the subplots of Figure 3. Should the paper be accepted, we will add discussion of this point after the statement of Theorem 3.3 to improve the clarity of the paper.

---

> > ### Author Rebuttal · Reviewer_ANZL · 2026-04-01
> >
> > I thank the authors for their response. I think adding these aspects will strengthen the paper. I will increase my score.

---

### Official Review · Reviewer_fMQq · 2026-03-13

**Soundness:** 3
**Presentation:** 1
**Significance:** 2
**Originality:** 3
**Overall Recommendation:** 3
**Confidence:** 3

**Summary:**

This paper considers how the choice of the embedding dimension impacts the estimation of the latent positions. In particular, the authors consider the signal-plus noise matrix model A = P + E such as the RDPG. The results  characterize how different choices of embedding dimension affect the estimation rate of Adjacency Spectral Embedding, measured in terms of (2, ∞)-norm.  Theorem 3.3 and the simulation results show that under a weighted version of the RDPG, when the embedding dimension is chosen too small, consistency in the (2, ∞)-norm is not guaranteed for the ASE. When the embedding dimension is correctly specified, all signal present in the network is captured and the optimal convergence rate is achieved.  On the other hand,  when the embedding dimension is chosen too large, consistency may still hold, as long as the embedding dimension is not too much larger than the true dimension.

**Compliance With Llm Reviewing Policy:**

Affirmed.

**Final Justification:**

After reading the authors' response to my questions, I have a better understanding of their contributions. I updated my recommendation accordingly. I still think the paper's organization can be improved and the discussion section can be expanded using some of the responses provided in the rebuttal.

**Key Questions For Authors:**

1) What are some implications of the theoretical results for downstream learning tasks on networks?
2) Are there any limitations of the presented theorems? For example, are the assumptions realistic for real networks?
3) How can these results be extended for different latent models?

**Limitations:**

The results are presented in a limited manner. I haven't found any explicit discussion of the limitations of the proposed methods for estimating consistency. Are the presented models realistic? How robust are the results to perturbations to this model?

**Strengths And Weaknesses:**

Soundness: The paper is technically sound. The assumptions in the theorems seem reasonable and closely follow exsiting literature on consistency derivation for embeddings. The experimental section could be better organized with a logical flow of the experiments and a better description. The experiments make sense and are used to verify the theorems.
Presentation:  The presentation of the paper can be improved. The organization of the paper should be improved with a dedicated section on notation and preliminaries on graphs and latent embedding models. The notations are not always well-defined, making it hard to follow the theoretical results. For example, it is not clear when \hat is used on the symbols as it is not explicitly spelled out. Similarly, it is not clear why absolute value of the eigenvalues is used in the ASE definition since for undirected weighted or binary graphs, the eigenvalues should always be non-negative. The figures are hard to read as there are no legends provided for the different curves.
Significance: While the addressed problem of investigating what happens with the choice of the embedding dimension is interesting, the paper fails to provide any broader impacts or applications to other fields of machine learning. For example, how would these results influence any downstream tasks on networks?
Originality: I think the results presented in the paper are original. The methods used in the proofs may not be completely novel as similar tools have been used to prove the consistency of ASE when the dimension is correct.

---

> ### Author Rebuttal · Authors · 2026-03-30
>
> __"The experimental section could be better organized [...] ."__
>
> Our experiment section has two subsections, corresponding to weighted and binary networks, each with two experiments. We will add more explicit signposting of the experiments.
>
> __"The organization of the paper should be improved with a dedicated section on notation and preliminaries on graphs and latent embedding models."__
>
> Notation is in the last paragraph of Section 1 (middle of col. 2 on page 2) . We will add a header to explicitly mark this. We will add more background material.
>
> __"The notations are not always well-defined [...]. For example, it is not clear when \hat is used on the symbols as it is not explicitly spelled out."__
>
> "hat" notations are in Definition 2.1 and at the bottom of col. 2 of page 4. We will add that this notation is meant to evoke that $\hat{X}$ estimates $X$ (and $\hat{U}$ estimates $U$, etc.). Beyond this, we are unsure which notations the reviewer finds unclear. We will make a thorough pass through the document to ensure that every symbol is explained when introduced.
>
> __"It is not clear why absolute value of the eigenvalues is used in the ASE definition since for undirected weighted or binary graphs, the eigenvalues should always be non-negative."__
>
> While the population eigenvalues in $S_{1:r}$ (see Eq. (2)) are positive, the leading eigenvalues of $A = P + E$ need not be positive: depending on $E$, some eigenvalues of $A$ may be negative. As $n$ grows, so long as $E$ is nice, the leading $r$ eigenvalues of $A$ will be positive, but eigenvalues beyond $r$ need not all be positive, regardless of $n$. We include the absolute value on $\hat{S}$ to handle this. We will make this explicit in the text.
>
> __"figures are hard to read as there are no legends provided [...] ."__
>
> We are not sure which figure(s) the reviewer means. All eight figures have legends at the bottom showing how line colors relate to parameters. All figures have captions explaining the layout and are discussed in detail in the main text.
>
> __"[...], the paper fails to provide any broader impacts or applications [...]. For example, how would these results influence any downstream tasks on networks?"__
>
> As discussed at the top of col. 1 on page 2, embeddings are used in many downstream tasks. We agree that the implications for these tasks are important. Our focus here is on performance of embeddings as estimators. These results are necessary before understanding their downstream implications. Fitting a treatment of both our estimation results and their downstream applications is a lot to fit into eight pages while maintaining clarity. If the paper is accepted, we will include mention of the downstream implications of Theorem 3.3. See our answer to Q1 below for more.
>
> __"I haven't found any explicit discussion of the limitations of the proposed methods for estimating consistency. Are the presented models realistic? How robust are the results to perturbations to this model?"__
>
> We are unsure what the reviewer means by "the proposed methods for estimating consistency". Our paper does not propose a new method. It describes the behavior of widely-used embedding methods when the embedding dimension is misspecified. For realism and robustness, see Questions 2 and 3 below.
>
> Key Questions
>
> 1) One implication of our results is that choosing the dimension too small may be detrimental to downstream tasks. Consider clustering the nodes of a network. A common approach is to embed the nodes and apply $k$-means. Our proof of Lemma 2.4 should imply that if the embedding dimension is too small, $k$-means provably misclusters a constant fraction of the nodes. Results for when the dimension is too large are far more difficult. Whether the $n^{-1/4}$ convergence rate is sufficient for good downstream performance is task-dependent. A full answer will require a lower-bound accompanying our upper-bound in Theorem 3.3. This will require delicate tools from RMT building on our arguments in this paper. We will include discussion of this point in Section 3, after Theorem 3.3.
>
> 2) As with all models, the realism of assumptions A1 through A7 are application-dependent. A1 through A6 are satisfied by the RDPG (with minor exception of A3; see our answer to Q1 from reviewer Z8ps). Since the RDPG includes the SBM, DC-SBM, and other popular models as special cases, it is clear that practitioners believe that assumptions similar to A1 through A6 are acceptable. A7 is common in the RMT literature and is satisfied by typical weighted network models. As mentioned in the manuscript, A7 does not hold for binary networks, but we believe A7 can be relaxed to include them, with substantial novel RMT work. We explore the case of binary networks through experiments in Section 4.1 and in Appendix A, and the results support our conjecture. We will add more discussion along these lines to the end of Section 2 by extending Remark 2.7.
>
> 3) Please see our responses to Q2 from ANZL and Q3 from Z8ps.

---

> > ### Author Rebuttal · Reviewer_fMQq · 2026-04-03
> >
> > I thank the authors for their detailed response to the questions. I updated my score accordingly.

---

### Decision · Program_Chairs · 2026-04-30

**Decision:**

Accept (regular)

**Comment:**

The paper studies the problem of "recovery" in random dot-product graphs (each vertex has a latent d-dimensional vector and an edge is placed proportional to the inner product of the latent vectors) by spectral methods, when the embedding dimension 'd' is misspecified. The authors show a lower bound (which is along the expected lines-- if the guess for d is too small, it cannot "capture" the embeddings well), but more interestingly, the authors show an upper bound that estimates remain consistent even if the guess is considerably larger than the true 'd'. Since the goal is to bound error in the 2->\infty norm, the paper uses results on eigenvector delocalization that have been developed over the last 10-15 years, in addition to classical spectral analysis.

The reviews are all generally positive about the paper. They note that the results are not too surprising but a worthy "sanity check". They also note some interesting contributions, e.g., the rate at which convergence occurs when the "guess" r is > d is slower than the case of r=d (n^{-1/4} instead of n^{-1/2}).